# A multi-layer perceptron neural network for varied conditional attributes in tabular dispersed data

**Małgorzata Przybyła-Kasperek**[ID]⊛*, **Kwabena Frimpong Marfo**⊛

Institute of Computer Science, University of Silesia in Katowice, Sosnowiec, Poland

⊛ These authors contributed equally to this work.
* malgorzata.przybyla-kasperek@us.edu.pl

**Data Availability Statement:** The data used in the article is publicly available from the UCI repository. The specific links are as follows: Vehicle

## Abstract

The paper introduces a novel approach for constructing a global model utilizing multilayer perceptron (MLP) neural networks and dispersed data sources. These dispersed data are independently gathered in various local tables, each potentially containing different objects and attributes, albeit with some shared elements (objects and attributes). Our approach involves the development of local models based on these local tables imputed with some artificial objects. Subsequently, local models are aggregated using weighted techniques. To complete, the global model is retrained using some global objects. In this study, the proposed method is compared with two existing approaches from the literature—homogeneous and heterogeneous multi-model classifiers. The analysis reveals that the proposed approach consistently outperforms these existing methods across multiple evaluation criteria including classification accuracy, balanced accuracy, $F1$−score, and precision. The results demonstrate that the proposed method significantly outperforms traditional ensemble classifiers and homogeneous ensembles of MLPs. Specifically, the proposed approach achieves an average classification accuracy improvement of 15% and a balanced accuracy enhancement of 12% over the baseline methods mentioned above. Moreover, in practical applications such as healthcare and smart agriculture, the model showcases superior properties by providing a single model that is easier to use and interpret. These improvements underscore the model's robustness and adaptability, making it a valuable tool for diverse real-world applications.

## 1 Introduction

Machine learning (ML) for dispersed data addresses the challenge of analyzing and utilizing data that is scattered across different sources, formats, and locations. This is increasingly important in the era of big data where data is often inconsistent, heterogeneous, and subject to privacy regulations. Data collected from various sources often lack a uniform structure, in that attributes and objects might differ significantly from one data set to another. In the healthcare sector, data on patients are stored across multiple hospitals and medical facilities, each with its

Silhouettes: https://archive.ics.uci.edu/dataset/149/statlog+vehicle+silhouettes Dry Bean: https://archive.ics.uci.edu/dataset/602/dry+bean+dataset Sensorless Drive Diagnosis: https://archive.ics.uci.edu/dataset/325/dataset+for+sensorless+drive+diagnosis Crowd Sourced: https://archive.ics.uci.edu/dataset/400/crowdsourced+mapping.

**Funding:** The author(s) received no specific funding for this work.

**Competing interests:** The authors have declared that no competing interests exist.

own data management system. These data sets might differ in structure, terminology, and format. Additionally, data protection regulations (e.g., HIPAA in the U.S., GDPR in Europe) prevent the sharing of sensitive patient information between institutions without proper safeguards. Suppose multiple hospitals are working together to develop a predictive model for identifying patients who are at high risk of sepsis. Each hospital has its own data set which includes varying attributes such as patient vitals, lab results, and medication histories. The problem is how to collaboratively train a sepsis prediction model leveraging on these diverse data sets. A very important aspect is the ability to make use of inconsistent data that is available in dispersed form, however, due to the arbitrariness of attributes and objects present in local data sets as well as data protection laws that restrict the free flow of data, one has to be meticulous when dealing with dispersed data. ML for dispersed data is crucial in leveraging the full potential of big data across domains where data is fragmented and regulated. It enables organizations to collaboratively develop sophisticated models.

To clarify, certain assumptions have been made that define the area of the problem under consideration. To begin with, there is the assumption that data are available in tabular and dispersed form. Also, data are provided by independent entities that do not want to share data—storing data on a central server with other data from other sources. For this reason, there are a set of local decision tables where objects and conditional attributes of such tables need not satisfy any constraints—they do not have to be equal or same but they may have some shared attributes as well as objects. In the situation considered in the paper, we do not guarantee full confidentiality. We assume that the local entities agree to disclose information about what attributes are stored in the table—the names of these attributes and certain characteristics of the values stored in the table such as mean, median, minimum and maximum of the attributes in the local tables.

Researchers have been seeking for solutions associated with dispersed data in domains such as federated learning and distributed learning. Federated learning puts a greater emphasis on data protection. The general approach here is to distribute an initial model from a central server to all local spaces for the model to be trained locally. Trained parameter values from all local models are then sent to the central server where some aggregations are performed to produce a global model. The global model is then sent back to the local spaces for verification—local units can accept, modify or reject the global model. Such a process is performed iteratively until some acceptable convergence metric is achieved. In [1], a detailed description can be found. For such a global model to be constructed in federated learning, the assumption of an equal set of conditional attributes present in all local tables must be satisfied. Distributed learning on the other hand assumes that all data are available in a centralized form, for example in a single decision table (see [2]). The division into local sets is intentional and aims to improve the quality of the model's classification or its ability to deal with huge data. Often, the process of creating local tables in distributed learning is focused on strengthening the local classifiers by sensitizing them to difficult cases. This approach assumes full access to all data and does not necessarily guarantee any data protection.

The model proposed in this paper is different from the two domains mentioned above. Namely, the proposed model does not impose any assumptions of homogeneity on the form of data present in local spaces whiles guaranteeing a certain level data protection. By sharing data on attribute names and general characteristics of the values stored in the tables, Individual data tuples and individual raw data are protected. Also, the proposed method does not employ an iterative process to reach a consensus but rather a non-iterative algorithm that leads to the construction of a global model.

The main contribution of the paper is to propose a method that generates a global neural network model based on dispersed data. To begin, local neural networks with the same

structure are trained based on local tables where local tables have varied and unrestricted form —no constraints on the set of objects and the set of attributes. In order to generate local networks with the same structure, it is necessary to somehow modify the local tables. The goal is to generate local networks whose input layers considers the full set of conditional attributes present in all local tables. In each of the local tables, values for missing attributes are imputed by using certain characteristics determined based on other local tables containing the missing attributes. In this way, a set of extended local tables are prepared and used to train local MLP networks. This study uses the MLP networks as this is the initial take on the proposed approach, thus, it is appropriate to start with the standard neural network suitable for classification. Other types of networks such as the radial basis function neural networks as well as autoencoders are planned to be used in future work. In the next step, the local neural networks are aggregated. In this study, two approaches of aggregating networks are considered— average and sum of the weights from the local models. Finally, the aggregated model is retrained with a sample of data that is shared and defined for the full set of conditional attributes. In this way the final global model is constructed.

In this paper, the proposed model is investigated in terms of many variants. The following features are tested:

- the method to substitute values of missing attributes in local tables (different number of artificial objects generated based on one original object are tested),

- the number of hidden layers in MLP networks ($k$-hidden layer networks are tested, $k \in \{1, 2\}$),

- the number of neurons in the hidden layer/s (different values are tested),

- the method of aggregating local neural networks (sum and average are tested).

The proposed method is compared with other methods from the literature to establish its quality. Two approaches are adopted as baseline methods. To begin, the ensembles of classifiers proposed in the paper [3]. It comprises creating three base classifiers: $k$-nearest neighbors, decision tree and naive bayes classifier (KNN, DT, NB) based on each local table. The final decision is made by voting. The second approach is a homogeneous ensemble of classifiers and consists of generating MLP networks based on each local table separately and then generating the final decision by voting. It is shown in this paper that the proposed model produces much better results than both baseline models. These differences are also confirmed with statistical tests.

The paper is organized as follows. In the second section, an overview of the literature is included. The third section presents the newly proposed adaptive approach. Here, a formal definition of dispersed data and description of steps of the process of building a global model based on dispersed data is included. The fourth section gives the experimental protocol and description of the data used. The fifth describes obtained results. Comparative analysis are also carried out in this section. Finally, a summary is presented in the conclusion section.

## 2 Related work

Artificial intelligence (AI) has transformed various sectors by integrating human-like abilities such as learning, reasoning, and perception into software systems. This advancement has enabled computers to execute tasks traditionally performed by humans. Fueled by enhancements in computational capacity, the availability of extensive data sets and the creation of state-of-the-art AI algorithms, AI applications have become widespread. Noteworthy examples

include finger vein recognition [4], diabetic retinopathy detection [5], RNA Engineering [6], cancer detection [7], biomathematical challenges [8], and smart agriculture [9].

Ensemble learning deals with distributed data which is similar to the issues in this paper. It is a very popular technique in machine learning that is employed to boost the predictive performance of learning algorithms. The underling reasoning of using this approach is to tackle problems involving data sets that are too large to handle at once [10] or in situations where having access to a very small data set, at which data sampling is necessary to obtain reasonable results [11]. Another rationale for using this approach may be to cope with the issue of identifying the right model for the considered problem [12]. To expound, rather than risking selecting the wrong model, one can use a heterogeneous approach of ensemble learning. This approach also works well for problems whose solution space is quite large, thus, faces the risk of getting stuck in local minima/maxima [13]. Many different approaches involving the use of neural networks to address the above mentioned problems have been proposed. Such solutions are proposed in areas such as the business field [14], malware detection [15] and audio classification [16], however, all these approaches assume free access to data and a necessary condition that all data is stored in a centralized form rather than a dispersed one.

Federated learning is another approach within distributed machine learning [1]. Different from classifier ensembles, it puts the greatest emphasis on data segregation and protection [17]. Here, the assumption is that data are available in separate sets that must not be centralized. The idea is to build local models separately and generate a global model in a central space by iteratively aggregating the local models. Neural networks are well applicable here as it is relatively simple to aggregate these models while maintaining high quality [18, 19]. There are types of federated learning: horizontal, vertical and hybrid federated learning. The latter approach is the closest to the approach proposed in this paper, however, unlike the proposed approach, hybrid federated learning requires that different parties share the data identity information which is a threat to the privacy of local clients [20]. Unfortunately, for the considered data sets, it is impossible to apply this approach due to the hybrid nature of the partitioning—regarding both objects and attributes—and the inability to obtain identity information about objects between dispersed data sets. Many different models are proposed in federated learning with various aggregation methods, network types and applications being considered in the literature [21–23].

Another approach to the problem of classification based on dispersed data is to build a separate model that aggregates prediction vectors generated by independent local models. Data privacy is also preserved here as only prediction vectors are consolidated. The form of the data can be completely arbitrary in this approach but here, a global model is not generated and the algorithm is non-iterative. Instead, it generates a separate model that only aggregates the prediction results obtained by local models. The local models can be of a completely different type than the aggregation model. In the literature, one can find papers that use neural networks, decision trees or other models as the aggregation model [24–28]. Statistical as well as dynamic approaches to this issue are also proposed which also consider conflicts or compatibility of local classifiers [29–31]. However, in the present study, the approach considered is different as the goal is to determine a global model based on dispersed data.

MLPs have been key in developing neural networks and machine learning. Although more complex models like Convolutional Neural Networks (CNNs) and Transformers have emerged, recent improvements have renewed the importance and usefulness of MLPs particularly where simplicity and efficiency are needed. Techniques such as Adam and RMSProp [32] have enhanced MLP training by dynamically adjusting learning rates, leading to faster convergence and improved generalization. Incorporating residual connections within MLPs akin to ResNet architectures [33] has mitigated the problem of gradient vanishing, enabling the

training of deeper MLP models. MLPs traditionally require large amounts of labeled data to perform effectively. Techniques such as data augmentation and transfer learning are being adapted to address this limitation [34]. Some of the techniques mentioned above (e.g Adam optimizer [35]) are used in this paper for MLP. But, to the best of our knowledge, MLP networks have never been used in the way that is proposed in this paper—for dispersed data with different sets of attributes using augmentation of missing attribute values.

## 3 Basic concepts and proposed global model

In this section, we present preliminary designations as well as a detailed discussion on the proposed method for generating a global MLP network model based on dispersed data.

### 3.1 Dispersed data

A necessary assumption made is that data are available in a dispersed form—separate independent predefined data sets which are free of any constraints. In real applications, independent units collect data in tabular form. In tables, both sets of conditional attributes and sets of objects do not necessarily have to be disjoint as they may share common elements.

Also, there is an assumption that a set of decision tables is given. The tables are collected independently by separate units. A set of decision tables—local tables $D_i = (U_i, A_i, d) i \in \{1, \ldots, n\}$ from one discipline is available, where $U_i$ is the universe, a set of objects; $A_i$ is a set of conditional attributes and $d$ is a decision attribute. Decision tables are collected independently so both sets of objects and sets of attributes can have any form. They can have common elements between tables, but not necessarily. The only condition that must be satisfied by all local tables is the collection of data from one discipline. Formally, this is satisfied by the assumption that the same decision attribute is present in all tables.

Since different sets of attributes appear in local tables, the construction of a MLP local model based on each of the tables separately would create a set of networks with completely different structures. This is because the input layer in each neural network would be different since the feature vectors are not the same across the local tables, thus, making it impossible to aggregate local MLP networks into a single global model.

The approach proposed in this paper is completely different from previous studies as it has not been proposed in the literature until now. The steps of the approach are listed below.

1. Determine a uniform MLP network structure for a set of local tables—dispersed data;

2. Train a MLP network based on each local table separately.

3. Aggregate MLP networks into a single model—a global MLP network;

4. Post-train the global MLP network with a sample of global data.

Fig 1 shows the general steps of building the global MLP network model from dispersed data. In the first step, there is dispersed data—local tables with different sets of conditional attributes and different sets of objects. In order to build local neural networks with the same structure (the input layer requires the most attention here), the training data in each local space is imputed so as to have the same set of attributes. This step is carried out with the help of certain characteristics calculated from local tables. It is important to emphasize that the raw data is not shared at any model construction stage. In the next step, local MLP networks are trained, after which they are aggregated to construct a global network. The final step is to retrain the global network. In the study, this is done using a validation set.

All the steps are discussed in detail in the subsequent subsections.

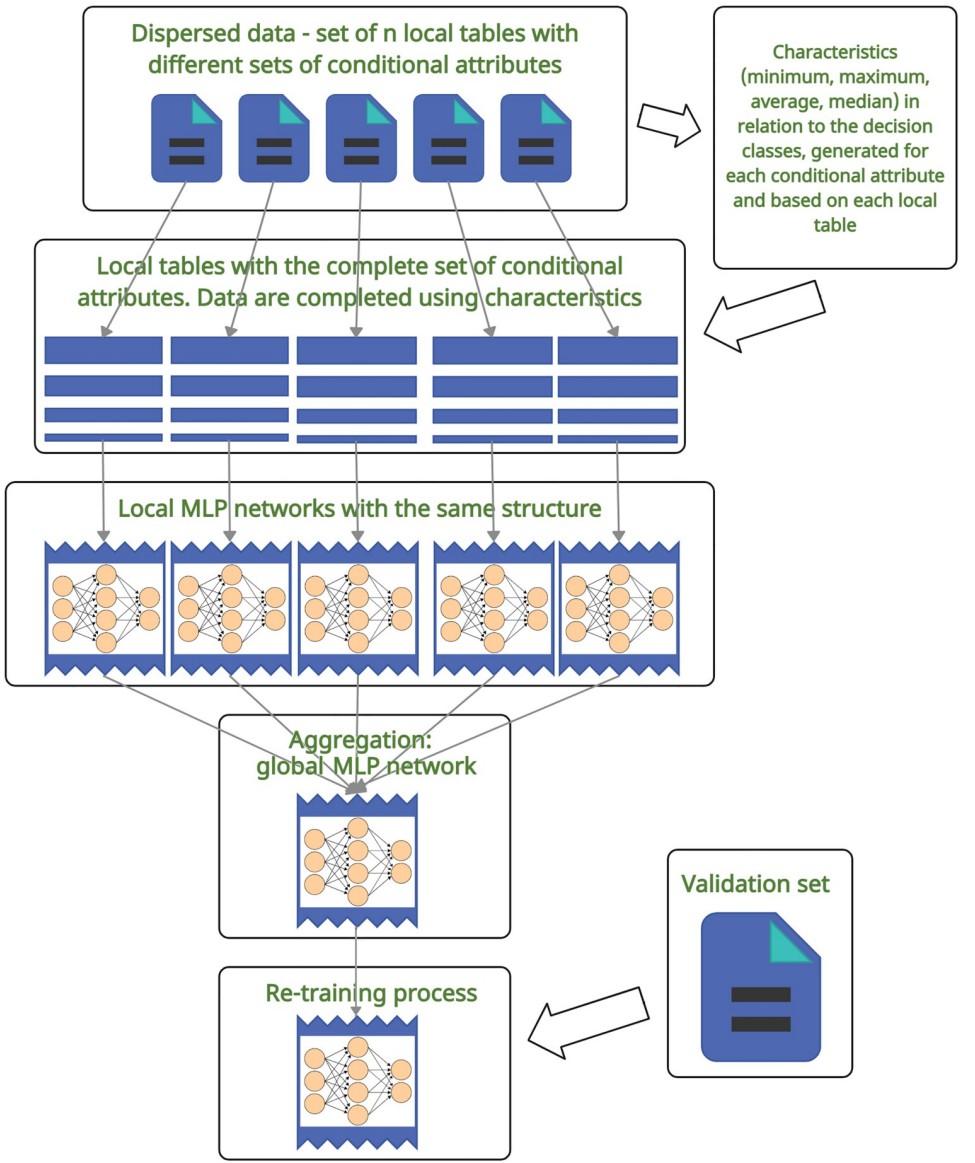

**Fig 1. Stages of building the global MLP network model based on dispersed data.**

## 3.2 Determine an uniform MLP network structure for a set of local tables—local models

Since the dispersed data need not satisfy any constraints, the key in determining the structure of the MLP network is in the number of neurons in the input layer. The output layer poses no problem since all local tables share the same decision attribute. The number of hidden layers as well as the number of neurons in the hidden layers are optimized experimentally. Thus, the most important challenge is to determine a common input layer. In this first study on the approach, it is proposed to unify the input layer by using all conditional attributes from local tables. So the input vector will have the dimension determined by the number of elements in

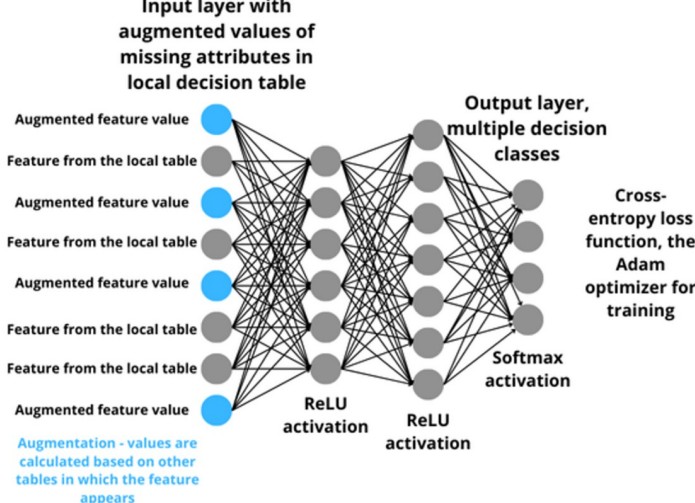

**Fig 2. Schematic diagram of MLP network structure for one local decision table.**

the sum of conditional attributes present in the local tables

$$\text{Number of neurons in the input layer} = \text{card}\Big\{\bigcup_{i=1}^{n} A_i\Big\},$$

where card{$X$} is the number of objects in the set $X$. Such a sum is not a simple concatenation of attributes. We operate on sets, and we recognize attributes by their names. So the sum of the sets skips multi-duplicates—in case when one attribute appears in several tables it only appears once in the sum. It should also be noted that such a sum does not mean summing tuples from a table, but only determining the set of names of all attributes appearing in local tables.

Here a problem arises because local tables contain objects for which values are known only on a certain subset of the set $\bigcup_{i=1}^{n} A_i$. The question arises on how to train the local MLP networks with the input layer defined as above based on a local table with such objects. Fig 2 shows the overall configuration of the MLP network—local model used for each local decision table. In each of the local tables, a certain number of attributes (features) are included but not all of them. In order to make the network structure common for all local tables, the completion of missing values for a given local table is made. Of course, in each local table other missing values may occur. Completion of missing values is carried out by calculating values from local tables in which the attribute occurs. Local models are neural networks trained specifically on artificially created objects. That is, those that have completed values on attributes that are not present in the actual given local table. So only these artificial objects are used to train the neural network, the original objects are not used. The training process for these models involves a standard neural network built using the Keras library in Python, employing backward propagation over multiple epochs and steps within each epoch. In the next section, an explanation on how this problem is solved is given.

### 3.3 Training a MLP network based on each local table separately

In this section, the explanation on how to train a local MLP network based on a local table is given. Let us assume that a local table $D_j = (U_j, A_j, d)$ is given, based on which a local MLP network is to be trained with an input layer containing card$\{\bigcup_{i=1}^{n} A_i\}$ neurons. For an object $\bar{x} \in$

$U_j$ from the local table $D_j$, values for attributes from the set $A_j$ are specified, which means for each $a \in A_j$ value $a(\bar{x})$ is given. Thus, in order to provide an input vector to the MLP network, the values on the other attributes from the set

$$\bigcup_{i=1,i \neq j}^{n} A_i$$

must be determined. Let us assume that attribute $b$ belongs to the set $\bigcup_{i=1,i \neq j}^{n} A_i$ and for this attribute one has to determine the value to be completed for the object $\bar{x}$. In the proposed approach, this value is determined based on certain statistical measures: minimum, maximum, median and average calculated for values of attribute $b$ occurring in other local tables in the dispersed data. In addition, the decision class of the object $\bar{x}$ is also taken into account. These measures were chosen as the most popular, frequently used in numerous calculations and characterize both the central tendency and the entire range of variation in the value of a given attribute. The paper is the first study of the approach using artificial objects. In future work, other statistical measures will be analyzed. It is planned to use quartiles and the average value offset by the standard deviation.

More strictly, let us assume that the object $\bar{x}$ has a decision value $v$, $d(\bar{x}) = v$, $v \in V^d$, where $V^d$ is the set of values of the decision attribute $d$. For each of the decision tables to which the attribute $b$ is present, the minimum, maximum, median and average are calculated for the values of the attribute $b$ based on the objects in the decision class $v$. For each decision table $D_i$ for which $b \in A_i$ the following values are computed:

$$MIN_{i,v}^{b} = min_{x \in U_i, d(x)=v} \; b(x), \quad MAX_{i,v}^{b} = max_{x \in U_i, d(x)=v} \; b(x),$$

$$AVG_{i,v}^{b} = avg_{x \in U_i, d(x)=v} \; b(x), \quad MED_{i,v}^{b} = median_{x \in U_i, d(x)=v} \; b(x)$$

In this way, values designated separately for each local table containing the attribute $b$ are obtained. To determine the final value which is completed in the object $\bar{x}$ and given to the input of the neural network, one of the statistical measures (minimum, maximum, mean or median) is applied on the local values determined in the previous step. Thus, one of the four measures for determining local values based on local tables and one of the four measures for determining the aggregate value. In all, there are 16 possible combinations from which one is chosen at random as the value of $b$. Suppose that for calculating local values the median is drawn, and for aggregate value, the minimum is drawn, then the value on attribute $b$ is determined as follows

$$b(\bar{x}) = min_{D_i : b \in A_i} MED_{i,v}^{b}.$$

This method is repeated for each of the missing attributes $\bigcup_{i=1,i \neq j}^{n} A_i$ for object $\bar{x}$.

In the generalized version of the above method, instead of one object, $k(k <= 16)$ objects are generated by selecting $k$ distinct values from the 16 possible values as the value of $b$ in each of the $k$ objects generated from the original object $\bar{x}$. Thus, based on object $\bar{x}$, $k$ new objects would be generated with all values on conditional attributes $\bigcup_{i=1}^{n} A_i$. This approach is also tested and the results are presented in the experimental part of the paper. Algorithm 1 presents the pseudo-code of the generalized version (in the basic version, it is enough to put $k = 1$), which implements this part of the model.

**Algorithm 1** Pseudo-code of algorithm generating objects from one local table used for training the local MPL network

```
Input: One local decision table Dⱼ = (Uⱼ, Aⱼ, d) for which we determine
the training set for the MLP network; measures minimum, maximum,
median and average MINᵇᵢ,ᵥ, MAXᵇᵢ,ᵥ, MEDᵇᵢ,ᵥ and AVGᵇᵢ,ᵥ computed for each deci-
sion value v ∈ Vᵈ and attribute b ∈ Aᵢ based on the values stored in
the table Dᵢ for each i ∈ {1, ..., n}; A = ⋃ⁿᵢ₌₁ Aᵢ a set of conditional
attributes from all local tables; k parameter value that determines
how many objects are generated based on one object from table Dᵢ.
Output: A data set used to train the MLP neural network, D̄ⱼ = (Ūⱼ, A, d).
foreach x ∈ Uⱼ
  for m = 1 to k do
  create an object x̄ from D̄ⱼ by assigning values on the set Aⱼ the same
as the object x has
    foreach attribute in the set b ∈ A\Aⱼ
      choose a pair (choice1, choice2) from the set
      {MIN, MAX, AVG, MED} × {MIN, MAX, AVG, MED}
      b(x̄) = choice2ᵢ:ᵦ∈Aᵢ(choice1ᵇᵢ,d(x))
    end foreach
  end foreach
end foreach
```

The computational complexity of the above method is linearly dependent on the number of objects in the local table $D_j$, value of parameter $k$, the number of conditional attributes card$\{A\}$ and the number of local tables in the dispersed data $n$. More precisely, the complexity resulting from the loop is $O(\text{card}\{U_j\} \cdot k \cdot \text{card}\{A\backslash A_j\} \cdot \text{card}\{D_i, i = 1, \ldots, n\})$. In the worst case, one can assume that there is only one conditional attribute in the table $D_j$, and for all other attributes, values have to be computed and the missing attributes are present in all other local tables except $D_j$. Then the complexity is $O(\text{card}\{U_j\} \cdot k \cdot (\text{card}\{A\} - 1) \cdot (n - 1))$. The linear complexity of the algorithm proves that it can be used even for large dispersed data.

The data prepared in the above way in the next step is used for training MLP neural networks. As mentioned earlier, the input layer is defined by a set of conditional attributes from all local tables. The number of neurons in the output layer is equal to the number of decision classes. Each of the neurons determines the probability with which the test object belong to a given decision class. In the experimental part, one or two hidden layers are considered. The number of neurons in the hidden layer is determined in proportion to the number of neurons in the input layer. Different proportions are checked from 0.25 to 5 times the number of neurons from the input layer. In the case of two hidden layers, all combinations of the number of neurons in the hidden layers are checked such that: the first layer had the number of neurons from the set $\{0.25 \times I, 0.5 \times I, 0.75 \times I, 1 \times I, 1.5 \times I, 1.75 \times I, 2 \times I, 2.5 \times I, 2.75 \times I, 3 \times I, 3.5 \times I, 3.75 \times I, 4 \times I, 4.5 \times I, 4.75 \times I, 5 \times I\}$, and the second layer had the number of neurons from the set $\{1 \times I, 2 \times I, 3 \times I, 4 \times I, 5 \times I\}$ where $I$ is the number of neurons in the input layer. For the hidden layer, the ReLU (Rectified Linear Unit) activation function is used, as it is the most popular activation function and gives very good results [36]. For the output layer, the softmax activation function is used, which is recommended when one deals with a multi-class problem [37]. In this paper, data sets containing from four to nineteen decision classes are analyzed. The neural network is trained by using the back-propagation method. A gradient descent method, with an adaptive step size is used in the back-propagation method. It is known that the softmax layer give good results with the Adam optimizer [35]. The Adam optimizer proposed in [38] and is one of the most popular adaptive step size methods. From [39], the categorical cross-entropy loss gives best results with softmax layer. That is why the Adam optimizer and the categorical cross-entropy loss function are used in the study.

The implementation of the MLP neural network from Keras library in Python is used. The algorithm that defines a neural network with one or two hidden layer with the rectified linear

unit (ReLU) activation function and the number of neurons in the first hidden layer dependent on the parameter. Softmax activation function is used in the output layer. In the compilation, the categorical cross-entropy loss function, the Adam optimizer and the accuracy as the learning rate are used. For two hidden layers approach the second hidden layer with the ReLU activation function and the number of neurons dependent on the parameter is used. In the way described above, a set of local MLP networks are obtained. The number of networks is equal to the number of local decision tables. All networks have the same structure and this is a very important property necessary for the next step.

## 3.4 Aggregation of MLP networks into a single model—a global MLP network

The result of the previous stage is a set of local MLP networks which are trained and all have the same structure. Aggregation of such networks into a single global MLP model is relatively simple. The global network has exactly the same structure as each of the local networks i.e. the same number of layers and the same number of neurons in each layer. However, during aggregation, each local model may have a different impact on the construction of the global model. This influence is proportional to the quality of each local model's classification on the training set. The method used is inspired by the second weighting system used in the AdaBoost algorithm [40].

For each local model, a classification error is estimated based on its training set (artificial objects generated using Algorithm 1). Let us denote by $e_i$ the classification error determined for the $i$−th local model $i \in \{1, \ldots, n\}$. Since local models are built based on a piece of data, their accuracy can be very different. It may sometimes happen that their classification error is above 0.5. In order not to eliminate such local models from the aggregation stage as they may contain important information on specific attributes that may have a positive impact in the global model, the min-max normalization is applied to the interval [0, 0.5] of all errors $e_i$, $i \in \{1, \ldots, n\}$. After, the weights $\omega_i$ for each local neural network $i \in \{1, \ldots, n\}$ is adjusted according to the formula:

$$\omega_i = ln\left(\frac{1 - e_i}{e_i}\right) \tag{1}$$

The weights of global model are determined by one of two approaches: in the first approach, the weights for the global network are determined by the weighted average of the corresponding weights (assigned to edges connecting exactly the same neurons) present in local MLP networks with weights $\omega_i$, $i \in \{1, \ldots, n\}$. The second approach is to determine the weight for the global network as the sum of the corresponding weights from the local networks with weights $\omega_i$, $i \in \{1, \ldots, n\}$. The two approaches are studied separately in the experimental part of the paper.

Fig 3 illustrates the process of aggregating local models into a global model. Since all local models share the same structure, this aggregation is relatively straightforward. Each connection between neurons in the global model corresponds to the connections in the local models. The critical aspect of this process is the determination of weights, which are based on the classification performance of local models on their respective sets of artificial objects (training sets). The weights assigned to each local model are crucial as they influence the global model's configuration. Local models that perform poorly in classification (possibly due to a higher number of missing attributes and thus less connection with reality, more values are artificial) are given smaller weight in shaping the global model. However, they are not entirely excluded from the aggregation. It will still contribute to the overall classification performance of the

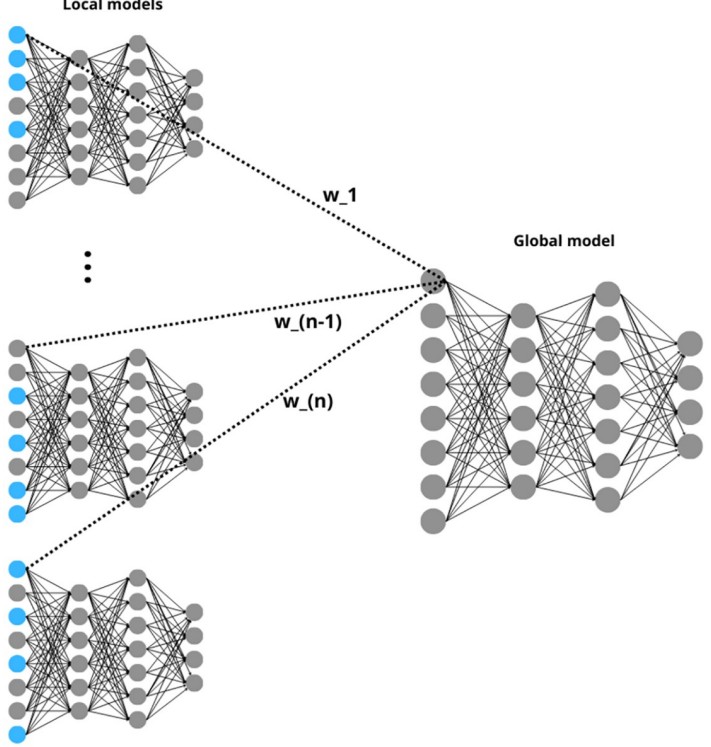

**Fig 3. Schematic diagram of global model.**

global model. This ensures that the global model benefits from the specialized capabilities of each local model, enhancing its overall classification quality.

The implementation of the global MLP network is done in Python. First the network's structure is defined—the number of layers and neurons in the layers are the same as in local networks. Then the weights are not trained but assigned based on average or sum with consideration of the weights for the local networks of corresponding connections in local networks.

## 3.5 Re-training the global MLP network with a sample of data

The retraining process with global objects is a step that enhances the proposed model's accuracy, generalization, and robustness. By carefully integrating and fine-tuning the local models, the global model achieves superior performance, making it a valuable tool for various real-world applications. This process integrates local models into a cohesive global model, enhancing overall accuracy and generalization. The global MLP network is re-trained using the validation set. This step involves adjusting the weights and biases of the aggregated model to fine-tune it for better performance. The retraining process ensures that the global model leverages the strengths of the local models while mitigating their individual weaknesses. A validation set, which is a subset of the training data, is used for the retraining process. The size of this validation set is smaller than the local models' training sets but is crucial for capturing the model's generality. The validation set helps in fine-tuning the global model to prevent overfitting and ensure it generalizes well to unseen data. What is important is that the objects in such a validation set must contain a global description of the objects, i.e. include attributes/characteristics present in all local tables. The integration of local models through retraining results in a

significant boost in classification accuracy. The model benefits from the collective knowledge of all local datasets, leading to more accurate predictions.

The last stage is to re-train the global network. The training objects needed in this step should have values on the set of all conditional attributes $\bigcup_{i=1}^{n} A_i$. In the paper, this is implemented by using a validation set. Such a validation set is much smaller than the training sets for local models and will have less influence on the final form of the global neural network. However, without the use of this last step, the obtained quality of classification is unsatisfactory and we miss to capture a generality of the model. For the approach of generating one artificial object, the size of a validation set is about 21% of the size of the training set for local model. In the case of generating three artificial objects, the size of a validation set is about 7% of the size of the training set for local model. In future works, it is planned to test the active learning approach [41, 42] instead. In active learning, the assumption is that the model builds its own training data or changes the original training data.

After the completion of this step, the final form of the global model is obtained and is evaluated by using an independent test data set.

It should be noted that the model avoids overfitting through a series of carefully planned steps. The final stage of training involved re-training the global MLP network with a validation set. This validation set is smaller than the training sets for local models, but crucial in capturing the generality of the model. For generating one artificial object, the validation set os about 21% of the local model's training set size. For three artificial objects, it is about 7%. This step is essential to prevent overfitting and ensure the model could generalize well to new, unseen data. Also during the selection of optimal parameters, we aimed for the best classification accuracy with the lowest possible model complexity, involving the fewest layers and neurons. This focus on simplicity helped in reducing the risk of overfitting.

## 4 Experimental setup

In order to assess the efficiency of the suggested model, the methodology of the experiment is analyzed within this section. The simulation platform, parameter allocation, and criteria for measuring performance are all elaborated upon. The scheme for describing the experimental methodology, which is widely used in the literature as shown in the work of [43, 44] is used below.

### 4.1 Simulation platform

All simulation is conducted using an open-source software Jupyter Notebook 6.5.4 and Anaconda 2023.09-0 (Sep 29, 2023) Installer Python Version: 3.11.5. The implementation of the proposed model is made using the Keras library in Python. The simulations were run on a computer with an Intel(R) Xeon(R) W-2235 CPU @ 3.80GHz 3.79 GHz processor and 32.0 GB RAM to avoid any bias in the analysis of the results, it is crucial that the study is conducted using the same compiler, on the same computing hardware, and with the same processing capabilities.

### 4.2 Data set

The experimental study uses data sets available in the UC Irvine Machine Learning Repository: Vehicle Silhouettes [45], Dry Bean [46], Sensorless Drive Diagnosis [47] and Crowd Sourced [48]. The characteristics of the data sets are given in Table 1.

Each of the data sets are originally available in non-dispersed form—each data in a single decision table. The training set are dispersed where different degrees of dispersion are considered. Each single data set is converted into five different dispersed versions: 3, 5, 7, 9 and 11

**Table 1. Basic characteristics of data sets.**

| Data set | No of examples in training set | No of examples in test set | No of attributes | No of decision classes |
|---|---|---|---|---|
| Vehicle Silhouettes | 592 | 254 | 18 | 4 |
| Dry Bean | 9527 | 4084 | 17 | 7 |
| Sensorless Drive Diagnosis | 38509 | 20000 | 49 | 11 |
| Crowd Sourced | 10546 | 300 | 29 | 6 |

local tables respectively. During the construction of the local tables, a subset of attributes in each local table is considered. The number of attributes is significantly reduced in local tables as compared to the original table with some attributes repeating among some tables to satisfy. This is done to make provision for the possibility that some local tables may share common attributes. The full set of objects is stored in each local table but without their identifiers. More precisely, the number of local tables is first determined (e,g dispersed version with 5 local tables). Then the number of original set of conditional attributes is divided evenly among the local tables (so that each local table had more or less the same number of attributes). In addition, it is assumed that there are common attributes between the selected local tables, e.g. between table one and two we have two common attributes, between table two and three we have one common attribute, and so on. With the initial assumptions made, attributes are then randomly distributed between local tables. Once we have established sets of attributes in local tables then entire columns from the original tables are rewritten into local tables. In this way, we have the same sets of objects in all local tables.

The Sensorless data set is balanced with each decision class containing 5319 objects. The Vehicle, Dry Bean and Crowd Sourced data sets are imbalanced (Fig 4). The data are balanced but it is worth emphasizing that this process is carried out after the dispersion (to keep the approach as consistent with the real situation as possible). The Synthetic Minority Over-sampling Technique (SMOTE) method is used [49] for each local decision table separately. The implementation of this algorithm available in WEKA [50] software is used. The data considered are multiclass labeled so in each decision table and for each decision class except the most dominant one, the SMOTE method is used. As a result, all decision classes have the same number of objects after balancing. Finally, for each of the three original data sets, 5 dispersed versions of imbalanced data and 5 dispersed versions of balanced data are obtained. Thus, a total of 35 dispersed data are considered in the experimental part.

## 4.3 Parameter assignments

The proposed model comprises of three phases: structure phase, training phase, and testing phase. The structure phase involves determining the structure of local and global MLP models. The input layer of the model is strictly dependent on the data set—the number of neurons is

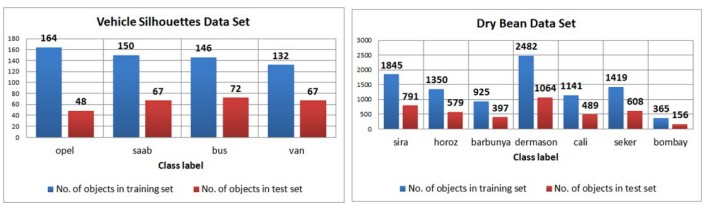

**Fig 4. Imbalance of data—cardinality of decision classes in training and test sets.**

equal to the number of attributes present in all local tables. The same is true of the output layer —the number of neurons is equal to the number of decision classes. However, as for the other parameters of the network, it is variable and determined experimentally. Also, the method of determining the value of missing attributes, as well as the number of artificial objects used is variable. Another parameter of the model is the method of aggregation of local networks. Different parameter values are tested. We conducted a comprehensive grid search to explore various configurations of the hyperparameters. This involved testing different combinations of the number of hidden layers, the number of neurons in each layer, methods for aggregating local neural networks, and strategies for handling missing attributes. By systematically varying these parameters, they are able to identify the optimal configuration that achieved the best performance. The optimal number of hidden layers and neurons in each layer is determined through the grid search. The authors tested various configurations, ranging from shallow networks with fewer layers to deeper networks with more layers and neurons. The chosen configuration provided the best trade-off between model complexity and classification accuracy. Different methods for aggregating the local neural networks are evaluated. The authors considered techniques such as averaging the weights of the local models and summing the weights. By employing a systematic and thorough approach to hyperparameter optimization, the authors ensured that their model is both robust and efficient. The experiments are carried out according to the following scheme:

- Different approaches to substitute values of missing attributes in local tables are studied— one or three artificial objects are generated based on one original object in local table.

- Different approaches to aggregating local neural networks are studied—using average of weights or sum of weights.

- Different numbers of hidden layers in the local and global networks are studied—one or two hidden layers.

- Different numbers of neurons in hidden layers are studied. The number is determined in proportion to the number of neurons in the input layer. The following values are tested: for the first hidden layer {0.25, 0.5, 0.75, 1, 1.5, 1.75, 2, 2.5, 2.75, 3, 3.5, 3.75, 4, 4.5, 4.75, 5} $\times$ the number of neurons in the input layer; for the second hidden layer {1, 2, 3, 4, 5} $\times$ the number of neurons in the input layer.

So in total, 384 different experiments for the proposed method are conducted—384 different settings of the approaches analyzed ($2 \cdot 2 \cdot 16 + 2 \cdot 2 \cdot 16 \cdot 5$). In the tables in Section 5, we show both the results from each parameters settings, as well as the optimal parameter values. The optimal parameters are chosen as those provide the best classification accuracy with the lowest possible model complexity—the lowest number of layers and the lowest number of neurons used in the model.

## 4.4 Formulations of the performance metrics

The quality of classification are evaluated based on the test set. A classification accuracy measure ($acc$) is used for this purpose. That is, a fraction of the total number of objects in the test set that are classified correctly. As is mentioned in the previous section, in the final step the aggregated model is re-trained. To do this, the use a validation set containing objects that have values on all conditional attributes present in the dispersed data is used—attributes occurring in all local tables. The validation set is obtained by dividing the original test set randomly but in a stratified manner into two equal parts. First, one part is used as the validation set (for re-training process) and the second part is used to assess the quality of classification. Then the

roles reverses as the second part acts as the validation set. Finally, both results are averaged. Each experiment is repeated three times; in the following section, all results given are the average of these three runs.

## 4.5 Reproducibility of the proposed model

The structure phase aims to prepare the local MLP neural networks with a consistent architecture. This involves identifying common attributes among the local tables and addressing any missing attributes. The next step involves supplementing the local tables with additional objects, assigning values to the missing attributes. Following this, the training phase optimizes the network weights. During the testing phase, the model's accuracy is validated on test objects with values defined for all attributes. To assess the effectiveness of the proposed model, a comparative analysis is performed against baseline training methods. Given the inherent randomness in the structure phase—where missing values are filled and new objects are created—and the training phase—where initial network weights are set randomly—the experiments are repeated three times and the results are averaged. The simulations are configured as follows to facilitate reproducibility. Benchmark data that is publicly available are used and the process of splitting (performed only once) into local tables is done as described as in Section 4.2. The local tables are then augmented with additional values on missing attributes. One setting of all parameters is selected. The experiments are performed three times and results are averaged. Such a procedure is repeated for all other parameter settings using the same predefined local tables.

## 4.6 Baseline methods

In the literature, there are no models dedicated for dispersed data to generate a single model based on local tables with different attributes (but some of them are common). Therefore, for comparison, an intermediate approach is used, which, although does not generate a global model and does not resolve differences between attributes, it generates local models and performs global classification by voting. In the paper, two approaches for building a local model are used.

- The first approach is an ensemble of homogeneous classifiers, where the base classifiers are MLP networks. However, it should be noted here that each network generated for local tables has a different structure as the input layer is different. Which means that no unification of the input layer is done by filling in the values of missing attributes. For a single local table, an MLP network is created, which in the input layer has neurons corresponding to attributes occurring exactly in that local table. In order to maintain transparency and integrity numbers of neurons in the hidden layer are tested as for the proposed method. The number of neurons in the output layer is the same in all local models as it is equal to the number of decision classes. The final decision of the ensemble is made by soft voting.

- The second approach is the method proposed in [3]. This ensemble of classifiers method consists of creating three base classifiers: $k$−nearest neighbors, decision tree and Naive Bayes classifier (KNN, DT, NB) based on each local table. The parameter $k = 3$ and the Gini index as a splitting criterion when building decision trees are used. Thus, three classifiers are defined for each local table. The final decision of the ensemble is also made by soft voting.

Both approaches are implemented in the Python programming language using implementations available in the sklearn library.

**Table 2. Results of classification accuracy *acc* for the proposed approach: One hidden layer, the method to substitute values of missing attributes in local tables— One artificial objects generated based on one original object, MLP networks aggregation using average of weights and various number of neurons in the hidden layer (1AO-1HL-AVG).** Designation I is used for the number of neurons in the input layer.

| Data set | No. tables | No. of neurons in hidden layer | | | | | | | | | | | | | | | |
|---|---|---|---|---|---|---|---|---|---|---|---|---|---|---|---|---|---|
| | | 0.25 × I | 0.5 × I | 0.75 × I | 1 × I | 1.5 × I | 1.75 × I | 2 × I | 2.5 × I | 2.75 × I | 3 × I | 3.5 × I | 3.75 × I | 4 × I | 4.5 × I | 4.75 × I | 5 × I |
| Vehicle imbalanced | 3 | 0.283 | 0.612 | 0.659 | 0.635 | 0.644 | 0.634 | 0.677 | 0.675 | 0.672 | 0.651 | 0.668 | 0.671 | **0.693** | 0.686 | 0.671 | 0.688 |
| | 5 | 0.51 | 0.665 | 0.629 | 0.663 | 0.678 | 0.639 | 0.677 | 0.681 | 0.698 | 0.685 | 0.673 | 0.689 | 0.681 | **0.703** | 0.681 | 0.665 |
| | 7 | 0.283 | 0.682 | 0.639 | 0.672 | 0.661 | 0.688 | 0.655 | 0.686 | 0.686 | 0.677 | 0.678 | 0.676 | 0.69 | **0.69** | 0.661 | 0.669 |
| | 9 | 0.283 | 0.623 | 0.656 | 0.542 | 0.648 | 0.646 | 0.652 | 0.655 | 0.669 | 0.654 | **0.677** | 0.656 | 0.667 | 0.669 | 0.648 | 0.655 |
| | 11 | 0.283 | 0.487 | 0.617 | 0.66 | 0.643 | 0.647 | 0.66 | 0.623 | 0.676 | 0.664 | 0.652 | 0.657 | 0.657 | 0.669 | 0.648 | **0.696** |
| Vehicle blanced | 3 | 0.584 | 0.665 | 0.642 | 0.682 | 0.688 | 0.711 | 0.703 | 0.713 | 0.694 | 0.714 | 0.707 | 0.723 | 0.718 | 0.724 | 0.69 | **0.727** |
| | 5 | 0.283 | 0.673 | 0.682 | 0.65 | 0.66 | 0.657 | 0.69 | 0.677 | 0.664 | 0.675 | 0.664 | 0.688 | 0.68 | **0.703** | 0.682 | 0.693 |
| | 7 | 0.283 | 0.697 | 0.688 | 0.692 | 0.705 | 0.701 | 0.714 | 0.699 | 0.709 | 0.717 | 0.715 | 0.72 | 0.698 | 0.714 | 0.714 | **0.722** |
| | 9 | 0.283 | 0.608 | 0.689 | 0.669 | 0.686 | 0.711 | 0.698 | 0.709 | 0.697 | 0.688 | 0.713 | 0.724 | 0.705 | 0.682 | 0.713 | **0.73** |
| | 11 | 0.283 | 0.6 | 0.663 | 0.654 | 0.665 | 0.675 | 0.668 | 0.665 | 0.682 | 0.652 | 0.686 | 0.668 | 0.684 | 0.655 | 0.669 | **0.688** |
| Dry Bean imbalanced | 3 | 0.841 | 0.888 | 0.89 | 0.895 | 0.894 | 0.897 | 0.9 | 0.904 | 0.905 | 0.903 | 0.907 | 0.908 | 0.907 | 0.909 | 0.909 | **0.909** |
| | 5 | 0.882 | 0.883 | 0.888 | 0.891 | 0.892 | 0.899 | 0.899 | 0.898 | 0.902 | 0.899 | 0.904 | 0.903 | 0.906 | 0.908 | 0.905 | **0.908** |
| | 7 | 0.879 | 0.884 | 0.889 | 0.892 | 0.896 | 0.897 | 0.898 | 0.899 | 0.899 | 0.897 | 0.902 | 0.904 | 0.902 | 0.903 | 0.907 | **0.908** |
| | 9 | 0.845 | 0.886 | 0.891 | 0.892 | 0.894 | 0.897 | 0.899 | 0.899 | 0.901 | 0.902 | 0.904 | 0.904 | 0.903 | 0.905 | **0.906** | 0.905 |
| | 11 | 0.864 | 0.887 | 0.886 | 0.893 | 0.894 | 0.895 | 0.898 | 0.899 | 0.9 | 0.901 | 0.901 | 0.904 | 0.905 | 0.905 | 0.904 | **0.906** |
| Dry Bean balanced | 3 | 0.87 | 0.894 | 0.898 | 0.898 | 0.904 | 0.902 | 0.902 | 0.908 | 0.904 | 0.907 | 0.908 | 0.906 | 0.91 | 0.911 | **0.913** | 0.911 |
| | 5 | 0.883 | 0.889 | 0.888 | 0.891 | 0.896 | 0.9 | 0.899 | 0.901 | 0.905 | 0.902 | 0.905 | 0.906 | 0.909 | 0.907 | 0.908 | **0.909** |
| | 7 | 0.884 | 0.887 | 0.886 | 0.89 | 0.894 | 0.898 | 0.897 | 0.897 | 0.901 | 0.901 | 0.904 | 0.901 | 0.906 | 0.905 | 0.904 | **0.907** |
| | 9 | 0.882 | 0.884 | 0.889 | 0.891 | 0.894 | 0.896 | 0.897 | 0.901 | 0.899 | 0.902 | 0.901 | 0.902 | 0.903 | 0.904 | 0.906 | **0.909** |
| | 11 | 0.867 | 0.886 | 0.885 | 0.891 | 0.894 | 0.893 | 0.896 | 0.899 | 0.898 | 0.898 | 0.9 | **0.904** | **0.904** | **0.904** | 0.903 | 0.902 |
| Sensorless | 3 | 0.883 | 0.895 | 0.899 | 0.903 | 0.912 | 0.909 | 0.911 | 0.917 | 0.913 | 0.918 | 0.915 | 0.914 | 0.917 | 0.921 | 0.918 | **0.921** |
| | 5 | 0.857 | 0.887 | 0.898 | 0.899 | 0.905 | 0.903 | 0.908 | 0.909 | 0.913 | 0.914 | 0.912 | 0.916 | 0.913 | 0.915 | **0.916** | 0.913 |
| | 7 | 0.875 | 0.889 | 0.895 | 0.9 | 0.907 | 0.908 | 0.908 | 0.915 | 0.913 | 0.911 | 0.915 | 0.915 | 0.914 | 0.918 | **0.919** | 0.915 |
| | 9 | 0.863 | 0.884 | 0.893 | 0.896 | 0.9 | 0.905 | 0.908 | 0.911 | 0.91 | 0.913 | 0.909 | **0.918** | 0.917 | **0.918** | 0.915 | 0.915 |
| | 11 | 0.833 | 0.876 | 0.894 | 0.893 | 0.903 | 0.905 | 0.909 | 0.909 | 0.907 | 0.909 | 0.912 | 0.915 | 0.917 | 0.914 | **0.918** | 0.915 |
| Crowd Sourced imbalanced | 3 | 0.729 | 0.777 | 0.766 | 0.737 | 0.768 | 0.8 | 0.797 | 0.814 | 0.813 | 0.811 | 0.809 | 0.8 | 0.804 | **0.827** | 0.806 | 0.809 |
| | 5 | 0.77 | 0.754 | 0.759 | 0.777 | 0.806 | 0.822 | 0.814 | 0.829 | 0.831 | 0.835 | 0.823 | 0.821 | 0.838 | **0.84** | 0.824 | 0.838 |
| | 7 | 0.79 | 0.812 | 0.808 | 0.828 | 0.847 | 0.83 | 0.835 | 0.835 | 0.848 | 0.841 | 0.846 | 0.844 | 0.839 | 0.838 | 0.838 | **0.851** |
| | 9 | 0.799 | 0.812 | 0.833 | 0.832 | 0.829 | 0.842 | 0.836 | 0.839 | **0.851** | 0.842 | 0.842 | 0.847 | 0.824 | 0.83 | 0.837 | 0.841 |
| | 11 | 0.807 | 0.807 | 0.827 | 0.826 | **0.846** | 0.83 | 0.833 | 0.832 | 0.821 | 0.829 | 0.83 | 0.824 | 0.838 | 0.833 | 0.823 | 0.838 |
| Crowd Sourced balanced | 3 | 0.658 | 0.661 | 0.71 | 0.723 | 0.782 | 0.819 | 0.786 | 0.845 | 0.841 | 0.842 | 0.857 | 0.841 | 0.871 | 0.884 | 0.846 | **0.892** |
| | 5 | 0.677 | 0.682 | 0.723 | 0.78 | 0.827 | 0.848 | 0.849 | 0.867 | 0.889 | 0.888 | 0.899 | 0.864 | 0.89 | 0.864 | **0.914** | 0.906 |
| | 7 | 0.703 | 0.773 | 0.813 | 0.838 | 0.876 | 0.893 | 0.886 | 0.905 | 0.906 | 0.919 | 0.908 | 0.914 | 0.916 | 0.911 | 0.907 | **0.922** |
| | 9 | 0.778 | 0.815 | 0.849 | 0.869 | 0.895 | 0.899 | 0.905 | 0.909 | 0.906 | 0.903 | 0.91 | 0.908 | **0.916** | 0.913 | 0.898 | 0.915 |
| | 11 | 0.805 | 0.836 | 0.873 | 0.883 | 0.886 | 0.902 | 0.895 | 0.897 | 0.888 | 0.891 | 0.891 | 0.886 | 0.892 | **0.904** | 0.88 | 0.894 |

## 5 Results and comparisons

The results of the experiments are shown in the tables below. Comparison of experimental results are made in terms of:

- The quality of classification for different numbers of artificial objects created based on one original object in local table.

- The quality of classification for different approaches: average of weights and sum of weights to aggregating local neural networks.

**Table 3. Results of classification accuracy *acc* for the proposed approach: One hidden layer, the method to substitute values of missing attributes in local tables—Three artificial objects generated based on one original object, MLP networks aggregation using average of weights and various number of neurons in the hidden layer (3AO-1HL-AVG).** Designation I is used for the number of neurons in the input layer.

| Data set | No. tables | No. of neurons in hidden layer | | | | | | | | | | | | | | | |
|---|---|---|---|---|---|---|---|---|---|---|---|---|---|---|---|---|---|
| | | 0.25 × I | 0.5 × I | 0.75 × I | 1 × I | 1.5 × I | 1.75 × I | 2 × I | 2.5 × I | 2.75 × I | 3 × I | 3.5 × I | 3.75 × I | 4 × I | 4.5 × I | 4.75 × I | 5 × I |
| Vehicle imbalanced | 3 | 0.282 | 0.592 | 0.654 | 0.690 | 0.672 | 0.664 | 0.661 | 0.685 | 0.668 | 0.652 | 0.675 | 0.705 | 0.671 | 0.689 | **0.718** | 0.685 |
| | 5 | 0.479 | 0.588 | 0.639 | 0.672 | 0.664 | 0.644 | 0.68 | 0.69 | 0.668 | 0.675 | 0.664 | 0.654 | 0.688 | **0.693** | 0.664 | 0.665 |
| | 7 | 0.283 | 0.538 | 0.619 | 0.638 | 0.676 | 0.654 | 0.66 | 0.672 | 0.68 | **0.686** | 0.66 | 0.685 | 0.675 | 0.671 | 0.675 | 0.684 |
| | 9 | 0.283 | 0.661 | 0.65 | 0.647 | 0.681 | 0.69 | 0.686 | 0.682 | 0.68 | 0.668 | 0.65 | 0.692 | 0.675 | **0.694** | 0.684 | 0.684 |
| | 11 | 0.353 | 0.615 | 0.655 | 0.646 | 0.672 | 0.672 | 0.65 | 0.642 | 0.661 | 0.673 | 0.675 | **0.69** | 0.672 | 0.684 | 0.66 | 0.655 |
| Vehicle balanced | 3 | 0.407 | 0.281 | 0.681 | 0.66 | 0.685 | 0.686 | 0.675 | 0.676 | 0.722 | 0.688 | 0.714 | 0.701 | 0.689 | 0.699 | 0.718 | **0.722** |
| | 5 | 0.533 | 0.647 | 0.652 | 0.661 | 0.652 | 0.657 | 0.672 | 0.66 | 0.693 | 0.668 | 0.682 | 0.698 | **0.72** | 0.693 | 0.669 | 0.659 |
| | 7 | 0.283 | 0.68 | 0.668 | 0.661 | 0.675 | 0.707 | 0.698 | 0.706 | 0.709 | 0.731 | 0.711 | 0.714 | 0.723 | **0.732** | 0.72 | 0.728 |
| | 9 | 0.283 | 0.606 | 0.539 | 0.668 | 0.698 | 0.68 | 0.68 | 0.693 | 0.69 | 0.689 | 0.694 | **0.702** | 0.692 | 0.69 | 0.689 | 0.697 |
| | 11 | 0.283 | 0.568 | 0.648 | 0.681 | 0.659 | 0.673 | 0.671 | 0.664 | 0.677 | 0.689 | 0.689 | 0.685 | 0.681 | **0.692** | 0.682 | 0.686 |
| Dry Bean imbalanced | 3 | 0.87 | 0.889 | 0.89 | 0.898 | 0.901 | 0.896 | 0.901 | 0.906 | 0.906 | 0.907 | 0.906 | 0.907 | 0.909 | 0.907 | 0.907 | **0.909** |
| | 5 | 0.88 | 0.886 | 0.888 | 0.893 | 0.895 | 0.897 | 0.895 | 0.902 | 0.903 | 0.903 | 0.903 | 0.906 | 0.908 | 0.906 | **0.91** | 0.909 |
| | 7 | 0.879 | 0.887 | 0.891 | 0.89 | 0.896 | 0.895 | 0.897 | 0.898 | 0.901 | 0.901 | 0.905 | 0.904 | 0.906 | **0.907** | 0.904 | 0.906 |
| | 9 | 0.832 | 0.886 | 0.888 | 0.892 | 0.893 | 0.896 | 0.897 | 0.899 | 0.899 | 0.899 | 0.902 | 0.904 | 0.904 | 0.905 | **0.906** | 0.904 |
| | 11 | 0.839 | 0.886 | 0.888 | 0.889 | 0.891 | 0.896 | 0.897 | 0.896 | 0.9 | 0.899 | 0.9 | 0.901 | 0.902 | 0.903 | 0.904 | **0.907** |
| Dry Bean balanced | 3 | 0.887 | 0.891 | 0.889 | 0.897 | 0.902 | 0.902 | 0.904 | 0.908 | 0.908 | 0.907 | 0.909 | 0.91 | 0.909 | **0.911** | 0.91 | **0.911** |
| | 5 | 0.838 | 0.888 | 0.889 | 0.894 | 0.897 | 0.896 | 0.9 | 0.902 | 0.901 | 0.899 | 0.904 | 0.906 | 0.907 | 0.906 | 0.908 | **0.91** |
| | 7 | 0.881 | 0.882 | 0.89 | 0.892 | 0.895 | 0.895 | 0.898 | 0.897 | 0.9 | 0.9 | 0.902 | 0.903 | 0.903 | **0.907** | 0.905 | 0.904 |
| | 9 | 0.878 | 0.884 | 0.888 | 0.893 | 0.891 | 0.892 | 0.895 | 0.899 | 0.898 | 0.898 | 0.904 | 0.901 | 0.904 | 0.904 | 0.905 | **0.906** |
| | 11 | 0.877 | 0.886 | 0.889 | 0.893 | 0.895 | 0.894 | 0.894 | 0.896 | 0.899 | 0.899 | 0.9 | 0.9 | 0.902 | 0.902 | 0.902 | **0.905** |
| Sensorless | 3 | 0.873 | 0.895 | 0.903 | 0.902 | 0.912 | 0.909 | 0.918 | 0.916 | 0.908 | 0.919 | 0.918 | 0.92 | 0.912 | 0.917 | **0.92** | 0.918 |
| | 5 | 0.882 | 0.888 | 0.902 | 0.899 | 0.907 | 0.906 | 0.907 | 0.908 | 0.911 | 0.91 | 0.914 | 0.909 | 0.911 | 0.912 | **0.914** | 0.912 |
| | 7 | 0.787 | 0.882 | 0.89 | 0.902 | 0.903 | 0.906 | 0.912 | 0.908 | 0.911 | 0.915 | 0.915 | **0.919** | 0.916 | 0.917 | 0.916 | 0.916 |
| | 9 | 0.862 | 0.885 | 0.888 | 0.898 | 0.903 | 0.907 | 0.912 | 0.909 | 0.911 | 0.909 | 0.91 | 0.909 | 0.911 | **0.916** | 0.914 | 0.914 |
| | 11 | 0.802 | 0.879 | 0.89 | 0.898 | 0.907 | 0.907 | 0.908 | 0.912 | 0.91 | 0.913 | 0.912 | 0.915 | 0.916 | **0.922** | 0.918 | 0.917 |
| Crowd Sourced imbalanced | 3 | 0.741 | 0.74 | 0.747 | 0.757 | 0.754 | 0.776 | 0.783 | 0.8 | 0.788 | 0.788 | **0.806** | 0.791 | 0.788 | 0.799 | 0.796 | 0.801 |
| | 5 | 0.689 | 0.752 | 0.773 | 0.783 | 0.811 | 0.8 | 0.818 | 0.818 | 0.823 | 0.805 | 0.81 | 0.821 | 0.818 | 0.808 | 0.821 | **0.827** |
| | 7 | 0.745 | 0.818 | 0.77 | 0.795 | 0.813 | 0.813 | 0.808 | 0.833 | 0.824 | 0.818 | 0.831 | 0.825 | 0.83 | 0.823 | **0.837** | 0.833 |
| | 9 | 0.79 | 0.779 | 0.806 | 0.814 | 0.834 | 0.817 | 0.823 | 0.829 | 0.828 | 0.84 | 0.839 | 0.837 | 0.833 | **0.846** | 0.843 | 0.845 |
| | 11 | 0.82 | 0.825 | 0.818 | 0.838 | 0.842 | 0.84 | 0.859 | 0.855 | **0.871** | 0.863 | 0.84 | 0.857 | 0.852 | 0.854 | 0.857 | 0.857 |
| Crowd Sourced balanced | 3 | 0.252 | 0.712 | 0.755 | 0.752 | 0.784 | 0.75 | 0.777 | 0.815 | 0.776 | 0.802 | 0.832 | 0.819 | 0.793 | 0.829 | **0.833** | 0.821 |
| | 5 | 0.394 | 0.726 | 0.685 | 0.8 | 0.789 | 0.82 | 0.831 | 0.847 | 0.843 | 0.871 | 0.88 | **0.892** | 0.879 | 0.881 | 0.878 | 0.884 |
| | 7 | 0.546 | 0.778 | 0.799 | 0.806 | 0.844 | 0.879 | 0.878 | 0.887 | 0.883 | 0.893 | 0.904 | 0.911 | 0.895 | 0.903 | **0.916** | 0.916 |
| | 9 | 0.759 | 0.807 | 0.788 | 0.855 | 0.869 | 0.878 | 0.907 | 0.901 | 0.9 | 0.894 | **0.921** | 0.914 | 0.917 | 0.915 | 0.905 | 0.895 |
| | 11 | 0.789 | 0.834 | 0.83 | 0.866 | 0.892 | 0.891 | 0.897 | 0.898 | 0.904 | 0.9 | 0.911 | 0.912 | 0.904 | 0.913 | 0.907 | **0.914** |

- The quality of classification for different number of hidden layers.

- The quality of classification for different number of neurons in the hidden layers.

- The quality of classification of the proposed method versus two other approaches from the literature—homogeneous and heterogeneous ensemble of classifiers.

The average classification accuracy obtained from three runs of the algorithm are presented. Tables 2–5 show the results obtained for one hidden layer, different number of artificial objects

**Table 4. Results of classification accuracy *acc* for the proposed approach: One hidden layer, the method to substitute values of missing attributes in local tables— One artificial objects generated based on one original object, MLP networks aggregation using sum of weights and various number of neurons in the hidden layer (1AO-1HL-SUM).** Designation I is used for the number of neurons in the input layer.

| Data set | No. tables | No. of neurons in hidden layer | | | | | | | | | | | | | | | |
|---|---|---|---|---|---|---|---|---|---|---|---|---|---|---|---|---|---|
| | | 0.25 × I | 0.5 × I | 0.75 × I | 1 × I | 1.5 × I | 1.75 × I | 2 × I | 2.5 × I | 2.75 × I | 3 × I | 3.5 × I | 3.75 × I | 4 × I | 4.5 × I | 4.75 × I | 5 × I |
| Vehicle imbalanced | 3 | 0.264 | 0.571 | 0.588 | 0.598 | 0.625 | 0.631 | 0.609 | 0.609 | 0.647 | **0.675** | 0.608 | 0.644 | 0.626 | 0.648 | 0.631 | 0.627 |
| | 5 | 0.4 | 0.626 | 0.638 | 0.625 | 0.602 | 0.664 | 0.598 | 0.66 | 0.636 | 0.671 | 0.646 | 0.665 | 0.659 | 0.661 | 0.661 | **0.673** |
| | 7 | 0.272 | 0.643 | 0.594 | 0.636 | 0.671 | 0.648 | 0.667 | 0.626 | **0.682** | 0.647 | 0.639 | 0.673 | 0.651 | 0.667 | 0.647 | 0.664 |
| | 9 | 0.283 | 0.496 | 0.633 | 0.539 | 0.646 | 0.659 | 0.643 | 0.655 | 0.654 | 0.657 | 0.669 | 0.667 | **0.697** | 0.644 | 0.685 | 0.686 |
| | 11 | 0.283 | 0.65 | 0.65 | 0.634 | 0.633 | 0.648 | 0.635 | 0.631 | 0.682 | 0.66 | 0.684 | 0.663 | 0.646 | 0.671 | 0.678 | **0.694** |
| Vehicle imbalanced | 3 | 0.461 | 0.423 | 0.568 | 0.64 | 0.639 | 0.635 | 0.661 | 0.692 | 0.656 | **0.703** | 0.66 | 0.685 | 0.647 | 0.686 | 0.643 | 0.68 |
| | 5 | 0.283 | 0.301 | 0.661 | 0.627 | 0.671 | 0.622 | 0.654 | 0.69 | 0.709 | **0.71** | 0.678 | 0.656 | 0.682 | 0.69 | 0.684 | 0.668 |
| | 7 | 0.283 | 0.665 | 0.693 | 0.655 | 0.669 | 0.685 | 0.707 | 0.711 | 0.693 | 0.682 | 0.715 | 0.702 | **0.723** | 0.701 | 0.677 | 0.714 |
| | 9 | 0.286 | 0.373 | 0.665 | 0.642 | 0.681 | 0.685 | 0.643 | 0.702 | 0.681 | 0.668 | 0.678 | 0.707 | 0.702 | 0.715 | 0.738 | **0.743** |
| | 11 | 0.283 | 0.463 | 0.677 | 0.615 | 0.63 | 0.675 | 0.675 | **0.714** | 0.668 | 0.689 | 0.706 | 0.707 | 0.709 | 0.709 | 0.693 | 0.707 |
| Dry Bean imbalanced | 3 | 0.826 | 0.891 | 0.887 | 0.902 | 0.901 | 0.905 | 0.904 | 0.909 | 0.907 | 0.91 | 0.91 | 0.911 | 0.911 | **0.912** | **0.912** | **0.912** |
| | 5 | 0.878 | 0.883 | 0.896 | 0.904 | 0.901 | 0.908 | 0.908 | 0.907 | 0.906 | 0.909 | 0.912 | 0.909 | **0.912** | 0.911 | 0.91 | 0.911 |
| | 7 | 0.858 | 0.88 | 0.889 | 0.9 | 0.903 | 0.908 | 0.907 | 0.906 | 0.908 | 0.908 | 0.908 | 0.91 | 0.911 | 0.911 | 0.913 | **0.914** |
| | 9 | 0.898 | 0.892 | 0.899 | 0.902 | 0.905 | 0.904 | 0.907 | 0.91 | 0.909 | 0.909 | 0.911 | 0.909 | 0.913 | 0.911 | **0.914** | 0.913 |
| | 11 | 0.877 | 0.892 | 0.897 | 0.901 | 0.904 | 0.906 | 0.907 | 0.908 | 0.908 | 0.91 | 0.911 | 0.907 | 0.91 | 0.909 | **0.913** | 0.91 |
| Dry Bean balanced | 3 | 0.809 | 0.901 | 0.9 | 0.905 | 0.91 | 0.898 | 0.907 | 0.907 | 0.9 | 0.909 | 0.91 | 0.91 | 0.909 | **0.912** | **0.912** | 0.91 |
| | 5 | 0.739 | 0.89 | 0.896 | 0.899 | 0.907 | 0.905 | 0.905 | 0.906 | 0.912 | 0.907 | 0.908 | 0.91 | 0.912 | 0.91 | **0.916** | 0.913 |
| | 7 | 0.867 | 0.892 | 0.896 | 0.9 | 0.904 | 0.9 | 0.906 | 0.907 | 0.906 | 0.91 | 0.911 | 0.908 | 0.912 | 0.911 | **0.912** | **0.912** |
| | 9 | 0.876 | 0.893 | 0.894 | 0.899 | 0.905 | 0.901 | 0.907 | 0.908 | 0.91 | 0.909 | 0.91 | 0.911 | 0.91 | **0.913** | 0.91 | 0.911 |
| | 11 | 0.773 | 0.881 | 0.895 | 0.898 | 0.902 | 0.904 | 0.906 | 0.909 | 0.904 | 0.908 | 0.908 | 0.911 | **0.912** | 0.91 | 0.91 | 0.911 |
| Sensorless | 3 | 0.864 | 0.887 | 0.895 | 0.901 | 0.904 | 0.896 | 0.899 | 0.903 | 0.909 | 0.906 | 0.908 | 0.901 | 0.916 | 0.911 | 0.918 | **0.92** |
| | 5 | 0.869 | 0.887 | 0.896 | 0.899 | 0.902 | 0.901 | 0.904 | 0.902 | 0.904 | 0.904 | 0.907 | 0.902 | **0.911** | 0.908 | 0.908 | 0.904 |
| | 7 | 0.869 | 0.886 | 0.895 | 0.895 | 0.903 | 0.905 | 0.902 | 0.908 | 0.906 | 0.91 | 0.899 | 0.911 | 0.909 | **0.915** | **0.915** | 0.914 |
| | 9 | 0.868 | 0.895 | 0.893 | 0.901 | 0.902 | 0.902 | 0.905 | 0.906 | 0.907 | 0.905 | 0.909 | 0.91 | 0.915 | **0.917** | 0.914 | 0.912 |
| | 11 | 0.846 | 0.893 | 0.894 | 0.898 | 0.903 | 0.906 | 0.906 | 0.908 | 0.909 | 0.909 | 0.912 | 0.909 | 0.912 | 0.911 | **0.914** | **0.914** |
| Crowd Sourced imbalanced | 3 | 0.612 | 0.756 | 0.753 | 0.753 | 0.757 | 0.777 | 0.773 | 0.782 | 0.783 | 0.793 | 0.792 | 0.777 | 0.79 | 0.793 | 0.775 | **0.796** |
| | 5 | 0.757 | 0.705 | 0.766 | 0.764 | 0.783 | 0.788 | 0.789 | 0.795 | 0.805 | 0.798 | 0.8 | 0.792 | 0.794 | 0.807 | **0.817** | 0.814 |
| | 7 | 0.747 | 0.733 | 0.77 | 0.766 | 0.763 | 0.772 | 0.782 | 0.798 | 0.808 | 0.778 | 0.814 | 0.81 | 0.811 | **0.829** | 0.803 | 0.82 |
| | 9 | 0.718 | 0.758 | 0.765 | 0.747 | 0.774 | 0.763 | 0.786 | 0.768 | 0.768 | 0.792 | 0.807 | 0.81 | 0.802 | 0.804 | 0.815 | **0.832** |
| | 11 | 0.734 | 0.758 | 0.752 | 0.751 | 0.764 | 0.785 | 0.787 | 0.806 | 0.817 | 0.811 | 0.815 | 0.779 | 0.801 | 0.803 | 0.823 | **0.827** |
| Crowd Sourced balanced | 3 | 0.688 | 0.699 | 0.695 | 0.741 | 0.753 | 0.794 | 0.798 | 0.786 | 0.817 | 0.802 | 0.82 | 0.814 | 0.832 | 0.841 | 0.849 | **0.864** |
| | 5 | 0.684 | 0.7 | 0.649 | 0.652 | 0.759 | 0.75 | 0.764 | 0.761 | 0.767 | 0.827 | 0.795 | 0.823 | 0.819 | 0.866 | **0.874** | 0.851 |
| | 7 | 0.65 | 0.716 | 0.72 | 0.681 | 0.763 | 0.769 | 0.746 | 0.758 | 0.776 | 0.8 | 0.831 | 0.799 | 0.83 | **0.864** | 0.862 | 0.839 |
| | 9 | 0.569 | 0.644 | 0.697 | 0.668 | 0.743 | 0.714 | 0.795 | 0.782 | 0.789 | 0.808 | 0.789 | 0.815 | 0.794 | **0.86** | 0.84 | 0.812 |
| | 11 | 0.62 | 0.711 | 0.663 | 0.657 | 0.726 | 0.77 | 0.797 | 0.796 | 0.812 | 0.833 | 0.786 | 0.789 | 0.837 | 0.845 | 0.844 | **0.861** |

(one or three artificial objects generated), different aggregation methods used (average or sum) and different number of neurons in the hidden layer. For simplicity, the designations are adopted:

- 1HL, 2HL—for one or two hidden layers,

- 1AO, 3AO—for one or three generated artificial objects,

- AVG, SUM—for the aggregation method—average and sum.

**Table 5. Results of classification accuracy *acc* for the proposed approach: One hidden layer, the method to substitute values of missing attributes in local tables—Three artificial objects generated based on one original object, MLP networks aggregation using sum of weights and various number of neurons in the hidden layer (3AO-1HL-SUM).** Designation I is used for the number of neurons in the input layer.

| Data set | No. tables | No. of neurons in hidden layer | | | | | | | | | | | | | | | |
|---|---|---|---|---|---|---|---|---|---|---|---|---|---|---|---|---|---|
| | | 0.25 × I | 0.5 × I | 0.75 × I | 1 × I | 1.5 × I | 1.75 × I | 2 × I | 2.5 × I | 2.75 × I | 3 × I | 3.5 × I | 3.75 × I | 4 × I | 4.5 × I | 4.75 × I | 5 × I |
| Vehicle imbalanced | 3 | 0.273 | 0.577 | 0.598 | 0.602 | 0.639 | 0.644 | 0.643 | 0.633 | 0.651 | 0.640 | 0.657 | 0.636 | **0.665** | 0.638 | 0.631 | 0.643 |
| | 5 | 0.304 | 0.482 | 0.591 | 0.644 | 0.643 | 0.639 | 0.648 | 0.661 | 0.663 | 0.592 | 0.639 | 0.634 | 0.623 | 0.651 | **0.671** | 0.635 |
| | 7 | 0.264 | 0.558 | 0.585 | 0.627 | 0.627 | 0.626 | 0.644 | 0.636 | 0.654 | 0.644 | 0.643 | 0.685 | 0.642 | **0.696** | 0.667 | 0.657 |
| | 9 | 0.283 | 0.486 | 0.636 | 0.614 | 0.677 | 0.663 | 0.682 | 0.646 | 0.701 | 0.661 | **0.717** | 0.702 | 0.701 | 0.682 | 0.705 | 0.675 |
| | 11 | 0.341 | 0.461 | 0.647 | 0.648 | 0.643 | 0.655 | 0.654 | 0.678 | 0.681 | 0.66 | 0.661 | 0.669 | **0.688** | 0.665 | 0.671 | 0.657 |
| Vehicle balanced | 3 | 0.386 | 0.281 | 0.584 | 0.543 | 0.657 | 0.6 | 0.627 | 0.618 | 0.663 | 0.648 | 0.665 | 0.661 | 0.623 | 0.644 | **0.677** | 0.633 |
| | 5 | 0.353 | 0.53 | 0.634 | 0.629 | 0.629 | 0.664 | **0.717** | 0.643 | 0.672 | 0.671 | 0.64 | 0.688 | 0.661 | 0.667 | 0.646 | 0.626 |
| | 7 | 0.264 | 0.659 | 0.618 | 0.609 | 0.675 | 0.718 | 0.699 | 0.71 | 0.702 | 0.709 | 0.707 | **0.735** | 0.698 | 0.701 | 0.717 | 0.72 |
| | 9 | 0.283 | 0.535 | 0.529 | 0.693 | 0.661 | 0.672 | 0.681 | 0.672 | **0.711** | 0.689 | 0.702 | 0.684 | 0.705 | 0.698 | 0.692 | 0.701 |
| | 11 | 0.283 | 0.555 | 0.613 | 0.638 | 0.652 | 0.66 | 0.68 | 0.692 | 0.639 | 0.693 | 0.681 | 0.663 | **0.705** | 0.701 | 0.686 | 0.696 |
| Dry Bean imbalanced | 3 | 0.796 | 0.895 | 0.901 | 0.901 | 0.904 | 0.904 | 0.904 | 0.909 | 0.906 | 0.907 | 0.909 | **0.911** | 0.91 | 0.909 | **0.911** | **0.911** |
| | 5 | 0.859 | 0.888 | 0.897 | 0.9 | 0.905 | 0.902 | 0.905 | 0.909 | 0.909 | 0.908 | 0.909 | 0.91 | 0.91 | 0.909 | **0.913** | **0.913** |
| | 7 | 0.878 | 0.889 | 0.899 | 0.891 | 0.906 | 0.907 | 0.904 | 0.91 | 0.91 | 0.91 | 0.909 | 0.911 | 0.91 | **0.913** | 0.911 | 0.912 |
| | 9 | 0.828 | 0.885 | 0.891 | 0.898 | 0.901 | 0.903 | 0.906 | 0.906 | 0.906 | 0.909 | 0.91 | 0.909 | 0.91 | **0.913** | 0.912 | 0.911 |
| | 11 | 0.869 | 0.887 | 0.891 | 0.899 | 0.902 | 0.906 | 0.9 | 0.907 | 0.904 | 0.906 | 0.907 | 0.908 | **0.911** | 0.909 | 0.909 | 0.91 |
| Dry Bean balanced | 3 | 0.845 | 0.884 | 0.887 | 0.895 | 0.905 | 0.906 | 0.909 | 0.91 | 0.908 | 0.909 | 0.907 | 0.91 | **0.911** | **0.911** | 0.91 | **0.911** |
| | 5 | 0.815 | 0.887 | 0.891 | 0.903 | 0.905 | 0.907 | 0.906 | 0.909 | 0.91 | 0.909 | 0.909 | 0.913 | 0.911 | 0.911 | 0.911 | **0.913** |
| | 7 | 0.864 | 0.888 | 0.894 | 0.901 | 0.901 | 0.906 | 0.908 | 0.903 | 0.908 | 0.906 | 0.911 | 0.911 | 0.909 | 0.911 | **0.913** | 0.909 |
| | 9 | 0.863 | 0.884 | 0.894 | 0.896 | 0.903 | 0.903 | 0.907 | 0.907 | 0.907 | 0.908 | 0.91 | 0.908 | **0.912** | 0.91 | 0.909 | **0.912** |
| | 11 | 0.886 | 0.885 | 0.89 | 0.892 | 0.9 | 0.899 | 0.903 | 0.905 | 0.906 | 0.909 | 0.907 | 0.911 | 0.91 | **0.912** | 0.911 | **0.912** |
| Sensorless | 3 | 0.795 | 0.862 | 0.9 | 0.887 | 0.905 | 0.909 | 0.911 | 0.916 | **0.918** | 0.914 | 0.915 | **0.918** | 0.911 | 0.914 | 0.917 | 0.917 |
| | 5 | 0.748 | 0.885 | 0.893 | 0.893 | 0.899 | 0.895 | 0.895 | 0.903 | 0.903 | 0.896 | 0.903 | 0.902 | 0.901 | 0.9 | **0.904** | **0.904** |
| | 7 | 0.785 | 0.881 | 0.868 | 0.886 | 0.895 | 0.896 | 0.9 | 0.902 | 0.898 | 0.904 | 0.904 | 0.905 | **0.908** | 0.903 | 0.906 | 0.902 |
| | 9 | 0.824 | 0.869 | 0.882 | 0.89 | 0.9 | 0.898 | 0.902 | 0.9 | 0.902 | 0.903 | 0.905 | 0.906 | 0.905 | **0.908** | 0.905 | 0.907 |
| | 11 | 0.794 | 0.854 | 0.858 | 0.888 | 0.896 | 0.892 | 0.9 | 0.9 | 0.903 | 0.905 | 0.9 | 0.907 | 0.907 | 0.906 | 0.908 | **0.91** |
| Crowd Sourced imbalanced | 3 | 0.741 | 0.74 | 0.747 | 0.757 | 0.754 | 0.776 | 0.783 | 0.8 | 0.788 | 0.788 | **0.806** | 0.791 | 0.788 | 0.799 | 0.796 | 0.801 |
| | 5 | 0.679 | 0.737 | 0.732 | 0.761 | 0.776 | 0.752 | 0.795 | 0.783 | 0.786 | 0.782 | 0.789 | 0.794 | 0.793 | 0.785 | 0.795 | **0.802** |
| | 7 | 0.74 | 0.753 | 0.738 | 0.756 | 0.748 | 0.773 | 0.777 | 0.795 | 0.789 | 0.77 | 0.794 | 0.798 | 0.812 | 0.814 | **0.818** | 0.802 |
| | 9 | 0.731 | 0.725 | 0.745 | 0.771 | 0.736 | 0.775 | 0.776 | 0.779 | 0.786 | 0.794 | 0.797 | 0.794 | **0.808** | 0.798 | 0.798 | 0.806 |
| | 11 | 0.716 | 0.77 | 0.759 | 0.759 | 0.781 | 0.793 | 0.767 | 0.799 | 0.784 | 0.798 | 0.804 | 0.805 | 0.82 | 0.826 | 0.818 | **0.842** |
| Crowd Sourced balanced | 3 | 0.26 | 0.648 | 0.714 | 0.699 | 0.741 | 0.722 | 0.731 | 0.757 | 0.742 | 0.74 | **0.782** | 0.776 | 0.756 | 0.768 | 0.779 | 0.776 |
| | 5 | 0.568 | 0.712 | 0.724 | 0.762 | 0.751 | 0.793 | 0.777 | 0.798 | 0.779 | 0.824 | 0.826 | 0.832 | 0.818 | 0.839 | 0.822 | **0.839** |
| | 7 | 0.646 | 0.724 | 0.741 | 0.748 | 0.759 | 0.793 | 0.818 | 0.803 | 0.81 | 0.841 | 0.827 | 0.843 | 0.856 | 0.851 | **0.866** | 0.857 |
| | 9 | 0.692 | 0.697 | 0.728 | 0.767 | 0.766 | 0.775 | 0.799 | 0.809 | 0.78 | 0.819 | 0.812 | 0.84 | **0.855** | 0.844 | 0.853 | 0.853 |
| | 11 | 0.672 | 0.699 | 0.722 | 0.729 | 0.747 | 0.767 | 0.768 | 0.797 | 0.817 | 0.828 | 0.85 | 0.834 | 0.856 | 0.838 | **0.863** | 0.821 |

Tables 6–21, show the results obtained for two hidden layers and also different number of artificial objects (one or three artificial objects generated), different aggregation methods used (mean and sum) and different number of neurons in the hidden layers. The columns show the number of neurons in the first hidden layer while the rows indicate the different number of neurons in the second hidden layer. The results obtained for different data sets are divided into separate tables due to the size of these tables. In all of these tables, the best result obtained for a given number of hidden layers, a number artificial objects and given aggregation method

**Table 6. Results of classification accuracy *acc* for the proposed approach and the Vehicle data sets: Two hidden layers, the method to substitute values of missing attributes in local tables—One artificial objects generated based on one original object, MLP networks aggregation using average of weights and various number of neurons in the hidden layer (1AO-2HL-AVG).** Designation I is used for the number of neurons in the input layer.

| Data set | No. tables | Second HL | No. of neurons in the first hidden layer | | | | | | | | | | | | | | | |
|---|---|---|---|---|---|---|---|---|---|---|---|---|---|---|---|---|---|---|
| | | | 0.25 × I | 0.5 × I | 0.75 × I | 1 × I | 1.5 × I | 1.75 × I | 2 × I | 2.5 × I | 2.75 × I | 3 × I | 3.5 × I | 3.75 × I | 4 × I | 4.5 × I | 4.75 × I | 5 × I |
| Vehicle imbalanced | 3 | 1 × I | 0.504 | 0.635 | 0.651 | 0.64 | 0.668 | 0.686 | 0.678 | 0.673 | 0.671 | 0.677 | 0.675 | 0.665 | 0.681 | 0.693 | 0.598 | 0.615 |
| | | 2 × I | 0.612 | 0.667 | 0.676 | 0.676 | 0.69 | 0.681 | 0.706 | 0.684 | 0.696 | 0.698 | 0.699 | 0.694 | 0.727 | 0.682 | 0.69 | 0.682 |
| | | 3 × I | 0.625 | 0.692 | 0.686 | 0.694 | 0.701 | 0.692 | 0.702 | 0.672 | 0.71 | 0.711 | 0.693 | 0.696 | 0.71 | 0.688 | 0.718 | 0.669 |
| | | 4 × I | 0.478 | 0.702 | 0.686 | 0.68 | 0.703 | 0.697 | 0.689 | **0.739** | 0.692 | 0.692 | 0.698 | 0.703 | 0.693 | 0.703 | 0.689 | 0.696 |
| | | 5 × I | 0.283 | 0.68 | 0.664 | 0.675 | 0.675 | 0.684 | 0.707 | 0.714 | 0.684 | 0.713 | 0.726 | 0.706 | 0.714 | 0.667 | 0.734 | 0.711 |
| | 5 | 1 × I | 0.625 | 0.636 | 0.639 | 0.657 | 0.621 | 0.657 | 0.663 | 0.677 | 0.646 | 0.673 | 0.671 | 0.636 | 0.664 | 0.657 | 0.669 | 0.671 |
| | | 2 × I | 0.409 | 0.663 | 0.643 | 0.65 | 0.684 | 0.678 | 0.692 | 0.696 | 0.656 | 0.68 | 0.692 | 0.668 | 0.672 | 0.659 | 0.65 | 0.664 |
| | | 3 × I | 0.634 | 0.694 | 0.648 | 0.68 | 0.642 | 0.681 | 0.643 | 0.669 | 0.673 | 0.692 | 0.684 | 0.669 | 0.678 | 0.689 | 0.664 | 0.661 |
| | | 4 × I | 0.584 | 0.664 | 0.684 | 0.677 | 0.671 | 0.688 | **0.709** | 0.705 | 0.665 | 0.693 | 0.671 | 0.671 | 0.688 | 0.684 | 0.677 | 0.692 |
| | | 5 × I | 0.667 | 0.665 | 0.671 | 0.663 | 0.69 | 0.678 | 0.692 | 0.654 | 0.665 | 0.709 | 0.665 | 0.68 | 0.682 | 0.69 | 0.644 | 0.694 |
| | 7 | 1 × I | 0.543 | 0.648 | 0.651 | 0.646 | 0.639 | 0.664 | 0.655 | 0.657 | 0.652 | 0.664 | 0.634 | 0.678 | 0.675 | 0.655 | 0.656 | 0.661 |
| | | 2 × I | 0.588 | 0.647 | 0.66 | 0.639 | 0.677 | 0.672 | 0.673 | 0.677 | 0.657 | 0.669 | 0.647 | 0.699 | 0.664 | 0.644 | 0.686 | 0.655 |
| | | 3 × I | 0.601 | 0.655 | 0.664 | 0.651 | 0.669 | 0.669 | 0.693 | 0.688 | 0.675 | 0.703 | 0.671 | 0.68 | 0.676 | 0.688 | 0.685 | **0.724** |
| | | 4 × I | 0.283 | 0.664 | 0.685 | 0.68 | 0.702 | 0.693 | 0.663 | 0.693 | 0.705 | 0.696 | 0.69 | 0.711 | 0.701 | 0.694 | 0.685 | 0.686 |
| | | 5 × I | 0.554 | 0.665 | 0.693 | 0.66 | 0.671 | 0.684 | 0.685 | 0.682 | 0.669 | 0.68 | 0.705 | 0.686 | 0.718 | 0.69 | 0.698 | 0.697 |
| | 9 | 1 × I | 0.583 | 0.633 | 0.651 | 0.663 | 0.68 | 0.669 | 0.647 | 0.623 | 0.661 | 0.668 | 0.671 | 0.657 | 0.652 | 0.681 | 0.65 | 0.675 |
| | | 2 × I | 0.587 | 0.63 | 0.642 | 0.663 | 0.684 | 0.675 | 0.668 | 0.664 | 0.688 | 0.659 | 0.68 | 0.685 | 0.675 | 0.664 | 0.696 | 0.668 |
| | | 3 × I | 0.333 | 0.646 | 0.668 | 0.665 | 0.673 | 0.688 | 0.648 | 0.698 | 0.682 | 0.686 | 0.675 | 0.684 | 0.669 | **0.713** | 0.675 | 0.688 |
| | | 4 × I | 0.564 | 0.677 | 0.66 | 0.639 | 0.678 | 0.686 | 0.685 | 0.693 | 0.681 | 0.685 | 0.681 | 0.701 | 0.698 | 0.682 | 0.685 | 0.678 |
| | | 5 × I | 0.63 | 0.652 | 0.663 | 0.684 | 0.693 | 0.684 | 0.665 | 0.676 | 0.705 | 0.681 | 0.689 | 0.69 | 0.692 | 0.696 | 0.696 | 0.675 |
| | 11 | 1 × I | 0.526 | 0.622 | 0.562 | 0.621 | 0.627 | 0.634 | 0.613 | 0.633 | 0.655 | 0.648 | 0.647 | 0.664 | 0.615 | 0.629 | 0.655 | 0.634 |
| | | 2 × I | 0.283 | 0.631 | 0.622 | 0.635 | 0.619 | 0.642 | 0.642 | 0.657 | 0.675 | 0.682 | 0.676 | 0.668 | 0.659 | 0.656 | 0.681 | 0.65 |
| | | 3 × I | 0.6 | 0.562 | 0.583 | 0.659 | 0.686 | 0.678 | 0.644 | 0.657 | 0.648 | 0.664 | 0.651 | 0.696 | 0.682 | 0.664 | 0.685 | 0.644 |
| | | 4 × I | 0.581 | 0.657 | 0.654 | 0.696 | 0.678 | 0.673 | 0.656 | 0.664 | 0.684 | 0.664 | 0.685 | 0.677 | 0.672 | 0.692 | 0.656 | 0.664 |
| | | 5 × I | 0.584 | 0.65 | 0.675 | 0.605 | 0.671 | 0.68 | 0.664 | 0.678 | 0.66 | 0.678 | 0.673 | 0.669 | **0.706** | 0.697 | 0.678 | 0.673 |
| Vehicle balanced | 3 | 1 × I | 0.659 | 0.675 | 0.678 | 0.663 | 0.699 | 0.642 | 0.705 | 0.675 | 0.684 | 0.703 | 0.682 | 0.651 | 0.661 | 0.688 | 0.686 | 0.711 |
| | | 2 × I | 0.648 | 0.635 | 0.709 | 0.707 | 0.682 | 0.709 | 0.713 | 0.698 | 0.73 | 0.706 | 0.718 | 0.693 | 0.713 | 0.707 | 0.707 | 0.693 |
| | | 3 × I | 0.63 | 0.696 | 0.707 | 0.718 | 0.722 | 0.724 | 0.731 | **0.736** | 0.714 | 0.711 | 0.707 | 0.702 | 0.714 | 0.717 | 0.731 | 0.724 |
| | | 4 × I | 0.609 | 0.58 | 0.719 | 0.732 | 0.711 | 0.694 | 0.717 | 0.726 | 0.709 | 0.711 | 0.709 | 0.724 | 0.699 | 0.718 | 0.722 | 0.703 |
| | | 5 × I | 0.563 | 0.705 | 0.697 | 0.718 | 0.71 | 0.719 | 0.703 | 0.714 | 0.706 | 0.723 | 0.718 | 0.723 | 0.73 | 0.73 | 0.711 | 0.713 |
| | 5 | 1 × I | 0.66 | 0.673 | 0.664 | 0.657 | 0.684 | 0.697 | 0.702 | 0.688 | 0.678 | 0.669 | 0.664 | 0.706 | 0.707 | 0.702 | 0.715 | 0.696 |
| | | 2 × I | 0.606 | 0.681 | 0.696 | 0.657 | 0.709 | 0.713 | 0.701 | 0.693 | 0.71 | 0.692 | 0.705 | 0.705 | 0.714 | 0.677 | 0.71 | 0.698 |
| | | 3 × I | 0.577 | 0.668 | 0.713 | 0.696 | 0.724 | 0.703 | 0.693 | 0.688 | 0.686 | 0.685 | 0.72 | 0.706 | 0.707 | 0.702 | 0.696 | 0.699 |
| | | 4 × I | 0.613 | 0.723 | 0.696 | 0.696 | 0.681 | 0.702 | 0.711 | **0.728** | 0.706 | 0.697 | 0.688 | 0.707 | 0.717 | 0.711 | 0.684 | 0.72 |
| | | 5 × I | 0.283 | 0.722 | 0.706 | 0.685 | 0.711 | 0.714 | 0.69 | 0.703 | 0.719 | 0.707 | 0.681 | 0.723 | 0.706 | 0.713 | 0.694 | 0.703 |
| | 7 | 1 × I | 0.48 | 0.644 | 0.705 | 0.646 | 0.703 | 0.676 | 0.694 | 0.692 | 0.714 | 0.698 | 0.719 | 0.699 | 0.699 | 0.726 | 0.694 | 0.699 |
| | | 2 × I | 0.518 | 0.718 | 0.717 | 0.706 | 0.697 | 0.722 | 0.717 | 0.684 | 0.705 | 0.705 | 0.717 | 0.713 | 0.711 | 0.682 | 0.684 | 0.699 |
| | | 3 × I | 0.374 | 0.702 | 0.701 | 0.715 | 0.703 | 0.702 | 0.709 | 0.684 | 0.728 | 0.723 | 0.715 | 0.719 | 0.714 | 0.717 | 0.72 | 0.713 |
| | | 4 × I | 0.705 | 0.705 | 0.71 | 0.686 | 0.718 | 0.741 | 0.727 | 0.719 | 0.714 | 0.728 | 0.726 | 0.743 | 0.711 | 0.705 | 0.73 | 0.714 |
| | | 5 × I | 0.681 | 0.706 | 0.703 | 0.713 | 0.715 | 0.713 | 0.711 | 0.72 | 0.739 | 0.72 | 0.73 | 0.696 | 0.748 | **0.757** | 0.715 | 0.713 |
| | 9 | 1 × I | 0.61 | 0.626 | 0.669 | 0.646 | 0.676 | 0.659 | 0.681 | 0.667 | 0.643 | 0.698 | 0.675 | 0.682 | 0.652 | 0.688 | 0.677 | 0.692 |
| | | 2 × I | 0.623 | 0.625 | 0.675 | 0.655 | 0.678 | 0.684 | 0.677 | 0.673 | 0.685 | 0.681 | 0.709 | 0.693 | 0.672 | 0.681 | 0.688 | 0.696 |
| | | 3 × I | 0.677 | 0.66 | 0.672 | 0.698 | 0.709 | 0.69 | 0.667 | 0.71 | 0.68 | 0.707 | 0.707 | 0.684 | 0.678 | 0.689 | 0.714 | **0.719** |
| | | 4 × I | 0.589 | 0.646 | 0.699 | 0.684 | 0.68 | 0.678 | 0.664 | 0.678 | 0.673 | 0.689 | 0.682 | 0.681 | 0.705 | 0.681 | 0.692 | 0.682 |
| | | 5 × I | 0.675 | 0.554 | 0.705 | 0.68 | 0.681 | 0.703 | 0.707 | 0.69 | 0.678 | 0.696 | 0.711 | 0.697 | 0.709 | 0.709 | 0.697 | 0.699 |
| | 11 | 1 × I | 0.584 | 0.593 | 0.579 | 0.659 | 0.655 | 0.587 | 0.648 | 0.644 | 0.685 | 0.663 | 0.685 | 0.646 | 0.665 | 0.678 | 0.688 | 0.655 |
| | | 2 × I | 0.559 | 0.617 | 0.678 | 0.654 | 0.678 | 0.675 | 0.673 | 0.694 | 0.669 | 0.685 | 0.69 | 0.677 | 0.697 | 0.685 | 0.698 | 0.709 |
| | | 3 × I | 0.526 | 0.634 | 0.672 | 0.669 | 0.668 | 0.692 | 0.698 | 0.688 | 0.697 | 0.686 | 0.682 | 0.702 | 0.702 | 0.702 | 0.692 | 0.685 |
| | | 4 × I | 0.659 | 0.593 | 0.66 | 0.672 | 0.685 | 0.681 | 0.692 | 0.703 | 0.714 | 0.714 | 0.711 | 0.709 | 0.706 | 0.703 | 0.719 | 0.697 |
| | | 5 × I | 0.568 | 0.581 | 0.654 | 0.688 | 0.696 | 0.699 | 0.715 | 0.71 | **0.723** | 0.717 | 0.702 | 0.682 | 0.707 | 0.682 | 0.718 | 0.709 |

**Table 7. Results of classification accuracy *acc* for the proposed approach and the Dry Bean data sets: Two hidden layer, the method to substitute values of missing attributes in local tables—One artificial objects generated based on one original object, MLP networks aggregation using average of weights and various number of neurons in the hidden layer (1AO-2HL-AVG).** Designation I is used for the number of neurons in the input layer.

| Data set | No. tables | Second HL | No. of neurons in the first hidden layer | | | | | | | | | | | | | | | |
|---|---|---|---|---|---|---|---|---|---|---|---|---|---|---|---|---|---|---|
| | | | 0.25 × I | 0.5 × I | 0.75 × I | 1 × I | 1.5 × I | 1.75 × I | 2 × I | 2.5 × I | 2.75 × I | 3 × I | 3.5 × I | 3.75 × I | 4 × I | 4.5 × I | 4.75 × I | 5 × I |
| Dry Bean imbalanced | 3 | 1 × I | 0.888 | 0.891 | 0.892 | 0.892 | 0.893 | 0.891 | 0.892 | 0.894 | 0.894 | 0.892 | 0.896 | 0.894 | 0.895 | 0.895 | 0.898 | 0.9 |
| | | 2 × I | 0.891 | 0.89 | 0.892 | 0.893 | 0.894 | 0.898 | 0.894 | 0.895 | 0.898 | 0.896 | 0.898 | 0.899 | 0.9 | 0.899 | 0.899 | 0.896 |
| | | 3 × I | 0.89 | 0.894 | 0.893 | 0.894 | 0.896 | 0.896 | 0.896 | 0.898 | 0.899 | 0.896 | 0.899 | 0.902 | 0.899 | 0.899 | 0.901 | 0.903 |
| | | 4 × I | 0.89 | 0.894 | 0.897 | 0.894 | 0.897 | 0.895 | 0.898 | 0.896 | 0.897 | 0.901 | 0.903 | 0.9 | 0.899 | 0.899 | 0.901 | **0.906** |
| | | 5 × I | 0.896 | 0.894 | 0.896 | 0.897 | 0.896 | 0.895 | 0.904 | 0.901 | 0.901 | 0.9 | 0.905 | 0.899 | 0.903 | 0.902 | 0.903 | 0.903 |
| | 5 | 1 × I | 0.855 | 0.888 | 0.89 | 0.889 | 0.89 | 0.894 | 0.892 | 0.896 | 0.893 | 0.894 | 0.894 | 0.897 | 0.894 | 0.894 | 0.895 | 0.896 |
| | | 2 × I | 0.82 | 0.887 | 0.89 | 0.892 | 0.893 | 0.893 | 0.893 | 0.895 | 0.895 | 0.896 | 0.894 | 0.897 | 0.896 | 0.897 | 0.898 | 0.894 |
| | | 3 × I | 0.89 | 0.895 | 0.895 | 0.893 | 0.896 | 0.895 | 0.897 | 0.897 | 0.9 | 0.897 | 0.898 | 0.897 | 0.896 | 0.895 | **0.902** | 0.9 |
| | | 4 × I | 0.89 | 0.889 | 0.897 | 0.895 | 0.893 | 0.898 | 0.894 | 0.899 | 0.896 | 0.895 | 0.898 | 0.897 | 0.898 | 0.901 | 0.898 | 0.898 |
| | | 5 × I | 0.89 | 0.891 | 0.895 | 0.895 | 0.896 | 0.896 | 0.899 | 0.897 | 0.898 | 0.9 | 0.899 | 0.898 | 0.9 | 0.901 | 0.9 | **0.902** |
| | 7 | 1 × I | 0.882 | 0.886 | 0.887 | 0.891 | 0.892 | 0.891 | 0.889 | 0.893 | 0.892 | 0.894 | 0.894 | 0.893 | 0.893 | 0.896 | 0.893 | 0.896 |
| | | 2 × I | 0.881 | 0.889 | 0.89 | 0.895 | 0.896 | 0.895 | 0.892 | 0.894 | 0.897 | 0.892 | 0.896 | 0.899 | 0.896 | 0.897 | 0.895 | 0.896 |
| | | 3 × I | 0.859 | 0.891 | 0.894 | 0.891 | 0.895 | 0.896 | 0.897 | 0.9 | 0.897 | 0.896 | 0.898 | 0.896 | 0.898 | 0.897 | 0.901 | 0.895 |
| | | 4 × I | 0.888 | 0.894 | 0.893 | 0.895 | 0.897 | 0.893 | 0.895 | 0.898 | 0.899 | 0.896 | 0.897 | 0.897 | 0.898 | 0.9 | 0.899 | 0.899 |
| | | 5 × I | 0.889 | 0.893 | 0.895 | 0.9 | 0.896 | 0.898 | 0.898 | 0.9 | 0.897 | 0.896 | 0.898 | 0.901 | 0.902 | 0.899 | **0.903** | 0.899 |
| | 9 | 1 × I | 0.643 | 0.723 | 0.8 | 0.887 | 0.885 | 0.887 | 0.84 | 0.884 | 0.889 | 0.893 | 0.832 | 0.839 | 0.894 | 0.892 | 0.891 | 0.892 |
| | | 2 × I | 0.675 | 0.885 | 0.887 | 0.89 | 0.834 | 0.894 | 0.896 | 0.896 | 0.898 | 0.887 | 0.886 | 0.896 | 0.894 | 0.9 | 0.897 | 0.898 |
| | | 3 × I | 0.763 | 0.886 | 0.884 | 0.862 | 0.853 | 0.876 | 0.888 | 0.888 | 0.897 | 0.893 | 0.897 | 0.897 | 0.898 | 0.9 | 0.901 | 0.897 |
| | | 4 × I | 0.796 | 0.876 | 0.874 | 0.893 | 0.895 | 0.892 | 0.897 | 0.898 | 0.899 | 0.897 | 0.899 | 0.9 | 0.899 | 0.899 | 0.902 | **0.904** |
| | | 5 × I | 0.882 | 0.888 | 0.882 | 0.877 | 0.899 | 0.895 | 0.893 | 0.9 | 0.9 | 0.9 | 0.9 | 0.902 | 0.902 | 0.903 | 0.9 | 0.903 |
| | 11 | 1 × I | 0.6 | 0.609 | 0.815 | 0.88 | 0.789 | 0.889 | 0.808 | 0.858 | 0.893 | 0.889 | 0.848 | 0.888 | 0.88 | 0.871 | 0.891 | 0.893 |
| | | 2 × I | 0.721 | 0.819 | 0.818 | 0.835 | 0.863 | 0.889 | 0.847 | 0.853 | 0.894 | 0.894 | 0.898 | 0.859 | 0.859 | 0.897 | 0.896 | 0.898 |
| | | 3 × I | 0.79 | 0.811 | 0.837 | 0.884 | 0.841 | 0.893 | 0.853 | 0.896 | 0.895 | 0.881 | 0.9 | 0.898 | 0.898 | 0.898 | 0.901 | 0.899 |
| | | 4 × I | 0.704 | 0.848 | 0.843 | 0.844 | 0.873 | 0.887 | 0.898 | 0.896 | 0.895 | 0.899 | 0.899 | 0.886 | 0.902 | **0.903** | 0.902 | 0.902 |
| | | 5 × I | 0.816 | 0.886 | 0.857 | 0.893 | 0.891 | 0.897 | 0.897 | 0.899 | 0.899 | 0.893 | 0.898 | 0.9 | 0.901 | 0.899 | 0.9 | 0.901 |
| Dry Bean balanced | 3 | 1 × I | 0.887 | 0.889 | 0.892 | 0.897 | 0.894 | 0.893 | 0.891 | 0.896 | 0.895 | 0.897 | 0.898 | 0.896 | 0.895 | 0.901 | 0.898 | 0.901 |
| | | 2 × I | 0.885 | 0.891 | 0.894 | 0.893 | 0.896 | 0.9 | 0.896 | 0.899 | 0.897 | 0.898 | 0.899 | 0.901 | 0.9 | 0.899 | 0.908 | 0.899 |
| | | 3 × I | 0.886 | 0.894 | 0.895 | 0.892 | 0.897 | 0.894 | 0.896 | 0.9 | 0.899 | 0.901 | 0.898 | 0.899 | 0.906 | 0.899 | 0.902 | 0.9 |
| | | 4 × I | 0.888 | 0.895 | 0.893 | 0.902 | 0.899 | 0.908 | 0.905 | 0.901 | 0.897 | 0.899 | 0.908 | 0.899 | 0.902 | 0.897 | 0.905 | 0.907 |
| | | 5 × I | 0.891 | 0.894 | 0.898 | 0.899 | 0.897 | 0.904 | 0.9 | **0.912** | 0.898 | 0.902 | 0.906 | 0.904 | 0.909 | 0.902 | 0.905 | 0.906 |
| | 5 | 1 × I | 0.86 | 0.886 | 0.891 | 0.891 | 0.892 | 0.89 | 0.895 | 0.892 | 0.893 | 0.894 | 0.893 | 0.893 | 0.897 | 0.896 | 0.893 | 0.895 |
| | | 2 × I | 0.888 | 0.891 | 0.891 | 0.894 | 0.895 | 0.895 | 0.895 | 0.896 | 0.898 | 0.9 | 0.896 | 0.893 | 0.896 | 0.895 | 0.898 | 0.897 |
| | | 3 × I | 0.891 | 0.893 | 0.894 | 0.896 | 0.894 | 0.897 | 0.897 | 0.895 | 0.897 | 0.897 | 0.892 | 0.897 | 0.897 | 0.899 | 0.895 | 0.895 |
| | | 4 × I | 0.889 | 0.89 | 0.892 | 0.894 | 0.897 | 0.898 | 0.896 | 0.897 | 0.898 | 0.898 | 0.897 | 0.899 | 0.898 | 0.894 | 0.9 | 0.896 |
| | | 5 × I | 0.889 | 0.892 | 0.897 | 0.897 | 0.898 | 0.895 | 0.897 | 0.897 | 0.898 | **0.901** | 0.895 | 0.899 | 0.9 | 0.898 | 0.898 | 0.899 |
| | 7 | 1 × I | 0.867 | 0.88 | 0.888 | 0.891 | 0.89 | 0.89 | 0.891 | 0.892 | 0.894 | 0.892 | 0.894 | 0.893 | 0.894 | 0.894 | 0.894 | 0.894 |
| | | 2 × I | 0.808 | 0.886 | 0.89 | 0.892 | 0.898 | 0.895 | 0.895 | 0.897 | 0.894 | 0.893 | 0.895 | 0.898 | 0.897 | 0.897 | 0.897 | 0.901 |
| | | 3 × I | 0.881 | 0.893 | 0.892 | 0.891 | 0.897 | 0.896 | 0.898 | 0.895 | 0.897 | 0.898 | 0.898 | 0.897 | 0.899 | 0.899 | 0.899 | 0.899 |
| | | 4 × I | 0.882 | 0.893 | 0.893 | 0.894 | 0.9 | 0.897 | 0.896 | 0.896 | 0.897 | 0.9 | 0.9 | 0.898 | 0.898 | **0.903** | 0.899 | 0.897 |
| | | 5 × I | 0.888 | 0.895 | 0.891 | 0.898 | 0.894 | 0.9 | 0.901 | 0.899 | 0.899 | 0.9 | 0.897 | 0.901 | **0.903** | 0.9 | 0.899 | 0.899 |
| | 9 | 1 × I | 0.753 | 0.638 | 0.844 | 0.784 | 0.857 | 0.887 | 0.849 | 0.888 | 0.893 | 0.89 | 0.891 | 0.893 | 0.894 | 0.896 | 0.893 | 0.897 |
| | | 2 × I | 0.867 | 0.851 | 0.891 | 0.883 | 0.881 | 0.896 | 0.897 | 0.893 | 0.867 | 0.897 | 0.898 | 0.898 | 0.895 | 0.898 | 0.896 | 0.9 |
| | | 3 × I | 0.797 | 0.887 | 0.891 | 0.889 | 0.893 | 0.878 | 0.896 | 0.895 | 0.896 | 0.899 | 0.9 | 0.897 | 0.895 | 0.901 | 0.899 | 0.896 |
| | | 4 × I | 0.808 | 0.888 | 0.875 | 0.894 | 0.899 | 0.894 | 0.902 | 0.897 | 0.899 | 0.901 | 0.898 | 0.899 | 0.9 | **0.903** | 0.9 | 0.902 |
| | | 5 × I | 0.847 | 0.886 | 0.886 | 0.896 | 0.896 | 0.896 | 0.9 | 0.897 | 0.902 | 0.897 | 0.899 | 0.901 | 0.901 | 0.901 | 0.901 | 0.9 |
| | 11 | 1 × I | 0.733 | 0.633 | 0.822 | 0.883 | 0.885 | 0.877 | 0.824 | 0.893 | 0.882 | 0.893 | 0.871 | 0.877 | 0.891 | 0.894 | 0.896 | 0.895 |
| | | 2 × I | 0.588 | 0.749 | 0.887 | 0.86 | 0.889 | 0.892 | 0.891 | 0.893 | 0.894 | 0.893 | 0.895 | 0.895 | 0.896 | 0.895 | 0.896 | 0.899 |
| | | 3 × I | 0.839 | 0.882 | 0.85 | 0.865 | 0.887 | 0.885 | 0.891 | 0.877 | 0.897 | 0.896 | 0.899 | 0.896 | 0.899 | 0.899 | 0.897 | 0.899 |
| | | 4 × I | 0.759 | 0.836 | 0.847 | 0.896 | 0.894 | 0.892 | 0.895 | 0.895 | 0.896 | 0.899 | 0.897 | 0.897 | 0.897 | 0.898 | **0.901** | 0.899 |
| | | 5 × I | 0.825 | 0.889 | 0.888 | 0.889 | 0.894 | 0.873 | 0.898 | 0.897 | 0.9 | 0.9 | 0.898 | 0.899 | 0.899 | 0.897 | 0.9 | 0.899 |

**Table 8. Results of classification accuracy *acc* for the proposed approach and the Sensorless data sets: Two hidden layer, the method to substitute values of missing attributes in local tables—One artificial objects generated based on one original object, MLP networks aggregation using average of weights and various number of neurons in the hidden layer (1AO-2HL-AVG). Designation I is used for the number of neurons in the input layer.**

| Data set | No. tables | Second HL | No. of neurons in the first hidden layer | | | | | | | | | | | | | | | |
|---|---|---|---|---|---|---|---|---|---|---|---|---|---|---|---|---|---|---|
| | | | 0.25 × I | 0.5 × I | 0.75 × I | 1 × I | 1.5 × I | 1.75 × I | 2 × I | 2.5 × I | 2.75 × I | 3 × I | 3.5 × I | 3.75 × I | 4 × I | 4.5 × I | 4.75 × I | 5 × I |
| Sensorless imbalanced | 3 | 1 × I | 0.895 | 0.92 | 0.92 | 0.933 | 0.923 | 0.93 | 0.925 | 0.937 | 0.93 | 0.935 | 0.938 | 0.933 | 0.936 | 0.936 | 0.938 | 0.933 |
| | | 2 × I | 0.915 | 0.932 | 0.93 | 0.936 | 0.94 | 0.937 | 0.938 | 0.939 | 0.942 | 0.938 | 0.942 | 0.942 | 0.944 | 0.935 | 0.94 | 0.939 |
| | | 3 × I | 0.928 | 0.933 | 0.937 | 0.936 | 0.943 | 0.94 | 0.942 | 0.944 | 0.943 | 0.944 | 0.942 | 0.939 | 0.943 | 0.943 | 0.945 | 0.942 |
| | | 4 × I | 0.926 | 0.93 | 0.933 | 0.943 | 0.94 | 0.942 | 0.944 | 0.947 | 0.949 | 0.945 | 0.944 | 0.94 | 0.943 | 0.942 | 0.942 | 0.945 |
| | | 5 × I | 0.927 | 0.94 | 0.935 | 0.944 | 0.946 | 0.944 | 0.945 | **0.95** | 0.944 | 0.945 | 0.948 | 0.945 | 0.942 | 0.943 | 0.943 | 0.944 |
| | 5 | 1 × I | 0.867 | 0.903 | 0.902 | 0.916 | 0.919 | 0.92 | 0.925 | 0.923 | 0.93 | 0.925 | 0.921 | 0.93 | 0.931 | 0.926 | 0.927 | 0.928 |
| | | 2 × I | 0.909 | 0.916 | 0.921 | 0.924 | 0.927 | 0.927 | 0.93 | 0.934 | 0.933 | 0.935 | 0.936 | 0.931 | 0.935 | 0.938 | 0.936 | 0.934 |
| | | 3 × I | 0.909 | 0.919 | 0.92 | 0.934 | 0.932 | 0.929 | 0.938 | 0.937 | 0.938 | 0.936 | 0.94 | 0.941 | 0.939 | 0.94 | 0.939 | 0.939 |
| | | 4 × I | 0.917 | 0.92 | 0.935 | 0.932 | 0.936 | 0.941 | 0.936 | 0.937 | 0.945 | 0.945 | 0.938 | 0.941 | 0.94 | 0.94 | 0.942 | 0.94 |
| | | 5 × I | 0.912 | 0.923 | 0.928 | 0.939 | 0.935 | 0.941 | 0.94 | 0.937 | 0.938 | 0.942 | 0.944 | 0.941 | **0.945** | 0.94 | 0.937 | 0.937 |
| | 7 | 1 × I | 0.829 | 0.892 | 0.905 | 0.911 | 0.914 | 0.92 | 0.919 | 0.925 | 0.922 | 0.925 | 0.925 | 0.923 | 0.925 | 0.93 | 0.927 | 0.929 |
| | | 2 × I | 0.9 | 0.916 | 0.918 | 0.92 | 0.925 | 0.928 | 0.928 | 0.929 | 0.932 | 0.933 | 0.932 | 0.934 | 0.937 | 0.939 | 0.938 | 0.935 |
| | | 3 × I | 0.911 | 0.919 | 0.921 | 0.926 | 0.938 | 0.929 | 0.934 | 0.934 | 0.938 | 0.939 | 0.943 | 0.94 | 0.938 | 0.939 | 0.943 | 0.941 |
| | | 4 × I | 0.917 | 0.922 | 0.927 | 0.929 | 0.941 | 0.935 | 0.937 | 0.935 | 0.942 | 0.938 | 0.933 | 0.946 | 0.935 | 0.939 | 0.941 | **0.947** |
| | | 5 × I | 0.913 | 0.927 | 0.927 | 0.941 | 0.928 | 0.934 | 0.939 | 0.942 | 0.945 | 0.941 | 0.941 | 0.944 | 0.943 | 0.938 | 0.937 | 0.94 |
| | 9 | 1 × I | 0.875 | 0.901 | 0.91 | 0.908 | 0.923 | 0.915 | 0.919 | 0.92 | 0.915 | 0.923 | 0.917 | 0.925 | 0.928 | 0.92 | 0.925 | 0.918 |
| | | 2 × I | 0.907 | 0.912 | 0.921 | 0.921 | 0.921 | 0.924 | 0.933 | 0.924 | 0.93 | 0.93 | 0.928 | 0.929 | 0.931 | 0.934 | 0.932 | 0.932 |
| | | 3 × I | 0.913 | 0.916 | 0.928 | 0.925 | 0.927 | 0.924 | 0.941 | 0.932 | 0.94 | 0.932 | 0.936 | 0.936 | 0.942 | 0.939 | 0.935 | 0.934 |
| | | 4 × I | 0.916 | 0.922 | 0.931 | 0.923 | 0.932 | 0.939 | 0.94 | 0.937 | 0.935 | 0.935 | 0.939 | 0.933 | 0.936 | 0.939 | 0.942 | 0.934 |
| | | 5 × I | 0.923 | 0.921 | 0.927 | 0.934 | 0.931 | 0.934 | 0.934 | 0.939 | 0.941 | 0.932 | 0.936 | 0.937 | 0.942 | 0.938 | 0.943 | 0.935 |
| | 11 | 1 × I | 0.844 | 0.903 | 0.916 | 0.912 | 0.908 | 0.914 | 0.923 | 0.915 | 0.927 | 0.925 | 0.917 | 0.927 | 0.924 | 0.914 | 0.929 | 0.928 |
| | | 2 × I | 0.901 | 0.911 | 0.912 | 0.914 | 0.927 | 0.924 | 0.922 | 0.924 | 0.927 | 0.93 | 0.932 | 0.925 | 0.926 | 0.933 | 0.925 | 0.925 |
| | | 3 × I | 0.905 | 0.92 | 0.922 | 0.925 | 0.922 | 0.929 | 0.937 | 0.934 | 0.93 | 0.931 | 0.936 | 0.932 | 0.933 | 0.934 | 0.926 | 0.936 |
| | | 4 × I | 0.909 | 0.919 | 0.926 | 0.932 | 0.93 | 0.934 | 0.937 | 0.935 | 0.939 | 0.938 | 0.931 | 0.934 | 0.935 | 0.938 | **0.942** | 0.935 |
| | | 5 × I | 0.917 | 0.926 | 0.929 | 0.927 | 0.936 | 0.933 | 0.936 | 0.937 | 0.929 | 0.938 | 0.932 | 0.931 | 0.933 | 0.941 | 0.939 | 0.935 |

**Table 9. Results of classification accuracy *acc* for the proposed approach and the Crowd Sourced data sets: Two hidden layer, the method to substitute values of missing attributes in local tables—One artificial objects generated based on one original object, MLP networks aggregation using average of weights and various number of neurons in the hidden layer (1AO-2HL-AVG).** Designation I is used for the number of neurons in the input layer.

| Data set | No. tables | Second HL | No. of neurons in hidden layer | | | | | | | | | | | | | | | |
|---|---|---|---|---|---|---|---|---|---|---|---|---|---|---|---|---|---|---|
| | | | 0.25 × I | 0.5 × I | 0.75 × I | 1 × I | 1.5 × I | 1.75 × I | 2 × I | 2.5 × I | 2.75 × I | 3 × I | 3.5 × I | 3.75 × I | 4 × I | 4.5 × I | 4.75 × I | 5 × I |
| Crowd Sourced imbalanced | 3 | 1 × I | 0.706 | 0.765 | 0.782 | 0.79 | 0.782 | 0.794 | 0.797 | 0.809 | 0.827 | 0.827 | 0.821 | 0.82 | 0.822 | 0.812 | 0.821 | 0.824 |
| | | 2 × I | 0.754 | 0.777 | 0.777 | 0.791 | 0.806 | 0.819 | 0.806 | 0.814 | 0.827 | 0.828 | 0.817 | 0.812 | 0.812 | 0.827 | 0.813 | 0.83 |
| | | 3 × I | 0.759 | 0.777 | 0.815 | 0.779 | 0.813 | 0.805 | 0.792 | 0.818 | 0.823 | 0.827 | 0.82 | 0.818 | 0.814 | 0.818 | 0.822 | 0.824 |
| | | 4 × I | 0.734 | 0.81 | 0.807 | 0.791 | 0.804 | 0.802 | 0.806 | 0.822 | 0.816 | 0.831 | 0.822 | 0.813 | 0.809 | 0.836 | **0.838** | 0.815 |
| | | 5 × I | 0.759 | 0.807 | 0.811 | 0.798 | 0.799 | 0.801 | 0.822 | 0.823 | 0.807 | 0.795 | 0.827 | 0.828 | 0.814 | 0.817 | 0.823 | 0.832 |
| | 5 | 1 × I | 0.832 | 0.84 | 0.838 | 0.838 | 0.85 | 0.851 | 0.859 | 0.856 | 0.857 | **0.875** | 0.86 | 0.863 | 0.872 | 0.864 | 0.852 | 0.851 |
| | | 2 × I | 0.823 | 0.842 | 0.83 | 0.85 | 0.843 | 0.853 | 0.847 | 0.86 | 0.843 | 0.859 | 0.863 | 0.851 | 0.846 | 0.847 | 0.861 | 0.855 |
| | | 3 × I | 0.823 | 0.822 | 0.849 | 0.841 | 0.853 | 0.84 | 0.847 | 0.839 | 0.844 | 0.832 | 0.85 | 0.847 | 0.847 | 0.854 | 0.864 | 0.846 |
| | | 4 × I | 0.801 | 0.825 | 0.835 | 0.841 | 0.842 | 0.85 | 0.851 | 0.842 | 0.854 | 0.833 | 0.866 | 0.853 | 0.838 | 0.847 | 0.839 | 0.838 |
| | | 5 × I | 0.808 | 0.83 | 0.842 | 0.839 | 0.833 | 0.846 | 0.846 | 0.818 | 0.853 | 0.859 | 0.834 | 0.847 | 0.834 | 0.855 | 0.833 | 0.87 |
| | 7 | 1 × I | 0.817 | 0.823 | 0.818 | 0.845 | 0.846 | 0.835 | **0.855** | 0.847 | 0.835 | 0.839 | 0.854 | 0.84 | 0.853 | 0.837 | 0.846 | 0.85 |
| | | 2 × I | 0.816 | 0.834 | 0.835 | 0.851 | 0.824 | 0.833 | 0.841 | 0.849 | 0.849 | 0.846 | 0.843 | 0.847 | 0.847 | 0.834 | 0.841 | 0.843 |
| | | 3 × I | 0.817 | 0.825 | 0.837 | 0.824 | 0.832 | 0.846 | 0.842 | 0.841 | 0.832 | 0.832 | 0.838 | 0.833 | 0.85 | 0.835 | 0.844 | 0.831 |
| | | 4 × I | 0.798 | 0.8 | 0.811 | 0.826 | 0.837 | 0.843 | 0.806 | 0.843 | 0.842 | 0.834 | 0.848 | 0.845 | 0.825 | 0.839 | 0.839 | 0.826 |
| | | 5 × I | 0.8 | 0.813 | 0.818 | 0.809 | 0.839 | 0.824 | 0.814 | 0.821 | 0.83 | 0.826 | 0.846 | 0.836 | 0.836 | 0.836 | 0.843 | 0.829 |
| | 9 | 1 × I | 0.808 | 0.829 | 0.835 | 0.835 | 0.852 | 0.847 | 0.84 | 0.837 | 0.833 | 0.835 | 0.842 | 0.841 | 0.847 | 0.843 | 0.852 | 0.845 |
| | | 2 × I | 0.81 | 0.827 | 0.841 | 0.845 | 0.835 | 0.858 | 0.835 | 0.84 | 0.84 | 0.844 | 0.846 | 0.846 | 0.843 | 0.842 | **0.859** | 0.848 |
| | | 3 × I | 0.815 | 0.824 | 0.83 | 0.838 | 0.835 | 0.855 | 0.853 | 0.839 | 0.848 | 0.841 | 0.844 | 0.848 | 0.853 | 0.842 | 0.846 | 0.855 |
| | | 4 × I | 0.813 | 0.826 | 0.826 | 0.833 | 0.839 | 0.848 | 0.845 | 0.847 | 0.853 | 0.847 | 0.849 | 0.858 | 0.84 | 0.842 | 0.851 | 0.847 |
| | | 5 × I | 0.797 | 0.803 | 0.811 | 0.823 | 0.843 | 0.835 | 0.837 | 0.834 | 0.849 | 0.832 | 0.833 | 0.845 | 0.83 | 0.822 | 0.838 | 0.845 |
| | 11 | 1 × I | 0.829 | 0.829 | 0.845 | 0.851 | 0.854 | 0.854 | 0.847 | 0.858 | 0.863 | 0.864 | 0.856 | 0.859 | 0.856 | 0.862 | 0.856 | 0.862 |
| | | 2 × I | 0.817 | 0.842 | 0.833 | 0.859 | 0.862 | 0.846 | 0.849 | 0.855 | 0.862 | 0.843 | 0.851 | 0.851 | 0.86 | 0.857 | 0.863 | 0.855 |
| | | 3 × I | 0.802 | 0.835 | 0.829 | 0.837 | 0.847 | 0.851 | 0.862 | 0.865 | 0.861 | 0.865 | 0.859 | 0.85 | 0.854 | 0.86 | 0.857 | 0.853 |
| | | 4 × I | 0.826 | 0.822 | 0.842 | 0.855 | 0.842 | 0.857 | 0.847 | 0.848 | 0.858 | 0.853 | **0.876** | 0.862 | 0.857 | 0.861 | 0.86 | 0.865 |
| | | 5 × I | 0.806 | 0.833 | 0.849 | 0.852 | 0.847 | 0.85 | 0.854 | 0.855 | 0.859 | 0.849 | 0.864 | 0.862 | 0.865 | 0.841 | 0.864 | 0.85 |
| Crowd Sourced balanced | 3 | 1 × I | 0.675 | 0.749 | 0.793 | 0.821 | 0.776 | 0.831 | 0.86 | 0.85 | 0.858 | 0.875 | 0.875 | 0.882 | 0.874 | 0.893 | 0.869 | 0.874 |
| | | 2 × I | 0.66 | 0.774 | 0.784 | 0.844 | 0.845 | 0.851 | 0.819 | 0.868 | 0.893 | 0.877 | 0.888 | 0.87 | 0.873 | 0.894 | 0.902 | 0.903 |
| | | 3 × I | 0.634 | 0.753 | 0.795 | 0.827 | 0.837 | 0.861 | 0.874 | 0.854 | 0.861 | 0.889 | 0.901 | 0.898 | 0.86 | 0.907 | 0.9 | 0.911 |
| | | 4 × I | 0.523 | 0.763 | 0.83 | 0.805 | 0.872 | 0.864 | 0.874 | 0.876 | 0.869 | 0.888 | 0.891 | 0.905 | 0.898 | 0.896 | 0.897 | 0.911 |
| | | 5 × I | 0.702 | 0.786 | 0.787 | 0.818 | 0.871 | 0.872 | 0.874 | 0.887 | 0.892 | 0.893 | 0.898 | 0.892 | 0.898 | 0.905 | 0.89 | **0.915** |
| | 5 | 1 × I | 0.821 | 0.849 | 0.867 | 0.882 | 0.892 | 0.901 | 0.909 | 0.908 | 0.91 | 0.915 | 0.91 | 0.905 | 0.913 | 0.918 | 0.9 | 0.907 |
| | | 2 × I | 0.812 | 0.863 | 0.887 | 0.894 | 0.902 | 0.908 | 0.898 | 0.912 | 0.912 | 0.915 | 0.919 | 0.92 | 0.917 | 0.917 | **0.927** | 0.913 |
| | | 3 × I | 0.822 | 0.88 | 0.896 | 0.895 | 0.913 | 0.904 | 0.908 | 0.914 | 0.912 | 0.914 | 0.906 | 0.916 | 0.92 | 0.917 | 0.917 | 0.916 |
| | | 4 × I | 0.845 | 0.878 | 0.898 | 0.889 | 0.9 | 0.906 | 0.909 | 0.916 | 0.907 | 0.917 | 0.911 | 0.917 | 0.915 | 0.913 | 0.918 | 0.915 |
| | | 5 × I | 0.838 | 0.869 | 0.895 | 0.899 | 0.899 | 0.911 | 0.908 | 0.909 | 0.909 | 0.91 | 0.921 | 0.922 | 0.916 | 0.914 | 0.917 | 0.909 |
| | 7 | 1 × I | 0.829 | 0.867 | 0.885 | 0.9 | 0.898 | 0.905 | 0.91 | 0.912 | **0.918** | 0.908 | 0.908 | 0.909 | 0.905 | 0.91 | 0.913 | **0.918** |
| | | 2 × I | 0.836 | 0.878 | 0.896 | 0.894 | 0.895 | 0.906 | 0.899 | 0.896 | 0.913 | 0.902 | 0.902 | 0.903 | 0.909 | 0.896 | 0.898 | 0.91 |
| | | 3 × I | 0.842 | 0.879 | 0.892 | 0.885 | 0.89 | 0.895 | 0.885 | 0.898 | 0.909 | 0.875 | 0.888 | 0.903 | 0.895 | 0.885 | 0.901 | 0.89 |
| | | 4 × I | 0.837 | 0.891 | 0.889 | 0.885 | 0.892 | 0.888 | 0.891 | 0.895 | 0.874 | 0.89 | 0.884 | 0.894 | 0.894 | 0.89 | 0.86 | 0.892 |
| | | 5 × I | 0.85 | 0.866 | 0.89 | 0.882 | 0.875 | 0.884 | 0.898 | 0.896 | 0.898 | 0.891 | 0.89 | 0.881 | 0.882 | 0.883 | 0.874 | 0.898 |
| | 9 | 1 × I | 0.838 | 0.878 | 0.885 | 0.89 | 0.903 | 0.904 | 0.891 | 0.911 | 0.907 | 0.895 | 0.896 | 0.896 | 0.9 | 0.91 | 0.867 | 0.905 |
| | | 2 × I | 0.836 | 0.87 | 0.893 | 0.888 | 0.89 | 0.906 | 0.898 | 0.902 | 0.886 | 0.888 | 0.913 | 0.87 | 0.894 | **0.915** | 0.912 | 0.912 |
| | | 3 × I | 0.825 | 0.835 | 0.869 | 0.869 | 0.896 | 0.898 | 0.869 | 0.874 | 0.9 | 0.907 | 0.89 | 0.893 | 0.897 | 0.886 | 0.879 | 0.862 |
| | | 4 × I | 0.831 | 0.852 | 0.882 | 0.881 | 0.865 | 0.888 | 0.88 | 0.883 | 0.896 | 0.886 | 0.892 | 0.878 | 0.892 | 0.879 | 0.899 | 0.898 |
| | | 5 × I | 0.854 | 0.856 | 0.847 | 0.869 | 0.883 | 0.817 | 0.85 | 0.886 | 0.883 | 0.828 | 0.872 | 0.895 | 0.839 | 0.904 | 0.877 | 0.895 |
| | 11 | 1 × I | 0.826 | 0.879 | 0.888 | 0.894 | 0.901 | 0.9 | 0.894 | 0.902 | 0.896 | 0.891 | **0.921** | 0.913 | 0.918 | 0.908 | 0.917 | 0.92 |
| | | 2 × I | 0.844 | 0.875 | 0.901 | 0.887 | 0.897 | 0.896 | 0.9 | 0.878 | 0.89 | 0.883 | 0.904 | 0.897 | 0.906 | 0.904 | 0.903 | 0.913 |
| | | 3 × I | 0.822 | 0.864 | 0.875 | 0.874 | 0.887 | 0.9 | 0.877 | 0.875 | 0.905 | 0.864 | 0.91 | 0.875 | 0.884 | 0.898 | 0.868 | 0.901 |
| | | 4 × I | 0.846 | 0.851 | 0.885 | 0.863 | 0.853 | 0.899 | 0.887 | 0.89 | 0.898 | 0.861 | 0.904 | 0.905 | 0.893 | 0.887 | 0.915 | 0.837 |
| | | 5 × I | 0.835 | 0.863 | 0.84 | 0.869 | 0.831 | 0.87 | 0.853 | 0.898 | 0.883 | 0.854 | 0.874 | 0.878 | 0.848 | 0.909 | 0.903 | 0.917 |

**Table 10. Results of classification accuracy *acc* for the proposed approach and the Vehicle data sets: Two hidden layers, the method to substitute values of missing attributes in local tables—Three artificial objects generated based on one original object, MLP networks aggregation using average of weights and various number of neurons in the hidden layer (3AO-2HL-AVG).** Designation I is used for the number of neurons in the input layer.

| Data set | No. tables | Second HL | No. of neurons in hidden layer | | | | | | | | | | | | | | | |
|---|---|---|---|---|---|---|---|---|---|---|---|---|---|---|---|---|---|---|
| | | | 0.25 × I | 0.5 × I | 0.75 × I | 1 × I | 1.5 × I | 1.75 × I | 2 × I | 2.5 × I | 2.75 × I | 3 × I | 3.5 × I | 3.75 × I | 4 × I | 4.5 × I | 4.75 × I | 5 × I |
| Vehicle imbalanced | 3 | 1 × I | 0.281 | 0.638 | 0.669 | 0.642 | 0.668 | 0.689 | 0.702 | 0.677 | 0.69 | 0.68 | 0.675 | 0.675 | 0.64 | 0.654 | 0.677 | 0.675 |
| | | 2 × I | 0.545 | 0.659 | 0.677 | 0.686 | 0.69 | 0.709 | 0.657 | 0.685 | 0.714 | 0.697 | 0.705 | 0.684 | 0.668 | 0.696 | 0.678 | 0.698 |
| | | 3 × I | 0.563 | 0.714 | 0.701 | 0.714 | 0.701 | 0.688 | 0.689 | 0.715 | 0.703 | 0.705 | 0.688 | 0.697 | 0.717 | 0.703 | 0.698 | 0.692 |
| | | 4 × I | 0.535 | 0.694 | 0.722 | 0.689 | **0.732** | 0.692 | 0.715 | 0.719 | 0.711 | 0.696 | 0.688 | 0.713 | 0.717 | 0.705 | 0.731 | 0.702 |
| | | 5 × I | 0.588 | 0.655 | 0.702 | 0.707 | 0.702 | 0.678 | 0.722 | 0.697 | 0.707 | 0.709 | 0.706 | 0.693 | 0.711 | 0.69 | 0.697 | 0.709 |
| | 5 | 1 × I | 0.472 | 0.594 | 0.66 | 0.681 | 0.682 | 0.66 | 0.66 | 0.677 | 0.659 | 0.677 | 0.68 | 0.673 | 0.671 | 0.669 | 0.671 | 0.689 |
| | | 2 × I | 0.28 | 0.661 | 0.676 | 0.665 | 0.667 | 0.676 | 0.669 | 0.686 | 0.648 | 0.689 | 0.673 | 0.69 | 0.677 | 0.685 | 0.686 | 0.678 |
| | | 3 × I | 0.283 | 0.629 | 0.694 | 0.696 | 0.705 | 0.696 | 0.692 | 0.686 | 0.66 | 0.699 | 0.694 | 0.681 | 0.678 | 0.672 | 0.699 | **0.711** |
| | | 4 × I | 0.505 | 0.655 | 0.701 | 0.636 | 0.686 | 0.703 | 0.69 | 0.668 | 0.672 | 0.673 | 0.665 | 0.693 | 0.689 | 0.682 | 0.696 | 0.685 |
| | | 5 × I | 0.564 | 0.68 | 0.673 | 0.685 | 0.698 | 0.685 | 0.684 | 0.671 | 0.701 | 0.667 | 0.706 | 0.692 | 0.688 | 0.694 | 0.669 | 0.688 |
| | 7 | 1 × I | 0.58 | 0.639 | 0.638 | 0.652 | 0.631 | 0.65 | 0.657 | 0.675 | 0.647 | 0.654 | 0.665 | 0.629 | 0.654 | 0.65 | 0.643 | 0.638 |
| | | 2 × I | 0.579 | 0.626 | 0.651 | 0.659 | 0.65 | 0.667 | 0.696 | 0.663 | 0.688 | 0.664 | 0.689 | 0.661 | 0.65 | 0.657 | 0.642 | 0.676 |
| | | 3 × I | 0.617 | 0.681 | 0.63 | 0.685 | 0.699 | 0.671 | 0.693 | 0.671 | 0.663 | 0.699 | **0.702** | 0.681 | 0.685 | 0.677 | 0.692 | 0.688 |
| | | 4 × I | 0.593 | 0.66 | 0.676 | 0.69 | 0.684 | 0.682 | 0.69 | 0.677 | 0.678 | 0.697 | 0.692 | 0.684 | 0.682 | 0.654 | 0.659 | 0.694 |
| | | 5 × I | 0.543 | 0.667 | 0.686 | 0.612 | 0.681 | 0.68 | 0.693 | 0.654 | 0.692 | 0.686 | 0.693 | 0.693 | 0.689 | 0.701 | 0.699 | 0.692 |
| | 9 | 1 × I | 0.537 | 0.496 | 0.613 | 0.633 | 0.612 | 0.559 | 0.626 | 0.605 | 0.617 | 0.629 | 0.587 | 0.646 | 0.634 | 0.65 | 0.638 | 0.614 |
| | | 2 × I | 0.587 | 0.598 | 0.64 | 0.656 | 0.639 | 0.598 | 0.661 | 0.671 | 0.625 | 0.638 | 0.612 | 0.642 | 0.656 | 0.652 | 0.643 | 0.627 |
| | | 3 × I | 0.517 | 0.572 | 0.651 | 0.635 | 0.668 | 0.64 | 0.639 | 0.65 | 0.669 | 0.668 | 0.643 | 0.652 | 0.65 | 0.669 | 0.63 | 0.643 |
| | | 4 × I | 0.322 | 0.587 | 0.623 | 0.659 | 0.634 | 0.686 | 0.629 | 0.682 | 0.655 | 0.64 | 0.638 | 0.659 | 0.663 | 0.631 | 0.659 | 0.68 |
| | | 5 × I | 0.52 | 0.642 | 0.621 | 0.655 | 0.647 | 0.639 | 0.654 | 0.652 | 0.657 | 0.63 | 0.661 | 0.667 | 0.676 | 0.639 | 0.646 | **0.686** |
| | 11 | 1 × I | 0.385 | 0.597 | 0.6 | 0.652 | 0.623 | 0.646 | 0.608 | 0.631 | 0.629 | 0.667 | 0.65 | 0.646 | 0.661 | 0.619 | 0.659 | 0.64 |
| | | 2 × I | 0.585 | 0.638 | 0.646 | 0.642 | 0.635 | 0.657 | 0.659 | 0.668 | 0.66 | 0.667 | 0.657 | 0.667 | 0.625 | 0.663 | 0.667 | 0.669 |
| | | 3 × I | 0.581 | 0.589 | 0.663 | 0.646 | 0.64 | 0.614 | 0.65 | 0.621 | 0.656 | 0.685 | 0.692 | 0.676 | 0.66 | 0.678 | 0.646 | 0.678 |
| | | 4 × I | 0.606 | 0.66 | 0.614 | 0.669 | 0.659 | 0.671 | 0.681 | 0.677 | 0.669 | **0.698** | 0.657 | 0.676 | 0.68 | 0.675 | 0.66 | 0.686 |
| | | 5 × I | 0.529 | 0.65 | 0.678 | 0.657 | 0.677 | 0.676 | 0.685 | 0.68 | 0.669 | 0.689 | 0.671 | 0.684 | 0.68 | 0.696 | 0.667 | 0.684 |
| Vehicle blanced | 3 | 1 × I | 0.571 | 0.684 | 0.69 | 0.724 | 0.72 | 0.713 | 0.73 | 0.709 | 0.693 | 0.703 | 0.715 | 0.699 | 0.707 | 0.759 | 0.726 | 0.283 |
| | | 2 × I | 0.703 | 0.702 | 0.726 | 0.739 | 0.724 | 0.718 | 0.739 | 0.743 | 0.752 | 0.731 | 0.736 | 0.731 | 0.732 | 0.73 | 0.734 | 0.718 |
| | | 3 × I | 0.698 | 0.715 | 0.702 | 0.745 | 0.734 | 0.728 | 0.722 | 0.738 | 0.719 | 0.736 | 0.73 | 0.732 | 0.727 | 0.711 | 0.747 | 0.745 |
| | | 4 × I | 0.49 | 0.752 | 0.757 | 0.727 | 0.731 | 0.723 | 0.726 | 0.719 | 0.752 | 0.726 | **0.76** | 0.745 | 0.743 | 0.751 | 0.74 | 0.743 |
| | | 5 × I | 0.669 | 0.681 | 0.73 | 0.722 | 0.724 | 0.739 | 0.702 | 0.732 | 0.744 | 0.722 | 0.74 | 0.736 | 0.72 | 0.724 | 0.749 | 0.744 |
| | 5 | 1 × I | 0.381 | 0.694 | 0.626 | 0.698 | 0.686 | 0.688 | 0.707 | 0.663 | 0.682 | 0.696 | 0.71 | 0.714 | 0.686 | 0.693 | 0.697 | 0.689 |
| | | 2 × I | 0.542 | 0.669 | 0.722 | 0.685 | 0.697 | 0.707 | 0.719 | 0.715 | 0.706 | 0.703 | 0.728 | 0.709 | 0.717 | 0.707 | 0.703 | 0.69 |
| | | 3 × I | 0.594 | 0.698 | 0.68 | 0.698 | 0.697 | 0.717 | 0.706 | 0.713 | **0.732** | 0.711 | 0.72 | 0.701 | 0.72 | 0.705 | 0.717 | 0.711 |
| | | 4 × I | 0.568 | 0.705 | 0.72 | 0.694 | 0.718 | 0.703 | 0.726 | 0.705 | 0.703 | 0.718 | 0.715 | 0.73 | 0.707 | 0.707 | 0.715 | 0.701 |
| | | 5 × I | 0.686 | 0.711 | 0.714 | 0.714 | 0.709 | 0.718 | 0.718 | 0.698 | 0.72 | 0.71 | 0.703 | 0.709 | 0.705 | 0.706 | 0.688 | 0.697 |
| | 7 | 1 × I | 0.448 | 0.587 | 0.673 | 0.682 | 0.682 | 0.678 | 0.692 | 0.692 | 0.71 | 0.702 | 0.709 | 0.682 | 0.718 | 0.688 | 0.718 | 0.646 |
| | | 2 × I | 0.605 | 0.64 | 0.684 | 0.672 | 0.705 | 0.706 | 0.705 | 0.71 | 0.711 | 0.689 | 0.71 | 0.699 | 0.719 | 0.706 | 0.718 | 0.735 |
| | | 3 × I | 0.283 | 0.702 | 0.684 | 0.703 | 0.718 | 0.707 | 0.734 | 0.702 | 0.709 | 0.703 | 0.71 | 0.677 | 0.723 | 0.735 | 0.718 | 0.718 |
| | | 4 × I | 0.513 | 0.672 | 0.673 | 0.73 | 0.731 | 0.724 | 0.713 | 0.728 | 0.726 | 0.728 | 0.699 | **0.74** | 0.701 | 0.69 | 0.726 | 0.711 |
| | | 5 × I | 0.283 | 0.735 | 0.723 | 0.711 | 0.724 | 0.71 | 0.705 | 0.711 | 0.719 | 0.718 | 0.722 | 0.735 | 0.71 | 0.693 | 0.707 | 0.738 |
| | 9 | 1 × I | 0.382 | 0.6 | 0.671 | 0.63 | 0.47 | 0.664 | 0.688 | 0.668 | 0.65 | 0.668 | 0.697 | 0.663 | 0.684 | 0.681 | 0.682 | 0.665 |
| | | 2 × I | 0.563 | 0.622 | 0.659 | 0.692 | 0.699 | 0.703 | 0.685 | 0.69 | 0.684 | 0.707 | 0.685 | 0.714 | 0.703 | 0.71 | 0.709 | 0.705 |
| | | 3 × I | 0.283 | 0.65 | 0.659 | 0.694 | 0.685 | 0.699 | 0.718 | 0.702 | 0.684 | 0.685 | 0.717 | 0.726 | 0.717 | 0.703 | 0.669 | 0.714 |
| | | 4 × I | 0.618 | 0.629 | 0.71 | 0.719 | 0.698 | 0.709 | 0.705 | 0.672 | 0.664 | 0.709 | 0.705 | 0.705 | 0.701 | 0.676 | 0.676 | 0.689 |
| | | 5 × I | 0.64 | 0.654 | 0.703 | 0.685 | 0.68 | 0.684 | 0.673 | 0.715 | 0.723 | 0.705 | 0.707 | 0.685 | **0.728** | 0.696 | 0.678 | 0.698 |
| | 11 | 1 × I | 0.588 | 0.566 | 0.575 | 0.588 | 0.64 | 0.673 | 0.647 | 0.66 | 0.626 | 0.661 | 0.656 | 0.61 | 0.671 | 0.655 | 0.664 | 0.676 |
| | | 2 × I | 0.577 | 0.6 | 0.631 | 0.661 | 0.652 | 0.657 | 0.655 | 0.657 | 0.69 | 0.686 | 0.685 | 0.686 | 0.698 | 0.673 | 0.68 | 0.682 |
| | | 3 × I | 0.6 | 0.581 | 0.657 | 0.664 | 0.681 | 0.675 | 0.703 | 0.694 | 0.699 | 0.694 | 0.685 | 0.685 | 0.696 | 0.669 | 0.709 | 0.677 |
| | | 4 × I | 0.556 | 0.642 | 0.701 | 0.669 | 0.694 | 0.68 | 0.705 | 0.675 | 0.703 | 0.699 | 0.684 | 0.707 | 0.702 | 0.696 | 0.689 | 0.701 |
| | | 5 × I | 0.629 | 0.602 | 0.678 | 0.696 | 0.682 | 0.693 | 0.684 | **0.71** | 0.672 | 0.661 | 0.69 | 0.689 | 0.69 | 0.707 | 0.69 | 0.686 |

**Table 11. Results of classification accuracy *acc* for the proposed approach and the Dry Bean data sets: Two hidden layers, the method to substitute values of missing attributes in local tables—Three artificial objects generated based on one original object, MLP networks aggregation using average of weights and various number of neurons in the hidden layer (3AO-2HL-AVG).** Designation I is used for the number of neurons in the input layer.

| Data set | No. tables | Second HL | No. of neurons in hidden layer | | | | | | | | | | | | | | | |
|---|---|---|---|---|---|---|---|---|---|---|---|---|---|---|---|---|---|---|
| | | | 0.25 × I | 0.5 × I | 0.75 × I | 1 × I | 1.5 × I | 1.75 × I | 2 × I | 2.5 × I | 2.75 × I | 3 × I | 3.5 × I | 3.75 × I | 4 × I | 4.5 × I | 4.75 × I | 5 × I |
| Dry Bean imbalanced | 3 | 1 × I | 0.888 | 0.892 | 0.893 | 0.894 | 0.895 | 0.893 | 0.895 | 0.894 | 0.894 | 0.897 | 0.895 | 0.899 | 0.898 | 0.898 | 0.899 | 0.895 |
| | | 2 × I | 0.89 | 0.893 | 0.894 | 0.892 | 0.895 | 0.896 | 0.895 | 0.893 | 0.897 | 0.896 | 0.895 | 0.897 | 0.897 | 0.897 | 0.899 | 0.902 |
| | | 3 × I | 0.891 | 0.892 | 0.894 | 0.895 | 0.896 | 0.897 | 0.899 | 0.896 | 0.901 | 0.897 | 0.899 | 0.899 | 0.902 | 0.899 | 0.904 | 0.906 |
| | | 4 × I | 0.891 | 0.894 | 0.897 | 0.897 | 0.897 | 0.898 | 0.899 | 0.895 | 0.899 | 0.9 | 0.899 | 0.9 | 0.902 | 0.898 | 0.904 | 0.903 |
| | | 5 × I | 0.893 | 0.892 | 0.892 | 0.898 | 0.9 | 0.899 | 0.899 | 0.9 | 0.901 | 0.904 | 0.897 | 0.903 | 0.904 | **0.91** | 0.902 | 0.902 |
| | 5 | 1 × I | 0.88 | 0.882 | 0.888 | 0.889 | 0.893 | 0.895 | 0.891 | 0.892 | 0.893 | 0.896 | 0.896 | 0.896 | 0.897 | 0.892 | 0.895 | 0.894 |
| | | 2 × I | 0.891 | 0.888 | 0.892 | 0.893 | 0.896 | 0.893 | 0.895 | 0.895 | 0.897 | 0.896 | 0.897 | 0.898 | 0.896 | 0.895 | 0.896 | 0.899 |
| | | 3 × I | 0.89 | 0.891 | 0.896 | 0.896 | 0.895 | 0.896 | 0.896 | 0.897 | 0.897 | 0.896 | 0.896 | 0.895 | 0.898 | 0.896 | 0.898 | 0.899 |
| | | 4 × I | 0.894 | 0.894 | 0.895 | 0.898 | 0.896 | 0.9 | 0.896 | 0.898 | 0.899 | 0.894 | 0.897 | 0.898 | 0.893 | 0.897 | 0.899 | 0.899 |
| | | 5 × I | 0.892 | 0.895 | 0.894 | 0.898 | 0.896 | 0.897 | 0.898 | 0.897 | 0.897 | 0.899 | 0.896 | 0.899 | 0.897 | **0.901** | 0.899 | 0.898 |
| | 7 | 1 × I | 0.651 | 0.888 | 0.866 | 0.885 | 0.891 | 0.894 | 0.89 | 0.894 | 0.894 | 0.892 | 0.894 | 0.893 | 0.893 | 0.895 | 0.892 | 0.895 |
| | | 2 × I | 0.887 | 0.889 | 0.89 | 0.888 | 0.893 | 0.894 | 0.893 | 0.892 | 0.897 | 0.896 | 0.896 | 0.895 | 0.899 | 0.896 | 0.899 | 0.896 |
| | | 3 × I | 0.89 | 0.886 | 0.893 | 0.893 | 0.896 | 0.896 | 0.895 | 0.897 | 0.895 | 0.896 | 0.897 | 0.895 | 0.899 | 0.899 | 0.894 | 0.895 |
| | | 4 × I | 0.88 | 0.895 | 0.894 | 0.893 | 0.896 | 0.897 | 0.897 | 0.894 | 0.898 | 0.897 | 0.897 | 0.9 | 0.897 | 0.897 | 0.899 | 0.9 |
| | | 5 × I | 0.889 | 0.889 | 0.893 | 0.897 | 0.892 | 0.895 | 0.895 | 0.896 | 0.902 | 0.896 | 0.897 | 0.899 | 0.896 | 0.898 | **0.902** | 0.899 |
| | 9 | 1 × I | 0.631 | 0.885 | 0.879 | 0.886 | 0.82 | 0.888 | 0.889 | 0.894 | 0.888 | 0.888 | 0.886 | 0.889 | 0.894 | 0.892 | 0.893 | 0.893 |
| | | 2 × I | 0.702 | 0.873 | 0.866 | 0.889 | 0.886 | 0.889 | 0.893 | 0.892 | 0.896 | 0.893 | 0.892 | 0.899 | 0.895 | 0.901 | 0.896 | 0.897 |
| | | 3 × I | 0.836 | 0.836 | 0.889 | 0.887 | 0.893 | 0.896 | 0.898 | 0.897 | 0.897 | 0.897 | 0.898 | 0.898 | 0.898 | 0.899 | 0.899 | 0.898 |
| | | 4 × I | 0.886 | 0.86 | 0.873 | 0.892 | 0.896 | 0.899 | 0.899 | 0.899 | 0.899 | 0.899 | 0.901 | 0.899 | 0.897 | 0.9 | 0.901 | 0.901 |
| | | 5 × I | 0.866 | 0.892 | 0.889 | 0.894 | 0.898 | 0.897 | 0.902 | 0.897 | 0.9 | 0.902 | 0.901 | 0.898 | 0.901 | 0.897 | 0.9 | **0.903** |
| | 11 | 1 × I | 0.642 | 0.672 | 0.738 | 0.823 | 0.865 | 0.8 | 0.885 | 0.891 | 0.889 | 0.843 | 0.863 | 0.893 | 0.851 | 0.857 | 0.892 | 0.893 |
| | | 2 × I | 0.705 | 0.742 | 0.831 | 0.826 | 0.889 | 0.847 | 0.854 | 0.897 | 0.892 | 0.894 | 0.889 | 0.889 | 0.88 | 0.881 | 0.894 | 0.897 |
| | | 3 × I | 0.8 | 0.835 | 0.827 | 0.837 | 0.855 | 0.893 | 0.868 | 0.894 | 0.897 | 0.896 | 0.898 | 0.895 | 0.888 | 0.901 | 0.895 | 0.9 |
| | | 4 × I | 0.757 | 0.833 | 0.855 | 0.875 | 0.895 | 0.867 | 0.895 | 0.899 | 0.897 | 0.897 | 0.899 | 0.876 | 0.902 | 0.902 | 0.9 | 0.9 |
| | | 5 × I | 0.831 | 0.84 | 0.888 | 0.87 | 0.854 | 0.896 | 0.871 | 0.896 | 0.899 | 0.896 | 0.873 | 0.899 | 0.902 | 0.9 | **0.903** | **0.903** |
| Dry Bean blanced | 3 | 1 × I | 0.889 | 0.889 | 0.892 | 0.895 | 0.896 | 0.894 | 0.894 | 0.897 | 0.896 | 0.897 | 0.898 | 0.902 | 0.902 | 0.9 | 0.899 | 0.903 |
| | | 2 × I | 0.894 | 0.892 | 0.895 | 0.896 | 0.899 | 0.899 | 0.903 | 0.9 | 0.904 | 0.9 | 0.905 | 0.898 | 0.903 | 0.899 | 0.9 | 0.901 |
| | | 3 × I | 0.892 | 0.894 | 0.894 | 0.898 | 0.897 | 0.899 | 0.903 | 0.899 | 0.901 | 0.905 | 0.903 | 0.903 | 0.904 | 0.907 | 0.907 | 0.906 |
| | | 4 × I | 0.897 | 0.895 | 0.896 | 0.903 | 0.896 | 0.899 | 0.901 | 0.901 | 0.903 | 0.904 | 0.898 | 0.901 | 0.907 | 0.91 | 0.906 | 0.903 |
| | | 5 × I | 0.89 | 0.895 | 0.896 | 0.896 | 0.899 | 0.9 | 0.901 | 0.909 | 0.907 | 0.904 | 0.904 | 0.899 | 0.91 | 0.905 | **0.918** | 0.907 |
| | 5 | 1 × I | 0.888 | 0.89 | 0.889 | 0.89 | 0.894 | 0.893 | 0.893 | 0.894 | 0.895 | 0.895 | 0.896 | 0.896 | 0.895 | 0.895 | 0.895 | 0.899 |
| | | 2 × I | 0.885 | 0.894 | 0.891 | 0.893 | 0.897 | 0.893 | 0.897 | 0.895 | 0.894 | 0.898 | 0.897 | 0.898 | 0.897 | 0.897 | 0.893 | 0.9 |
| | | 3 × I | 0.886 | 0.889 | 0.895 | 0.897 | 0.894 | 0.896 | 0.894 | 0.897 | 0.895 | 0.899 | 0.896 | 0.898 | 0.896 | 0.899 | 0.899 | 0.9 |
| | | 4 × I | 0.893 | 0.893 | 0.897 | 0.894 | 0.895 | 0.893 | 0.897 | 0.898 | 0.898 | 0.898 | 0.897 | 0.898 | **0.901** | 0.896 | **0.901** | **0.901** |
| | | 5 × I | 0.889 | 0.893 | 0.894 | 0.895 | 0.895 | 0.9 | 0.9 | 0.9 | 0.9 | 0.899 | 0.9 | 0.899 | 0.897 | 0.899 | 0.899 | 0.9 |
| | 7 | 1 × I | 0.593 | 0.882 | 0.889 | 0.886 | 0.891 | 0.895 | 0.892 | 0.895 | 0.894 | 0.892 | 0.894 | 0.893 | 0.891 | 0.895 | 0.895 | 0.894 |
| | | 2 × I | 0.885 | 0.85 | 0.892 | 0.893 | 0.893 | 0.897 | 0.892 | 0.895 | 0.896 | 0.896 | 0.896 | 0.895 | 0.898 | 0.896 | 0.898 | 0.897 |
| | | 3 × I | 0.878 | 0.888 | 0.891 | 0.89 | 0.895 | 0.892 | 0.894 | 0.896 | 0.896 | 0.898 | 0.897 | 0.897 | 0.896 | 0.898 | 0.901 | 0.898 |
| | | 4 × I | 0.882 | 0.892 | 0.895 | 0.893 | 0.896 | 0.897 | 0.898 | 0.898 | 0.9 | 0.894 | 0.895 | 0.898 | 0.895 | 0.896 | 0.898 | 0.897 |
| | | 5 × I | 0.891 | 0.895 | 0.897 | 0.898 | 0.897 | 0.897 | 0.899 | 0.897 | 0.899 | 0.899 | 0.899 | 0.898 | 0.897 | 0.9 | 0.901 | **0.902** |
| | 9 | 1 × I | 0.688 | 0.842 | 0.733 | 0.829 | 0.871 | 0.89 | 0.885 | 0.89 | 0.893 | 0.888 | 0.891 | 0.889 | 0.897 | 0.894 | 0.891 | 0.895 |
| | | 2 × I | 0.681 | 0.818 | 0.886 | 0.893 | 0.891 | 0.881 | 0.893 | 0.895 | 0.893 | 0.897 | 0.896 | 0.896 | 0.898 | 0.898 | 0.9 | 0.896 |
| | | 3 × I | 0.88 | 0.836 | 0.888 | 0.842 | 0.895 | 0.899 | 0.895 | 0.897 | 0.896 | 0.9 | 0.899 | 0.897 | 0.899 | 0.898 | 0.897 | 0.9 |
| | | 4 × I | 0.766 | 0.891 | 0.865 | 0.893 | 0.897 | 0.871 | 0.897 | 0.9 | 0.895 | 0.899 | 0.899 | 0.902 | 0.903 | **0.904** | 0.902 | 0.901 |
| | | 5 × I | 0.855 | 0.892 | 0.886 | 0.897 | 0.9 | 0.897 | 0.897 | 0.9 | 0.9 | 0.901 | 0.903 | 0.901 | 0.9 | 0.9 | 0.903 | 0.901 |
| | 11 | 1 × I | 0.671 | 0.869 | 0.88 | 0.795 | 0.892 | 0.888 | 0.892 | 0.853 | 0.889 | 0.886 | 0.845 | 0.893 | 0.897 | 0.895 | 0.896 | 0.892 |
| | | 2 × I | 0.686 | 0.882 | 0.886 | 0.839 | 0.892 | 0.871 | 0.893 | 0.89 | 0.894 | 0.895 | 0.895 | 0.896 | 0.861 | 0.895 | 0.9 | 0.895 |
| | | 3 × I | 0.813 | 0.837 | 0.85 | 0.89 | 0.892 | 0.897 | 0.894 | 0.882 | 0.899 | 0.9 | 0.899 | 0.895 | 0.898 | 0.897 | 0.898 | 0.899 |
| | | 4 × I | 0.829 | 0.888 | 0.889 | 0.891 | 0.895 | 0.897 | 0.898 | 0.899 | 0.9 | 0.887 | 0.901 | 0.902 | 0.903 | 0.899 | 0.899 | 0.901 |
| | | 5 × I | 0.808 | 0.836 | 0.879 | 0.882 | 0.894 | 0.899 | 0.896 | 0.898 | **0.904** | 0.9 | 0.9 | 0.898 | 0.901 | 0.902 | 0.9 | 0.902 |

**Table 12. Results of classification accuracy *acc* for the proposed approach and the Sensorless data sets: Two hidden layer, the method to substitute values of missing attributes in local tables—Three artificial objects generated based on one original object, MLP networks aggregation using average of weights and various number of neurons in the hidden layer (3AO-2HL-AVG).** Designation I is used for the number of neurons in the input layer.

| Data set | No. tables | Second HL | No. of neurons in hidden layer | | | | | | | | | | | | | | | |
| --- | --- | --- | --- | --- | --- | --- | --- | --- | --- | --- | --- | --- | --- | --- | --- | --- | --- | --- |
| | | | 0.25 × I | 0.5 × I | 0.75 × I | 1 × I | 1.5 × I | 1.75 × I | 2 × I | 2.5 × I | 2.75 × I | 3 × I | 3.5 × I | 3.75 × I | 4 × I | 4.5 × I | 4.75 × I | 5 × I |
| Sensorless imbalanced | 3 | 1 × I | 0.892 | 0.921 | 0.926 | 0.926 | 0.92 | 0.929 | 0.938 | 0.933 | 0.934 | 0.935 | 0.93 | 0.936 | 0.937 | 0.942 | 0.936 | 0.938 |
| | | 2 × I | 0.921 | 0.928 | 0.935 | 0.94 | 0.939 | 0.942 | 0.938 | 0.938 | 0.941 | 0.934 | 0.937 | 0.94 | 0.948 | 0.945 | 0.944 | 0.94 |
| | | 3 × I | 0.932 | 0.936 | 0.94 | 0.944 | 0.941 | 0.948 | 0.944 | 0.941 | 0.948 | 0.946 | 0.943 | 0.943 | 0.945 | 0.942 | 0.945 | 0.943 |
| | | 4 × I | 0.935 | 0.951 | 0.942 | 0.945 | 0.945 | 0.949 | 0.948 | 0.944 | 0.949 | 0.946 | 0.945 | 0.947 | 0.948 | 0.941 | 0.934 | 0.94 |
| | | 5 × I | 0.941 | 0.94 | 0.949 | 0.946 | 0.944 | 0.949 | **0.951** | 0.948 | 0.947 | 0.948 | 0.941 | 0.945 | 0.944 | 0.945 | 0.944 | 0.945 |
| | 5 | 1 × I | 0.886 | 0.902 | 0.912 | 0.928 | 0.922 | 0.923 | 0.92 | 0.925 | 0.934 | 0.927 | 0.926 | 0.935 | 0.932 | 0.929 | 0.934 | 0.928 |
| | | 2 × I | 0.907 | 0.916 | 0.923 | 0.923 | 0.924 | 0.929 | 0.927 | 0.934 | 0.933 | 0.933 | 0.936 | 0.933 | 0.94 | 0.934 | 0.935 | 0.934 |
| | | 3 × I | 0.923 | 0.922 | 0.928 | 0.931 | 0.937 | 0.937 | 0.938 | 0.937 | 0.939 | 0.935 | 0.935 | 0.94 | 0.94 | 0.942 | 0.942 | 0.938 |
| | | 4 × I | 0.918 | 0.927 | 0.936 | 0.935 | 0.939 | 0.938 | 0.937 | 0.939 | 0.938 | 0.941 | 0.935 | 0.94 | 0.941 | 0.94 | 0.937 | 0.939 |
| | | 5 × I | 0.924 | 0.931 | 0.931 | 0.932 | 0.938 | 0.943 | 0.94 | 0.936 | 0.937 | **0.944** | 0.942 | 0.94 | 0.942 | 0.942 | 0.943 | 0.941 |
| | 7 | 1 × I | 0.894 | 0.911 | 0.913 | 0.912 | 0.914 | 0.919 | 0.917 | 0.92 | 0.917 | 0.927 | 0.926 | 0.92 | 0.924 | 0.926 | 0.929 | 0.929 |
| | | 2 × I | 0.909 | 0.913 | 0.924 | 0.923 | 0.925 | 0.929 | 0.928 | 0.935 | 0.931 | 0.924 | 0.936 | 0.927 | 0.934 | 0.934 | 0.933 | 0.932 |
| | | 3 × I | 0.916 | 0.918 | 0.927 | 0.925 | 0.936 | 0.929 | 0.934 | 0.93 | 0.936 | 0.933 | 0.937 | 0.935 | 0.934 | 0.941 | 0.935 | 0.932 |
| | | 4 × I | 0.917 | 0.927 | 0.929 | 0.927 | 0.934 | 0.935 | 0.934 | 0.932 | 0.93 | 0.935 | 0.937 | 0.938 | 0.937 | 0.942 | 0.942 | 0.932 |
| | | 5 × I | 0.914 | 0.926 | 0.931 | 0.933 | 0.939 | 0.937 | 0.937 | 0.938 | 0.94 | 0.942 | 0.939 | 0.938 | 0.939 | **0.943** | 0.941 | 0.941 |
| | 9 | 1 × I | 0.874 | 0.902 | 0.908 | 0.916 | 0.916 | 0.909 | 0.923 | 0.929 | 0.918 | 0.918 | 0.921 | 0.917 | 0.926 | 0.933 | 0.928 | 0.926 |
| | | 2 × I | 0.909 | 0.919 | 0.922 | 0.921 | 0.922 | 0.922 | 0.928 | 0.926 | 0.93 | 0.923 | 0.927 | 0.932 | 0.929 | 0.92 | 0.932 | 0.931 |
| | | 3 × I | 0.912 | 0.916 | 0.925 | 0.926 | 0.931 | 0.937 | 0.927 | 0.929 | 0.93 | 0.929 | 0.935 | 0.932 | 0.935 | 0.932 | 0.938 | 0.934 |
| | | 4 × I | 0.912 | 0.93 | 0.924 | 0.933 | 0.934 | 0.932 | 0.936 | 0.935 | 0.931 | 0.933 | 0.936 | 0.937 | 0.932 | 0.931 | 0.929 | 0.933 |
| | | 5 × I | 0.918 | 0.934 | 0.933 | 0.932 | 0.933 | 0.927 | 0.94 | 0.934 | 0.934 | **0.94** | 0.936 | 0.939 | 0.932 | 0.936 | 0.931 | 0.938 |
| | 11 | 1 × I | 0.884 | 0.903 | 0.901 | 0.916 | 0.92 | 0.926 | 0.921 | 0.923 | 0.927 | 0.922 | 0.918 | 0.924 | 0.922 | 0.929 | 0.921 | 0.928 |
| | | 2 × I | 0.9 | 0.909 | 0.927 | 0.924 | 0.926 | 0.918 | 0.923 | 0.922 | 0.926 | 0.923 | 0.921 | 0.926 | 0.923 | 0.925 | 0.93 | 0.93 |
| | | 3 × I | 0.913 | 0.916 | 0.922 | 0.922 | 0.924 | 0.927 | 0.929 | 0.924 | 0.931 | 0.927 | 0.932 | 0.93 | 0.929 | 0.937 | 0.928 | 0.931 |
| | | 4 × I | 0.912 | 0.915 | 0.931 | 0.922 | 0.933 | 0.939 | 0.935 | 0.931 | 0.932 | 0.925 | 0.926 | 0.933 | 0.934 | 0.927 | 0.936 | 0.926 |
| | | 5 × I | 0.917 | 0.927 | 0.933 | 0.93 | 0.937 | 0.931 | 0.935 | 0.938 | **0.939** | 0.927 | 0.93 | 0.932 | 0.93 | 0.925 | 0.934 | 0.932 |

**Table 13. Results of classification accuracy *acc* for the proposed approach and the Crowd Sourced data sets: Two hidden layers, the method to substitute values of missing attributes in local tables—Three artificial objects generated based on one original object, MLP networks aggregation using average of weights and various number of neurons in the hidden layer (3AO-2HL-AVG).** Designation I is used for the number of neurons in the input layer.

| Data set | No. tables | Second HL | No. of neurons in hidden layer | | | | | | | | | | | | | | | |
|---|---|---|---|---|---|---|---|---|---|---|---|---|---|---|---|---|---|---|
| | | | 0.25 × I | 0.5 × I | 0.75 × I | 1 × I | 1.5 × I | 1.75 × I | 2 × I | 2.5 × I | 2.75 × I | 3 × I | 3.5 × I | 3.75 × I | 4 × I | 4.5 × I | 4.75 × I | 5 × I |
| Crowd Sourced imbalanced | 3 | 1 × I | 0.717 | 0.762 | 0.755 | 0.748 | 0.773 | 0.796 | 0.756 | 0.788 | 0.772 | 0.806 | 0.788 | 0.808 | 0.801 | 0.825 | 0.8 | 0.805 |
| | | 2 × I | 0.651 | 0.759 | 0.763 | 0.77 | 0.799 | 0.8 | 0.764 | 0.782 | 0.788 | 0.796 | 0.819 | 0.816 | 0.817 | 0.819 | 0.813 | 0.791 |
| | | 3 × I | 0.7 | 0.769 | 0.8 | 0.749 | 0.787 | 0.794 | 0.798 | 0.808 | 0.793 | 0.803 | 0.82 | 0.794 | 0.796 | 0.808 | 0.822 | **0.833** |
| | | 4 × I | 0.688 | 0.733 | 0.765 | 0.794 | 0.782 | 0.8 | 0.823 | 0.807 | 0.818 | 0.798 | 0.821 | 0.823 | 0.806 | 0.802 | 0.809 | 0.79 |
| | | 5 × I | 0.7 | 0.758 | 0.764 | 0.794 | 0.794 | 0.796 | 0.784 | 0.781 | 0.792 | 0.814 | 0.79 | 0.803 | 0.813 | 0.802 | 0.825 | 0.83 |
| | 5 | 1 × I | 0.8 | 0.823 | 0.812 | 0.841 | 0.847 | 0.853 | 0.839 | 0.853 | 0.855 | 0.852 | 0.862 | 0.863 | 0.857 | **0.869** | 0.846 | 0.86 |
| | | 2 × I | 0.805 | 0.823 | 0.814 | 0.848 | 0.845 | 0.84 | 0.863 | 0.848 | 0.854 | 0.855 | 0.862 | 0.851 | 0.852 | 0.858 | 0.853 | 0.848 |
| | | 3 × I | 0.819 | 0.847 | 0.794 | 0.845 | 0.844 | 0.851 | 0.861 | 0.847 | 0.86 | 0.858 | 0.861 | 0.86 | 0.857 | 0.863 | 0.851 | 0.862 |
| | | 4 × I | 0.811 | 0.821 | 0.827 | 0.845 | 0.844 | 0.853 | 0.847 | 0.861 | 0.855 | 0.859 | 0.86 | 0.863 | 0.856 | 0.859 | 0.859 | 0.86 |
| | | 5 × I | 0.8 | 0.834 | 0.823 | 0.845 | 0.832 | 0.851 | 0.848 | 0.857 | 0.847 | 0.851 | 0.853 | 0.852 | 0.853 | 0.858 | 0.865 | 0.862 |
| | 7 | 1 × I | 0.794 | 0.841 | 0.833 | 0.849 | 0.859 | 0.852 | 0.855 | 0.859 | 0.85 | 0.848 | 0.851 | 0.86 | 0.846 | 0.861 | 0.854 | 0.847 |
| | | 2 × I | 0.822 | 0.83 | 0.841 | 0.842 | 0.852 | 0.852 | 0.858 | 0.852 | 0.858 | 0.846 | 0.866 | 0.854 | 0.855 | 0.848 | 0.864 | 0.856 |
| | | 3 × I | 0.822 | 0.819 | 0.854 | 0.847 | 0.858 | 0.86 | 0.863 | 0.858 | 0.849 | 0.847 | 0.847 | 0.847 | 0.84 | 0.852 | **0.867** | 0.853 |
| | | 4 × I | 0.799 | 0.825 | 0.836 | 0.841 | 0.834 | 0.853 | 0.86 | 0.854 | 0.849 | 0.859 | 0.853 | 0.854 | 0.852 | 0.865 | 0.853 | 0.853 |
| | | 5 × I | 0.821 | 0.847 | 0.851 | 0.843 | 0.856 | 0.853 | 0.841 | 0.846 | 0.849 | 0.857 | 0.847 | 0.846 | 0.849 | 0.846 | 0.847 | 0.851 |
| | 9 | 1 × I | 0.823 | 0.825 | 0.832 | 0.845 | 0.85 | 0.841 | 0.844 | 0.854 | 0.847 | 0.846 | 0.841 | 0.851 | 0.839 | 0.849 | 0.84 | 0.837 |
| | | 2 × I | 0.817 | 0.824 | 0.839 | 0.829 | 0.846 | 0.837 | 0.841 | 0.845 | 0.847 | 0.858 | 0.844 | 0.853 | 0.846 | 0.843 | 0.845 | 0.852 |
| | | 3 × I | 0.803 | 0.829 | 0.836 | 0.848 | 0.835 | 0.84 | 0.83 | 0.835 | 0.855 | 0.841 | 0.839 | 0.842 | 0.85 | 0.839 | 0.837 | 0.849 |
| | | 4 × I | 0.782 | 0.818 | 0.834 | 0.842 | 0.847 | 0.825 | 0.841 | 0.831 | 0.839 | 0.845 | 0.845 | 0.836 | 0.843 | 0.84 | 0.85 | 0.854 |
| | | 5 × I | 0.798 | 0.814 | 0.833 | 0.835 | 0.835 | **0.859** | 0.843 | 0.845 | 0.853 | 0.839 | 0.835 | 0.834 | 0.837 | 0.847 | 0.844 | 0.843 |
| | 11 | 1 × I | 0.824 | 0.828 | 0.835 | 0.835 | 0.837 | 0.833 | 0.842 | 0.841 | 0.834 | 0.854 | 0.85 | 0.844 | 0.847 | **0.861** | 0.841 | 0.852 |
| | | 2 × I | 0.814 | 0.833 | 0.847 | 0.843 | 0.84 | 0.84 | 0.84 | 0.842 | 0.852 | 0.85 | 0.853 | 0.841 | 0.847 | 0.848 | 0.847 | 0.846 |
| | | 3 × I | 0.819 | 0.827 | 0.84 | 0.836 | 0.846 | 0.831 | 0.841 | 0.841 | 0.852 | 0.834 | 0.851 | 0.841 | 0.844 | 0.847 | 0.85 | 0.85 |
| | | 4 × I | 0.816 | 0.814 | 0.83 | 0.824 | 0.842 | 0.842 | 0.847 | 0.848 | 0.84 | 0.836 | 0.843 | 0.849 | 0.85 | 0.846 | 0.85 | 0.844 |
| | | 5 × I | 0.804 | 0.822 | 0.814 | 0.83 | 0.829 | 0.843 | 0.851 | 0.843 | 0.848 | 0.841 | 0.847 | 0.838 | 0.858 | 0.848 | 0.847 | 0.85 |
| Crowd Sourced balanced | 3 | 1 × I | 0.609 | 0.689 | 0.671 | 0.732 | 0.76 | 0.762 | 0.768 | 0.779 | 0.772 | 0.837 | 0.799 | 0.755 | 0.806 | 0.821 | 0.733 | 0.803 |
| | | 2 × I | 0.683 | 0.649 | 0.734 | 0.753 | 0.737 | 0.746 | 0.787 | 0.759 | 0.78 | 0.793 | 0.807 | 0.81 | **0.858** | 0.786 | 0.819 | 0.803 |
| | | 3 × I | 0.639 | 0.68 | 0.667 | 0.696 | 0.713 | 0.804 | 0.796 | 0.806 | 0.781 | 0.807 | 0.788 | 0.805 | 0.828 | 0.763 | 0.831 | 0.854 |
| | | 4 × I | 0.573 | 0.602 | 0.732 | 0.701 | 0.739 | 0.687 | 0.767 | 0.596 | 0.816 | 0.788 | 0.765 | 0.829 | 0.824 | 0.82 | 0.791 | 0.806 |
| | | 5 × I | 0.662 | 0.727 | 0.653 | 0.74 | 0.766 | 0.733 | 0.761 | 0.76 | 0.804 | 0.812 | 0.727 | 0.757 | 0.833 | 0.778 | 0.807 | 0.851 |
| | 5 | 1 × I | 0.711 | 0.824 | 0.839 | 0.854 | 0.851 | 0.891 | 0.87 | 0.899 | 0.895 | 0.875 | 0.885 | 0.911 | 0.901 | 0.908 | 0.905 | **0.919** |
| | | 2 × I | 0.611 | 0.832 | 0.83 | 0.853 | 0.875 | 0.892 | 0.893 | 0.888 | 0.905 | 0.905 | 0.867 | 0.906 | 0.905 | 0.899 | 0.91 | 0.909 |
| | | 3 × I | 0.721 | 0.822 | 0.817 | 0.874 | 0.866 | 0.899 | 0.898 | 0.898 | 0.89 | 0.907 | 0.893 | 0.895 | 0.911 | 0.909 | 0.907 | 0.907 |
| | | 4 × I | 0.736 | 0.858 | 0.868 | 0.854 | 0.89 | 0.903 | 0.883 | 0.904 | 0.892 | 0.866 | 0.91 | 0.911 | 0.896 | 0.883 | 0.915 | 0.915 |
| | | 5 × I | 0.763 | 0.856 | 0.875 | 0.881 | 0.896 | 0.886 | 0.897 | 0.889 | 0.907 | 0.891 | 0.898 | 0.898 | 0.91 | 0.915 | 0.899 | 0.91 |
| | 7 | 1 × I | 0.809 | 0.864 | 0.879 | 0.888 | 0.899 | 0.902 | 0.887 | 0.907 | 0.916 | 0.911 | 0.917 | 0.906 | 0.917 | 0.913 | 0.916 | 0.91 |
| | | 2 × I | 0.825 | 0.882 | 0.876 | 0.891 | 0.894 | 0.892 | 0.896 | 0.912 | 0.91 | 0.912 | 0.913 | 0.91 | 0.902 | 0.917 | 0.917 | 0.915 |
| | | 3 × I | 0.823 | 0.872 | 0.868 | 0.889 | 0.911 | 0.909 | 0.901 | 0.911 | 0.911 | 0.913 | 0.908 | 0.913 | 0.904 | 0.908 | 0.917 | 0.914 |
| | | 4 × I | 0.832 | 0.853 | 0.888 | 0.895 | 0.909 | 0.907 | 0.908 | 0.916 | 0.913 | 0.912 | 0.902 | 0.913 | 0.91 | 0.916 | 0.916 | 0.914 |
| | | 5 × I | 0.843 | 0.877 | 0.894 | 0.903 | 0.906 | 0.901 | 0.903 | 0.906 | 0.9 | 0.907 | 0.891 | 0.911 | 0.915 | **0.918** | 0.914 | 0.912 |
| | 9 | 1 × I | 0.785 | 0.848 | 0.874 | 0.897 | 0.899 | 0.905 | 0.906 | 0.904 | 0.907 | 0.909 | 0.902 | 0.912 | 0.909 | 0.912 | 0.913 | 0.914 |
| | | 2 × I | 0.837 | 0.863 | 0.893 | 0.898 | 0.903 | 0.901 | 0.91 | 0.913 | 0.906 | 0.906 | 0.897 | 0.912 | 0.91 | 0.9 | 0.902 | **0.916** |
| | | 3 × I | 0.826 | 0.854 | 0.892 | 0.883 | 0.902 | 0.891 | 0.89 | 0.897 | 0.887 | 0.9 | 0.891 | 0.902 | 0.898 | 0.901 | 0.889 | 0.914 |
| | | 4 × I | 0.809 | 0.848 | 0.887 | 0.894 | 0.903 | 0.892 | 0.909 | 0.889 | 0.897 | 0.898 | 0.885 | 0.899 | 0.898 | 0.907 | 0.904 | 0.901 |
| | | 5 × I | 0.839 | 0.852 | 0.876 | 0.902 | 0.884 | 0.903 | 0.893 | 0.891 | 0.901 | 0.899 | 0.91 | 0.905 | 0.909 | 0.886 | 0.903 | 0.895 |
| | 11 | 1 × I | 0.819 | 0.845 | 0.885 | 0.893 | 0.875 | 0.91 | 0.903 | 0.908 | 0.898 | 0.909 | 0.905 | 0.909 | 0.908 | 0.913 | 0.909 | 0.915 |
| | | 2 × I | 0.826 | 0.86 | 0.853 | 0.899 | 0.889 | 0.901 | 0.89 | 0.919 | 0.915 | 0.908 | 0.913 | **0.923** | 0.915 | 0.897 | 0.91 | 0.906 |
| | | 3 × I | 0.834 | 0.85 | 0.887 | 0.887 | 0.891 | 0.877 | 0.907 | 0.901 | 0.897 | 0.908 | 0.891 | 0.906 | 0.905 | 0.918 | 0.919 | 0.921 |
| | | 4 × I | 0.837 | 0.84 | 0.873 | 0.866 | 0.895 | 0.875 | 0.907 | 0.892 | 0.893 | 0.903 | 0.911 | 0.911 | 0.89 | 0.917 | 0.916 | 0.907 |
| | | 5 × I | 0.822 | 0.85 | 0.84 | 0.871 | 0.885 | 0.879 | 0.891 | 0.889 | 0.903 | 0.913 | 0.911 | 0.909 | 0.9 | 0.909 | 0.908 | 0.912 |

**Table 14. Results of classification accuracy *acc* for the proposed approach and the Vehicle data sets: Two hidden layers, the method to substitute values of missing attributes in local tables—One artificial objects generated based on one original object, MLP networks aggregation using sum of weights and various number of neurons in the hidden layer (1AO-2HL-SUM).** Designation I is used for the number of neurons in the input layer.

| Data set | No. tables | Second HL | No. of neurons in hidden layer | | | | | | | | | | | | | | | |
|---|---|---|---|---|---|---|---|---|---|---|---|---|---|---|---|---|---|---|
| | | | 0.25 × I | 0.5 × I | 0.75 × I | 1 × I | 1.5 × I | 1.75 × I | 2 × I | 2.5 × I | 2.75 × I | 3 × I | 3.5 × I | 3.75 × I | 4 × I | 4.5 × I | 4.75 × I | 5 × I |
| Vehicle imbalanced | 3 | 1 × I | 0.283 | 0.42 | 0.521 | 0.556 | 0.629 | 0.651 | 0.673 | 0.654 | 0.68 | 0.661 | 0.636 | 0.664 | 0.652 | 0.668 | 0.575 | 0.285 |
| | | 2 × I | 0.53 | 0.604 | 0.663 | 0.657 | 0.643 | 0.659 | 0.638 | 0.663 | 0.647 | 0.684 | 0.601 | 0.69 | 0.699 | 0.602 | 0.635 | 0.657 |
| | | 3 × I | 0.475 | 0.684 | 0.49 | 0.663 | 0.689 | 0.655 | 0.671 | 0.671 | 0.676 | 0.688 | 0.652 | 0.673 | 0.675 | 0.664 | 0.638 | 0.656 |
| | | 4 × I | 0.411 | 0.664 | 0.63 | 0.639 | 0.685 | 0.698 | 0.654 | 0.676 | 0.694 | 0.647 | 0.678 | 0.699 | 0.681 | 0.698 | 0.676 | 0.686 |
| | | 5 × I | 0.283 | 0.64 | 0.6 | 0.702 | 0.643 | 0.657 | 0.707 | 0.656 | 0.66 | 0.702 | 0.651 | **0.713** | **0.713** | 0.664 | 0.696 | 0.661 |
| | 5 | 1 × I | 0.558 | 0.598 | 0.636 | 0.631 | 0.593 | 0.673 | 0.647 | 0.65 | 0.635 | 0.626 | 0.385 | 0.648 | 0.631 | 0.663 | 0.318 | 0.642 |
| | | 2 × I | 0.28 | 0.621 | 0.622 | 0.642 | 0.681 | 0.654 | 0.652 | 0.675 | 0.664 | 0.677 | 0.673 | 0.684 | 0.68 | 0.676 | 0.661 | 0.663 |
| | | 3 × I | 0.597 | 0.609 | 0.66 | 0.672 | 0.682 | 0.673 | 0.648 | 0.688 | 0.673 | 0.652 | 0.678 | 0.665 | 0.664 | 0.686 | 0.684 | 0.672 |
| | | 4 × I | 0.467 | 0.614 | 0.647 | 0.643 | 0.709 | 0.669 | 0.705 | 0.684 | 0.677 | 0.668 | 0.672 | 0.706 | 0.686 | 0.671 | 0.702 | 0.681 |
| | | 5 × I | 0.642 | 0.631 | 0.676 | 0.654 | 0.709 | 0.682 | 0.639 | 0.657 | 0.698 | **0.715** | 0.698 | 0.677 | 0.699 | 0.656 | 0.682 | 0.669 |
| | 7 | 1 × I | 0.647 | 0.618 | 0.644 | 0.648 | 0.692 | 0.681 | 0.681 | 0.706 | 0.608 | 0.63 | 0.36 | 0.68 | 0.688 | 0.693 | 0.69 | 0.688 |
| | | 2 × I | 0.613 | 0.69 | 0.634 | 0.696 | 0.693 | 0.705 | 0.693 | 0.714 | 0.715 | 0.713 | 0.668 | 0.719 | 0.705 | 0.667 | **0.735** | 0.706 |
| | | 3 × I | 0.516 | 0.714 | 0.668 | 0.698 | 0.702 | 0.69 | 0.702 | 0.706 | 0.696 | 0.705 | 0.71 | 0.711 | 0.715 | 0.694 | 0.703 | 0.698 |
| | | 4 × I | 0.283 | 0.657 | 0.693 | 0.723 | 0.699 | 0.706 | 0.715 | 0.71 | 0.693 | 0.714 | 0.711 | 0.693 | 0.714 | 0.717 | 0.722 | 0.724 |
| | | 5 × I | 0.516 | 0.684 | 0.709 | 0.686 | 0.703 | 0.677 | 0.701 | 0.714 | 0.702 | 0.734 | 0.731 | 0.723 | 0.724 | 0.728 | 0.698 | 0.703 |
| | 9 | 1 × I | 0.591 | 0.676 | 0.612 | 0.675 | 0.731 | 0.713 | 0.68 | 0.692 | 0.667 | 0.709 | 0.623 | 0.684 | 0.719 | 0.402 | 0.698 | 0.709 |
| | | 2 × I | 0.623 | 0.69 | 0.675 | 0.718 | 0.715 | 0.696 | 0.694 | 0.752 | 0.694 | 0.705 | 0.727 | 0.706 | 0.711 | 0.732 | 0.727 | 0.718 |
| | | 3 × I | 0.446 | 0.618 | 0.69 | 0.718 | 0.71 | 0.728 | 0.72 | 0.71 | 0.74 | 0.689 | 0.739 | 0.709 | 0.735 | 0.73 | 0.715 | 0.719 |
| | | 4 × I | 0.56 | 0.688 | 0.72 | 0.732 | 0.693 | 0.697 | 0.728 | 0.747 | 0.709 | 0.738 | 0.718 | 0.732 | 0.734 | 0.724 | 0.728 | 0.735 |
| | | 5 × I | 0.634 | 0.663 | 0.702 | 0.711 | 0.743 | 0.734 | 0.738 | 0.734 | 0.734 | 0.723 | 0.735 | 0.735 | 0.697 | 0.748 | **0.752** | 0.722 |
| | 11 | 1 × I | 0.596 | 0.652 | 0.636 | 0.69 | 0.69 | 0.699 | 0.69 | 0.698 | 0.651 | 0.648 | 0.697 | 0.714 | 0.709 | 0.656 | 0.723 | 0.681 |
| | | 2 × I | 0.283 | 0.634 | 0.685 | 0.698 | 0.699 | 0.703 | 0.709 | 0.696 | 0.717 | 0.727 | 0.705 | 0.717 | 0.684 | 0.701 | 0.728 | 0.693 |
| | | 3 × I | 0.564 | 0.594 | 0.604 | 0.71 | 0.686 | 0.677 | 0.698 | 0.739 | 0.689 | 0.706 | 0.734 | 0.72 | 0.717 | 0.717 | 0.713 | 0.697 |
| | | 4 × I | 0.605 | 0.693 | 0.678 | 0.701 | 0.705 | 0.71 | 0.717 | 0.723 | 0.732 | 0.726 | 0.689 | 0.724 | 0.715 | 0.701 | **0.743** | 0.715 |
| | | 5 × I | 0.576 | 0.623 | 0.714 | 0.715 | 0.715 | 0.723 | 0.738 | 0.73 | 0.727 | 0.714 | 0.732 | 0.706 | 0.719 | 0.69 | 0.715 | 0.732 |
| Vehicle blanced | 3 | 1 × I | 0.533 | 0.562 | 0.587 | 0.643 | 0.631 | 0.623 | 0.657 | 0.64 | 0.722 | 0.699 | 0.638 | 0.692 | 0.592 | 0.678 | 0.692 | 0.69 |
| | | 2 × I | 0.614 | 0.555 | 0.655 | 0.672 | 0.677 | 0.661 | 0.669 | 0.681 | 0.705 | 0.638 | 0.663 | 0.689 | 0.659 | 0.701 | 0.688 | 0.68 |
| | | 3 × I | 0.518 | 0.664 | 0.688 | 0.643 | 0.676 | 0.677 | 0.682 | 0.705 | 0.719 | 0.702 | 0.693 | 0.665 | 0.681 | 0.699 | 0.693 | 0.646 |
| | | 4 × I | 0.57 | 0.479 | 0.676 | 0.706 | 0.686 | 0.652 | 0.665 | 0.688 | 0.693 | 0.707 | 0.685 | 0.672 | 0.719 | 0.697 | 0.696 | 0.693 |
| | | 5 × I | 0.547 | 0.671 | 0.642 | 0.699 | 0.659 | 0.696 | 0.714 | 0.677 | 0.696 | **0.726** | 0.678 | 0.667 | 0.688 | 0.715 | 0.699 | 0.693 |
| | 5 | 1 × I | 0.636 | 0.664 | 0.651 | 0.661 | 0.688 | 0.654 | 0.69 | 0.673 | 0.698 | 0.677 | 0.671 | 0.707 | 0.694 | 0.677 | 0.694 | 0.699 |
| | | 2 × I | 0.589 | 0.672 | 0.709 | 0.659 | 0.697 | 0.713 | 0.726 | 0.707 | 0.731 | 0.705 | 0.715 | 0.718 | 0.697 | 0.713 | 0.719 | 0.682 |
| | | 3 × I | 0.52 | 0.692 | 0.675 | 0.69 | 0.706 | 0.724 | 0.703 | 0.717 | 0.692 | 0.738 | 0.711 | 0.732 | 0.735 | 0.724 | 0.706 | 0.719 |
| | | 4 × I | 0.591 | 0.692 | 0.702 | 0.701 | 0.699 | 0.688 | 0.706 | 0.702 | 0.686 | 0.743 | 0.722 | 0.719 | 0.743 | 0.726 | 0.702 | 0.71 |
| | | 5 × I | 0.283 | 0.672 | 0.696 | 0.68 | **0.748** | 0.726 | 0.717 | 0.696 | 0.724 | 0.727 | 0.722 | 0.696 | 0.706 | 0.723 | 0.719 | 0.723 |
| | 7 | 1 × I | 0.549 | 0.547 | 0.709 | 0.707 | 0.686 | 0.702 | 0.731 | 0.714 | 0.726 | 0.73 | 0.722 | 0.713 | 0.743 | 0.741 | 0.739 | 0.718 |
| | | 2 × I | 0.5 | 0.707 | 0.71 | 0.727 | 0.69 | 0.724 | 0.735 | 0.735 | 0.723 | 0.717 | 0.736 | 0.727 | 0.745 | 0.73 | 0.726 | 0.739 |
| | | 3 × I | 0.583 | 0.702 | 0.709 | 0.723 | 0.723 | 0.732 | 0.741 | 0.71 | 0.749 | 0.745 | 0.738 | 0.734 | 0.738 | 0.749 | 0.748 | 0.756 |
| | | 4 × I | 0.694 | 0.707 | 0.715 | 0.707 | 0.724 | 0.741 | 0.743 | 0.734 | 0.751 | 0.72 | 0.734 | 0.74 | 0.734 | 0.749 | 0.745 | 0.751 |
| | | 5 × I | 0.643 | 0.685 | 0.719 | 0.74 | 0.706 | 0.738 | 0.724 | 0.732 | 0.741 | 0.728 | 0.739 | 0.743 | **0.759** | 0.747 | 0.738 | 0.743 |
| | 9 | 1 × I | 0.618 | 0.579 | 0.69 | 0.656 | 0.701 | 0.682 | 0.694 | 0.705 | 0.713 | 0.723 | 0.736 | 0.676 | 0.714 | 0.731 | 0.686 | 0.731 |
| | | 2 × I | 0.619 | 0.612 | 0.684 | 0.684 | 0.697 | 0.698 | 0.713 | 0.719 | 0.722 | 0.71 | 0.734 | 0.724 | 0.713 | 0.719 | 0.73 | 0.719 |
| | | 3 × I | 0.609 | 0.689 | 0.675 | 0.689 | 0.735 | 0.72 | 0.732 | 0.717 | 0.711 | 0.724 | 0.707 | 0.731 | 0.724 | 0.719 | 0.739 | 0.731 |
| | | 4 × I | 0.577 | 0.644 | 0.692 | 0.694 | 0.732 | 0.709 | 0.743 | 0.724 | 0.738 | 0.711 | **0.745** | 0.73 | 0.726 | 0.68 | 0.724 | 0.734 |
| | | 5 × I | 0.589 | 0.56 | 0.715 | 0.713 | 0.717 | 0.731 | 0.74 | 0.718 | 0.723 | 0.73 | 0.713 | 0.728 | 0.736 | 0.707 | 0.707 | 0.705 |
| | 11 | 1 × I | 0.581 | 0.657 | 0.692 | 0.711 | 0.724 | 0.689 | 0.723 | 0.699 | 0.736 | 0.71 | 0.719 | 0.697 | 0.692 | 0.709 | 0.726 | 0.654 |
| | | 2 × I | 0.626 | 0.65 | 0.71 | 0.692 | 0.717 | 0.741 | 0.72 | 0.709 | 0.718 | 0.732 | 0.73 | 0.73 | 0.738 | 0.715 | 0.751 | 0.719 |
| | | 3 × I | 0.556 | 0.686 | 0.726 | 0.73 | 0.723 | 0.705 | 0.743 | 0.735 | 0.734 | 0.759 | 0.761 | 0.726 | 0.731 | 0.724 | 0.701 | 0.759 |
| | | 4 × I | 0.618 | 0.703 | 0.702 | 0.751 | 0.709 | 0.724 | 0.752 | 0.745 | 0.757 | 0.734 | 0.757 | 0.755 | **0.768** | 0.723 | 0.738 | 0.761 |
| | | 5 × I | 0.528 | 0.672 | 0.71 | 0.734 | 0.751 | 0.747 | 0.73 | 0.72 | 0.756 | 0.751 | 0.741 | 0.751 | 0.738 | **0.768** | 0.749 | 0.739 |

**Table 15. Results of classification accuracy *acc* for the proposed approach and the Dry Bean data sets: Two hidden layers, the method to substitute values of missing attributes in local tables—One artificial objects generated based on one original object, MLP networks aggregation using sum of weights and various number of neurons in the hidden layer (1AO-2HL-SUM).** Designation I is used for the number of neurons in the input layer.

| Data set | No. tables | Second HL | No. of neurons in the first hidden layer | | | | | | | | | | | | | | | |
|---|---|---|---|---|---|---|---|---|---|---|---|---|---|---|---|---|---|---|
| | | | 0.25 × I | 0.5 × I | 0.75 × I | 1 × I | 1.5 × I | 1.75 × I | 2 × I | 2.5 × I | 2.75 × I | 3 × I | 3.5 × I | 3.75 × I | 4 × I | 4.5 × I | 4.75 × I | 5 × I |
| Dry Bean imbalanced | 3 | 1 × I | 0.883 | 0.895 | 0.901 | 0.895 | 0.898 | 0.906 | 0.901 | 0.909 | 0.905 | 0.901 | 0.904 | 0.913 | 0.907 | 0.912 | 0.907 | 0.917 |
| | | 2 × I | 0.905 | 0.89 | 0.909 | 0.902 | 0.909 | 0.916 | 0.913 | 0.915 | 0.91 | 0.913 | 0.911 | 0.913 | 0.918 | 0.915 | 0.915 | 0.912 |
| | | 3 × I | 0.901 | 0.899 | 0.908 | 0.907 | 0.906 | 0.908 | 0.909 | 0.913 | 0.917 | 0.913 | 0.913 | 0.918 | 0.914 | 0.915 | 0.914 | 0.916 |
| | | 4 × I | 0.891 | 0.903 | 0.907 | 0.914 | 0.903 | 0.91 | 0.91 | 0.915 | 0.914 | 0.918 | **0.92** | 0.917 | 0.915 | 0.918 | 0.918 | 0.92 |
| | | 5 × I | 0.905 | 0.912 | 0.908 | 0.899 | 0.912 | 0.915 | 0.919 | 0.916 | 0.915 | 0.915 | **0.92** | 0.918 | 0.919 | 0.917 | 0.918 | 0.916 |
| | 5 | 1 × I | 0.887 | 0.887 | 0.888 | 0.895 | 0.898 | 0.897 | 0.902 | 0.907 | 0.901 | 0.896 | 0.908 | 0.906 | 0.906 | 0.91 | 0.909 | 0.909 |
| | | 2 × I | 0.887 | 0.893 | 0.903 | 0.904 | 0.902 | 0.898 | 0.905 | 0.903 | 0.906 | 0.908 | 0.913 | 0.912 | 0.914 | 0.914 | 0.91 | 0.913 |
| | | 3 × I | 0.892 | 0.889 | 0.898 | 0.903 | 0.91 | 0.912 | 0.91 | 0.913 | 0.909 | 0.908 | 0.913 | 0.911 | 0.915 | 0.912 | 0.913 | 0.916 |
| | | 4 × I | 0.881 | 0.898 | 0.898 | 0.905 | 0.908 | 0.909 | 0.91 | 0.911 | 0.912 | 0.913 | 0.912 | 0.915 | 0.913 | 0.914 | 0.914 | **0.918** |
| | | 5 × I | 0.899 | 0.899 | 0.901 | 0.901 | 0.91 | 0.914 | 0.907 | 0.913 | 0.915 | 0.913 | 0.911 | 0.916 | 0.913 | 0.914 | 0.917 | 0.916 |
| | 7 | 1 × I | 0.895 | 0.894 | 0.893 | 0.896 | 0.901 | 0.897 | 0.895 | 0.899 | 0.91 | 0.903 | 0.907 | 0.904 | 0.899 | 0.902 | 0.902 | 0.914 |
| | | 2 × I | 0.889 | 0.899 | 0.896 | 0.903 | 0.902 | 0.902 | 0.905 | 0.904 | 0.899 | 0.907 | 0.904 | 0.911 | 0.905 | 0.908 | 0.904 | 0.91 |
| | | 3 × I | 0.894 | 0.895 | 0.895 | 0.897 | 0.905 | 0.91 | 0.911 | 0.913 | 0.909 | 0.911 | 0.91 | 0.913 | 0.911 | 0.912 | 0.914 | 0.915 |
| | | 4 × I | 0.894 | 0.893 | 0.9 | 0.903 | 0.902 | 0.907 | 0.905 | 0.912 | 0.913 | 0.911 | 0.913 | 0.914 | 0.912 | 0.914 | 0.913 | 0.915 |
| | | 5 × I | 0.887 | 0.898 | 0.897 | 0.909 | 0.914 | 0.909 | 0.905 | 0.91 | 0.911 | 0.909 | 0.911 | 0.916 | 0.916 | 0.916 | 0.914 | **0.917** |
| | 9 | 1 × I | 0.885 | 0.892 | 0.889 | 0.889 | 0.894 | 0.894 | 0.898 | 0.898 | 0.892 | 0.898 | 0.902 | 0.895 | 0.899 | 0.896 | 0.9 | 0.901 |
| | | 2 × I | 0.888 | 0.893 | 0.895 | 0.899 | 0.898 | 0.895 | 0.9 | 0.898 | 0.901 | 0.901 | 0.903 | 0.899 | 0.903 | 0.901 | 0.908 | 0.909 |
| | | 3 × I | 0.892 | 0.897 | 0.897 | 0.899 | 0.897 | 0.908 | 0.901 | 0.904 | 0.903 | 0.908 | 0.905 | 0.908 | 0.905 | 0.909 | 0.906 | 0.911 |
| | | 4 × I | 0.89 | 0.896 | 0.895 | 0.9 | 0.899 | 0.901 | 0.902 | 0.902 | 0.905 | 0.905 | 0.907 | 0.906 | 0.912 | 0.906 | 0.912 | 0.907 |
| | | 5 × I | 0.893 | 0.896 | 0.899 | 0.895 | 0.903 | 0.898 | 0.905 | 0.904 | 0.909 | 0.911 | 0.909 | **0.915** | 0.912 | 0.911 | 0.914 | 0.913 |
| | 11 | 1 × I | 0.897 | 0.888 | 0.892 | 0.891 | 0.894 | 0.893 | 0.894 | 0.895 | 0.898 | 0.895 | 0.897 | 0.9 | 0.897 | 0.898 | 0.898 | 0.904 |
| | | 2 × I | 0.888 | 0.897 | 0.896 | 0.891 | 0.896 | 0.899 | 0.898 | 0.896 | 0.902 | 0.901 | 0.896 | 0.899 | 0.9 | 0.906 | 0.898 | 0.904 |
| | | 3 × I | 0.885 | 0.894 | 0.892 | 0.9 | 0.897 | 0.899 | 0.901 | 0.899 | 0.899 | 0.907 | 0.905 | 0.903 | 0.899 | 0.899 | 0.906 | 0.903 |
| | | 4 × I | 0.889 | 0.891 | 0.896 | 0.899 | 0.898 | 0.9 | 0.901 | 0.905 | 0.902 | 0.903 | 0.907 | 0.902 | 0.905 | 0.898 | 0.907 | 0.906 |
| | | 5 × I | 0.89 | 0.895 | 0.895 | 0.893 | 0.897 | 0.903 | 0.901 | 0.902 | 0.906 | 0.902 | 0.906 | 0.902 | 0.909 | **0.914** | 0.908 | 0.907 |
| Dry Bean blanced | 3 | 1 × I | 0.889 | 0.888 | 0.899 | 0.908 | 0.908 | 0.905 | 0.901 | 0.912 | 0.913 | 0.907 | 0.912 | 0.91 | 0.911 | 0.912 | 0.914 | 0.912 |
| | | 2 × I | 0.884 | 0.899 | 0.902 | 0.907 | 0.911 | 0.912 | 0.911 | 0.907 | 0.911 | 0.915 | 0.906 | 0.917 | 0.916 | 0.916 | 0.916 | 0.918 |
| | | 3 × I | 0.871 | 0.893 | 0.909 | 0.901 | 0.91 | 0.909 | 0.908 | 0.914 | 0.917 | 0.914 | 0.91 | 0.913 | 0.914 | 0.916 | 0.916 | 0.916 |
| | | 4 × I | 0.902 | 0.909 | 0.913 | 0.913 | 0.915 | 0.915 | 0.912 | 0.917 | 0.915 | 0.915 | 0.914 | 0.916 | 0.917 | 0.917 | 0.917 | 0.918 |
| | | 5 × I | 0.891 | 0.91 | 0.915 | 0.907 | 0.916 | 0.916 | 0.913 | 0.919 | 0.917 | 0.916 | 0.916 | 0.915 | 0.917 | 0.917 | **0.919** | 0.916 |
| | 5 | 1 × I | 0.889 | 0.888 | 0.897 | 0.905 | 0.903 | 0.909 | 0.903 | 0.912 | 0.902 | 0.908 | 0.91 | 0.906 | 0.91 | 0.907 | 0.913 | 0.911 |
| | | 2 × I | 0.893 | 0.896 | 0.898 | 0.905 | 0.905 | 0.91 | 0.911 | 0.91 | 0.913 | 0.908 | 0.915 | 0.914 | 0.909 | 0.915 | 0.913 | 0.916 |
| | | 3 × I | 0.909 | 0.903 | 0.902 | 0.902 | 0.91 | 0.909 | 0.912 | 0.914 | 0.914 | **0.917** | 0.911 | 0.913 | 0.915 | **0.917** | 0.912 | 0.916 |
| | | 4 × I | 0.887 | 0.893 | 0.902 | 0.904 | 0.908 | 0.912 | 0.911 | 0.914 | 0.914 | 0.913 | 0.916 | 0.914 | 0.915 | 0.914 | 0.916 | 0.914 |
| | | 5 × I | 0.892 | 0.893 | 0.909 | 0.906 | 0.915 | 0.912 | 0.915 | 0.914 | **0.917** | 0.915 | 0.915 | **0.917** | **0.917** | 0.915 | **0.917** | 0.916 |
| | 7 | 1 × I | 0.887 | 0.895 | 0.895 | 0.889 | 0.899 | 0.895 | 0.897 | 0.898 | 0.904 | 0.901 | 0.903 | 0.904 | 0.904 | 0.911 | 0.904 | 0.905 |
| | | 2 × I | 0.893 | 0.892 | 0.893 | 0.899 | 0.908 | 0.899 | 0.905 | 0.91 | 0.901 | 0.911 | 0.906 | 0.911 | 0.91 | 0.911 | 0.909 | 0.907 |
| | | 3 × I | 0.894 | 0.892 | 0.893 | 0.904 | 0.902 | 0.906 | 0.909 | 0.905 | 0.908 | 0.913 | 0.911 | 0.908 | 0.914 | 0.914 | 0.912 | 0.912 |
| | | 4 × I | 0.889 | 0.897 | 0.903 | 0.899 | 0.905 | 0.912 | 0.91 | 0.909 | 0.912 | 0.913 | 0.915 | 0.914 | 0.912 | 0.918 | **0.919** | 0.915 |
| | | 5 × I | 0.898 | 0.895 | 0.905 | 0.91 | 0.91 | 0.908 | 0.911 | 0.914 | 0.907 | 0.915 | 0.913 | 0.913 | 0.917 | 0.912 | **0.919** | 0.915 |
| | 9 | 1 × I | 0.885 | 0.886 | 0.893 | 0.896 | 0.901 | 0.896 | 0.898 | 0.9 | 0.899 | 0.9 | 0.905 | 0.907 | 0.904 | 0.911 | 0.9 | 0.905 |
| | | 2 × I | 0.887 | 0.891 | 0.892 | 0.9 | 0.9 | 0.897 | 0.899 | 0.897 | 0.906 | 0.901 | 0.908 | 0.91 | 0.901 | 0.907 | 0.907 | 0.909 |
| | | 3 × I | 0.894 | 0.893 | 0.895 | 0.903 | 0.902 | 0.9 | 0.904 | 0.899 | 0.908 | 0.906 | 0.915 | 0.913 | 0.906 | 0.913 | 0.913 | 0.911 |
| | | 4 × I | 0.893 | 0.896 | 0.892 | 0.907 | 0.905 | 0.906 | 0.905 | 0.905 | 0.907 | 0.904 | 0.906 | 0.913 | 0.909 | 0.915 | 0.914 | 0.916 |
| | | 5 × I | 0.89 | 0.896 | 0.895 | 0.904 | 0.906 | 0.91 | 0.908 | 0.909 | 0.913 | 0.916 | **0.917** | 0.914 | 0.912 | 0.913 | 0.914 | 0.915 |
| | 11 | 1 × I | 0.888 | 0.891 | 0.894 | 0.895 | 0.89 | 0.894 | 0.897 | 0.902 | 0.9 | 0.896 | 0.9 | 0.901 | 0.9 | 0.908 | 0.903 | 0.899 |
| | | 2 × I | 0.888 | 0.886 | 0.893 | 0.896 | 0.896 | 0.894 | 0.898 | 0.908 | 0.903 | 0.914 | 0.9 | 0.903 | 0.908 | 0.91 | 0.907 | 0.905 |
| | | 3 × I | 0.893 | 0.896 | 0.899 | 0.897 | 0.899 | 0.897 | 0.902 | 0.908 | 0.899 | 0.904 | 0.911 | 0.908 | 0.908 | 0.913 | 0.908 | 0.909 |
| | | 4 × I | 0.89 | 0.893 | 0.893 | 0.897 | 0.905 | 0.902 | 0.905 | 0.907 | 0.905 | 0.907 | 0.912 | 0.914 | 0.913 | **0.917** | 0.912 | 0.915 |
| | | 5 × I | 0.888 | 0.897 | 0.904 | 0.899 | 0.901 | 0.906 | 0.913 | 0.906 | 0.913 | 0.91 | 0.912 | **0.917** | 0.915 | 0.914 | 0.915 | 0.915 |

**Table 16. Results of classification accuracy *acc* for the proposed approach and the Sensorless data sets: Two hidden layers, the method to substitute values of missing attributes in local tables—One artificial objects generated based on one original object, MLP networks aggregation using sum of weights and various number of neurons in the hidden layer (1AO-2HL-SUM). Designation I is used for the number of neurons in the input layer.**

| Data set | No. tables | Second HL | No. of neurons in the first hidden layer | | | | | | | | | | | | | | | |
|---|---|---|---|---|---|---|---|---|---|---|---|---|---|---|---|---|---|---|
| | | | $0.25 \times I$ | $0.5 \times I$ | $0.75 \times I$ | $1 \times I$ | $1.5 \times I$ | $1.75 \times I$ | $2 \times I$ | $2.5 \times I$ | $2.75 \times I$ | $3 \times I$ | $3.5 \times I$ | $3.75 \times I$ | $4 \times I$ | $4.5 \times I$ | $4.75 \times I$ | $5 \times I$ |
| Sensorless imbalanced | 3 | $1 \times I$ | 0.9 | 0.91 | 0.92 | 0.912 | 0.935 | 0.942 | 0.937 | 0.942 | 0.941 | 0.943 | 0.94 | 0.937 | 0.931 | 0.932 | 0.942 | 0.938 |
| | | $2 \times I$ | 0.91 | 0.923 | 0.939 | 0.941 | 0.939 | 0.941 | 0.943 | 0.945 | 0.941 | 0.936 | 0.941 | 0.943 | 0.941 | 0.941 | 0.938 | 0.946 |
| | | $3 \times I$ | 0.937 | 0.931 | 0.939 | 0.937 | 0.939 | 0.945 | **0.949** | 0.939 | 0.938 | 0.945 | 0.94 | 0.939 | 0.937 | 0.939 | 0.935 | 0.937 |
| | | $4 \times I$ | 0.923 | 0.942 | 0.932 | 0.945 | 0.947 | 0.947 | 0.942 | 0.943 | 0.945 | 0.937 | 0.939 | 0.937 | 0.942 | 0.94 | 0.938 | 0.934 |
| | | $5 \times I$ | 0.924 | 0.94 | 0.944 | 0.939 | 0.942 | 0.942 | 0.946 | 0.943 | 0.948 | 0.946 | 0.945 | 0.941 | 0.933 | 0.934 | 0.936 | 0.934 |
| | 5 | $1 \times I$ | 0.9 | 0.912 | 0.919 | 0.92 | 0.923 | 0.93 | 0.932 | 0.927 | 0.926 | 0.928 | 0.928 | 0.93 | 0.93 | 0.925 | 0.924 | 0.924 |
| | | $2 \times I$ | 0.915 | 0.921 | 0.926 | 0.917 | 0.931 | 0.927 | 0.931 | **0.934** | 0.932 | 0.931 | 0.931 | 0.926 | 0.926 | 0.927 | 0.927 | 0.927 |
| | | $3 \times I$ | 0.916 | 0.919 | 0.927 | 0.923 | 0.928 | 0.925 | 0.932 | 0.93 | 0.925 | 0.928 | 0.931 | 0.921 | 0.922 | 0.927 | 0.926 | 0.927 |
| | | $4 \times I$ | 0.916 | 0.918 | 0.924 | 0.929 | 0.926 | 0.929 | 0.931 | 0.923 | 0.931 | 0.927 | 0.924 | 0.926 | 0.928 | 0.925 | 0.927 | 0.917 |
| | | $5 \times I$ | 0.915 | 0.926 | 0.93 | 0.928 | 0.923 | 0.931 | 0.927 | 0.925 | 0.927 | 0.925 | 0.924 | 0.916 | 0.921 | 0.923 | 0.923 | 0.918 |
| | 7 | $1 \times I$ | 0.906 | 0.922 | 0.918 | 0.926 | 0.927 | 0.932 | 0.936 | 0.935 | 0.936 | 0.933 | 0.93 | 0.933 | 0.936 | 0.934 | 0.933 | 0.933 |
| | | $2 \times I$ | 0.904 | 0.931 | 0.927 | 0.932 | 0.933 | 0.934 | 0.933 | 0.934 | 0.938 | 0.938 | 0.932 | 0.933 | 0.931 | 0.935 | 0.929 | 0.936 |
| | | $3 \times I$ | 0.92 | 0.924 | 0.928 | 0.931 | 0.939 | 0.928 | 0.939 | 0.938 | 0.929 | 0.936 | 0.936 | 0.934 | 0.93 | 0.928 | 0.93 | 0.93 |
| | | $4 \times I$ | 0.922 | 0.933 | 0.931 | 0.937 | 0.939 | 0.93 | 0.939 | 0.934 | 0.933 | 0.935 | 0.93 | 0.935 | 0.931 | 0.929 | 0.934 | 0.929 |
| | | $5 \times I$ | 0.922 | 0.93 | 0.936 | 0.933 | **0.942** | 0.937 | 0.927 | 0.933 | 0.932 | 0.936 | 0.933 | 0.931 | 0.93 | 0.933 | 0.927 | 0.933 |
| | 9 | $1 \times I$ | 0.91 | 0.913 | 0.923 | 0.922 | 0.921 | 0.931 | 0.929 | 0.934 | 0.93 | 0.932 | 0.932 | 0.935 | 0.93 | 0.932 | 0.93 | 0.93 |
| | | $2 \times I$ | 0.909 | 0.923 | 0.923 | 0.936 | 0.933 | 0.934 | 0.936 | 0.935 | 0.932 | 0.932 | 0.936 | 0.93 | 0.932 | 0.935 | 0.935 | 0.937 |
| | | $3 \times I$ | 0.917 | 0.931 | 0.928 | 0.934 | 0.938 | 0.936 | 0.934 | 0.937 | 0.929 | 0.934 | 0.934 | 0.934 | 0.936 | 0.931 | 0.93 | 0.936 |
| | | $4 \times I$ | 0.927 | 0.932 | 0.932 | 0.931 | 0.933 | 0.934 | **0.942** | 0.93 | 0.937 | 0.929 | 0.928 | 0.934 | 0.935 | 0.936 | 0.93 | 0.934 |
| | | $5 \times I$ | 0.92 | 0.93 | 0.93 | 0.932 | 0.933 | 0.934 | 0.934 | 0.928 | 0.935 | 0.937 | 0.935 | 0.932 | 0.926 | 0.935 | 0.924 | 0.932 |
| | 11 | $1 \times I$ | 0.908 | 0.911 | 0.923 | 0.923 | 0.928 | 0.93 | 0.926 | 0.933 | 0.929 | 0.934 | 0.935 | 0.936 | 0.929 | 0.928 | 0.926 | 0.931 |
| | | $2 \times I$ | 0.904 | 0.917 | 0.924 | 0.934 | 0.928 | 0.935 | 0.933 | 0.94 | 0.932 | 0.932 | 0.937 | 0.933 | 0.937 | 0.936 | 0.937 | 0.931 |
| | | $3 \times I$ | 0.908 | 0.931 | 0.927 | 0.933 | **0.941** | 0.933 | 0.929 | 0.933 | 0.936 | 0.939 | 0.93 | 0.932 | 0.935 | 0.934 | 0.933 | 0.937 |
| | | $4 \times I$ | 0.92 | 0.927 | 0.932 | 0.935 | 0.936 | 0.936 | 0.936 | 0.936 | 0.936 | 0.929 | 0.94 | 0.937 | 0.938 | 0.935 | 0.935 | 0.934 |
| | | $5 \times I$ | 0.918 | 0.929 | 0.931 | 0.935 | 0.934 | 0.938 | 0.937 | 0.932 | 0.935 | 0.933 | 0.931 | 0.938 | 0.928 | 0.93 | 0.93 | 0.931 |

**Table 17. Results of classification accuracy *acc* for the proposed approach and the Crowd Sourced data sets: Two hidden layers, the method to substitute values of missing attributes in local tables—One artificial objects generated based on one original object, MLP networks aggregation using sum of weights and various number of neurons in the hidden layer (1AO-2HL-SUM).** Designation I is used for the number of neurons in the input layer.

| Data set | No. tables | Second HL | No. of neurons in the first hidden layer | | | | | | | | | | | | | | | |
|---|---|---|---|---|---|---|---|---|---|---|---|---|---|---|---|---|---|---|
| | | | 0.25 × I | 0.5 × I | 0.75 × I | 1 × I | 1.5 × I | 1.75 × I | 2 × I | 2.5 × I | 2.75 × I | 3 × I | 3.5 × I | 3.75 × I | 4 × I | 4.5 × I | 4.75 × I | 5 × I |
| Crowd Sourced imbalanced | 3 | 1 × I | 0.646 | 0.713 | 0.734 | 0.777 | 0.759 | 0.775 | 0.785 | 0.804 | 0.78 | 0.792 | 0.795 | 0.794 | 0.8 | 0.806 | 0.82 | 0.808 |
| | | 2 × I | 0.699 | 0.762 | 0.767 | 0.754 | 0.794 | 0.808 | 0.785 | 0.812 | 0.81 | 0.789 | 0.813 | 0.812 | 0.802 | 0.802 | 0.817 | **0.838** |
| | | 3 × I | 0.657 | 0.745 | 0.785 | 0.757 | 0.801 | 0.777 | 0.776 | 0.806 | 0.779 | 0.802 | 0.811 | 0.799 | 0.798 | 0.801 | 0.817 | 0.818 |
| | | 4 × I | 0.709 | 0.765 | 0.796 | 0.754 | 0.779 | 0.78 | 0.787 | 0.796 | 0.799 | 0.806 | 0.811 | 0.804 | 0.79 | 0.812 | 0.817 | 0.801 |
| | | 5 × I | 0.674 | 0.774 | 0.781 | 0.773 | 0.788 | 0.771 | 0.806 | 0.809 | 0.815 | 0.782 | 0.812 | 0.813 | 0.813 | 0.806 | 0.828 | 0.815 |
| | 5 | 1 × I | 0.627 | 0.763 | 0.75 | 0.748 | 0.777 | 0.791 | 0.79 | 0.801 | 0.814 | 0.776 | 0.819 | 0.818 | 0.832 | 0.794 | 0.829 | 0.798 |
| | | 2 × I | 0.659 | 0.748 | 0.779 | 0.783 | 0.753 | 0.79 | 0.808 | 0.826 | 0.792 | 0.814 | 0.822 | 0.827 | 0.806 | 0.821 | 0.84 | 0.819 |
| | | 3 × I | 0.67 | 0.753 | 0.796 | 0.778 | 0.806 | 0.816 | 0.804 | 0.823 | 0.83 | 0.806 | 0.821 | **0.845** | 0.833 | 0.796 | 0.837 | 0.821 |
| | | 4 × I | 0.628 | 0.768 | 0.781 | 0.79 | 0.811 | 0.819 | 0.814 | 0.807 | 0.832 | 0.807 | 0.835 | 0.837 | 0.824 | 0.836 | 0.838 | 0.824 |
| | | 5 × I | 0.662 | 0.736 | 0.797 | 0.809 | 0.826 | 0.823 | 0.82 | 0.825 | 0.822 | 0.835 | 0.829 | 0.843 | 0.809 | 0.816 | 0.837 | 0.822 |
| | 7 | 1 × I | 0.652 | 0.74 | 0.798 | 0.772 | 0.77 | 0.802 | 0.779 | 0.799 | 0.806 | 0.768 | 0.838 | 0.806 | 0.787 | 0.815 | 0.831 | 0.835 |
| | | 2 × I | 0.694 | 0.74 | 0.78 | 0.756 | 0.802 | 0.814 | 0.819 | 0.813 | 0.822 | 0.809 | 0.804 | 0.813 | 0.812 | 0.818 | 0.829 | 0.805 |
| | | 3 × I | 0.677 | 0.79 | 0.727 | 0.814 | 0.773 | 0.795 | 0.803 | 0.812 | 0.827 | 0.814 | 0.819 | 0.823 | 0.816 | 0.82 | 0.828 | 0.833 |
| | | 4 × I | 0.611 | 0.757 | 0.722 | 0.786 | 0.806 | 0.802 | 0.828 | 0.823 | 0.825 | 0.832 | 0.808 | 0.823 | 0.821 | **0.842** | 0.831 | 0.823 |
| | | 5 × I | 0.717 | 0.777 | 0.792 | 0.818 | 0.794 | 0.797 | 0.817 | 0.821 | 0.828 | 0.82 | 0.815 | 0.831 | 0.825 | 0.815 | 0.826 | 0.84 |
| | 9 | 1 × I | 0.723 | 0.734 | 0.773 | 0.794 | 0.794 | 0.796 | 0.788 | 0.785 | 0.813 | 0.775 | 0.818 | 0.827 | 0.801 | 0.813 | 0.828 | 0.828 |
| | | 2 × I | 0.769 | 0.756 | 0.801 | 0.803 | 0.8 | 0.822 | 0.795 | 0.8 | 0.799 | 0.791 | 0.819 | 0.821 | 0.815 | **0.839** | 0.828 | 0.835 |
| | | 3 × I | 0.647 | 0.775 | 0.779 | 0.786 | 0.811 | 0.77 | 0.82 | 0.814 | 0.823 | 0.83 | 0.829 | 0.813 | 0.826 | 0.82 | 0.824 | 0.831 |
| | | 4 × I | 0.729 | 0.788 | 0.778 | 0.788 | 0.82 | 0.803 | 0.813 | 0.792 | 0.813 | 0.818 | 0.831 | 0.822 | 0.83 | 0.822 | 0.834 | 0.823 |
| | | 5 × I | 0.735 | 0.786 | 0.796 | 0.83 | 0.809 | 0.799 | 0.823 | 0.818 | 0.824 | 0.827 | 0.828 | 0.819 | 0.837 | 0.813 | 0.825 | 0.829 |
| | 11 | 1 × I | 0.619 | 0.76 | 0.777 | 0.798 | 0.805 | 0.789 | 0.812 | 0.806 | 0.824 | 0.837 | 0.816 | 0.844 | 0.829 | 0.832 | 0.83 | 0.835 |
| | | 2 × I | 0.772 | 0.767 | 0.786 | 0.815 | 0.77 | 0.823 | 0.826 | 0.835 | 0.833 | 0.809 | 0.831 | 0.84 | 0.836 | 0.841 | 0.825 | 0.844 |
| | | 3 × I | 0.741 | 0.791 | 0.806 | 0.8 | 0.808 | 0.835 | 0.794 | 0.833 | 0.812 | 0.826 | 0.821 | 0.844 | 0.831 | **0.861** | 0.823 | 0.845 |
| | | 4 × I | 0.676 | 0.79 | 0.802 | 0.773 | 0.828 | 0.807 | 0.837 | 0.829 | 0.831 | 0.826 | 0.833 | 0.842 | 0.839 | 0.844 | 0.842 | 0.841 |
| | | 5 × I | 0.734 | 0.769 | 0.809 | 0.808 | 0.823 | 0.818 | 0.82 | 0.83 | 0.837 | 0.826 | 0.844 | 0.837 | 0.837 | 0.816 | 0.826 | 0.838 |
| Crowd Sourced balanced | 3 | 1 × I | 0.463 | 0.676 | 0.705 | 0.769 | 0.759 | 0.778 | 0.807 | 0.815 | 0.825 | 0.82 | 0.82 | 0.832 | 0.828 | 0.852 | 0.812 | 0.786 |
| | | 2 × I | 0.453 | 0.688 | 0.741 | 0.78 | 0.763 | 0.792 | 0.785 | 0.835 | 0.832 | 0.823 | 0.841 | 0.829 | 0.842 | 0.856 | 0.863 | 0.847 |
| | | 3 × I | 0.481 | 0.652 | 0.761 | 0.781 | 0.802 | 0.806 | 0.821 | 0.816 | 0.825 | 0.826 | 0.866 | 0.834 | 0.854 | 0.871 | **0.879** | 0.863 |
| | | 4 × I | 0.579 | 0.661 | 0.756 | 0.744 | 0.805 | 0.802 | 0.849 | 0.84 | 0.847 | 0.838 | 0.84 | 0.852 | 0.834 | 0.846 | 0.846 | 0.868 |
| | | 5 × I | 0.6 | 0.698 | 0.684 | 0.725 | 0.809 | 0.84 | 0.807 | 0.836 | 0.839 | 0.826 | 0.856 | 0.848 | 0.837 | 0.868 | 0.846 | 0.873 |
| | 5 | 1 × I | 0.436 | 0.703 | 0.673 | 0.751 | 0.756 | 0.742 | 0.746 | 0.823 | 0.773 | 0.86 | 0.869 | 0.858 | 0.811 | 0.868 | 0.862 | 0.835 |
| | | 2 × I | 0.521 | 0.688 | 0.771 | 0.763 | 0.82 | 0.785 | 0.84 | 0.838 | 0.838 | 0.832 | 0.873 | 0.858 | 0.854 | 0.875 | 0.864 | 0.866 |
| | | 3 × I | 0.618 | 0.751 | 0.731 | 0.817 | 0.821 | 0.808 | 0.823 | 0.838 | 0.86 | 0.87 | 0.871 | 0.882 | 0.869 | 0.863 | 0.859 | 0.872 |
| | | 4 × I | 0.636 | 0.708 | 0.739 | 0.783 | 0.817 | 0.838 | 0.849 | 0.836 | 0.848 | 0.867 | 0.872 | 0.876 | 0.887 | 0.885 | 0.875 | **0.894** |
| | | 5 × I | 0.6 | 0.743 | 0.793 | 0.787 | 0.833 | 0.869 | 0.856 | 0.872 | 0.885 | 0.867 | 0.87 | 0.883 | 0.859 | 0.884 | 0.894 | 0.89 |
| | 7 | 1 × I | 0.472 | 0.709 | 0.701 | 0.705 | 0.76 | 0.743 | 0.798 | 0.739 | 0.823 | 0.741 | 0.837 | 0.79 | 0.835 | 0.871 | 0.87 | 0.821 |
| | | 2 × I | 0.639 | 0.718 | 0.77 | 0.785 | 0.801 | 0.791 | 0.741 | 0.849 | 0.842 | 0.858 | 0.882 | 0.866 | 0.845 | 0.889 | 0.875 | 0.876 |
| | | 3 × I | 0.566 | 0.713 | 0.75 | 0.8 | 0.77 | 0.816 | 0.82 | 0.796 | 0.854 | 0.835 | 0.837 | 0.879 | 0.859 | 0.853 | 0.885 | 0.887 |
| | | 4 × I | 0.612 | 0.748 | 0.792 | 0.78 | 0.824 | 0.808 | 0.834 | 0.85 | 0.855 | 0.867 | 0.858 | 0.867 | 0.878 | 0.891 | 0.884 | **0.899** |
| | | 5 × I | 0.622 | 0.75 | 0.776 | 0.806 | 0.849 | 0.837 | 0.843 | 0.854 | 0.869 | 0.867 | 0.865 | 0.881 | 0.875 | 0.887 | 0.881 | 0.886 |
| | 9 | 1 × I | 0.62 | 0.702 | 0.724 | 0.723 | 0.779 | 0.797 | 0.751 | 0.83 | 0.82 | 0.816 | 0.832 | 0.854 | 0.86 | 0.858 | 0.853 | 0.864 |
| | | 2 × I | 0.58 | 0.675 | 0.698 | 0.731 | 0.817 | 0.76 | 0.83 | 0.83 | 0.829 | 0.854 | 0.856 | 0.869 | 0.851 | 0.874 | 0.885 | 0.871 |
| | | 3 × I | 0.495 | 0.705 | 0.789 | 0.758 | 0.823 | 0.825 | 0.809 | 0.857 | 0.853 | 0.846 | 0.853 | 0.865 | 0.877 | 0.873 | 0.859 | 0.892 |
| | | 4 × I | 0.662 | 0.686 | 0.793 | 0.789 | 0.838 | 0.833 | 0.825 | 0.84 | 0.867 | 0.843 | 0.873 | 0.845 | 0.877 | 0.873 | 0.88 | 0.883 |
| | | 5 × I | 0.608 | 0.753 | 0.782 | 0.811 | 0.827 | 0.855 | 0.818 | 0.849 | 0.869 | 0.872 | 0.878 | 0.887 | 0.881 | 0.874 | 0.873 | **0.893** |
| | 11 | 1 × I | 0.637 | 0.675 | 0.7 | 0.749 | 0.729 | 0.753 | 0.719 | 0.83 | 0.81 | 0.82 | 0.831 | 0.807 | 0.846 | 0.83 | 0.842 | 0.858 |
| | | 2 × I | 0.654 | 0.736 | 0.751 | 0.749 | 0.761 | 0.808 | 0.78 | 0.803 | 0.825 | 0.859 | 0.852 | 0.84 | 0.875 | 0.874 | 0.873 | 0.885 |
| | | 3 × I | 0.679 | 0.734 | 0.762 | 0.803 | 0.786 | 0.848 | 0.787 | 0.853 | 0.859 | 0.852 | 0.868 | 0.854 | 0.884 | 0.895 | 0.89 | 0.885 |
| | | 4 × I | 0.659 | 0.738 | 0.802 | 0.805 | 0.825 | 0.817 | 0.831 | 0.819 | 0.857 | 0.869 | 0.876 | 0.89 | 0.885 | 0.882 | 0.897 | 0.878 |
| | | 5 × I | 0.737 | 0.757 | 0.774 | 0.787 | 0.843 | 0.849 | 0.871 | 0.874 | 0.873 | 0.853 | 0.868 | 0.87 | 0.884 | 0.892 | **0.903** | 0.899 |

**Table 18. Results of classification accuracy *acc* for the proposed approach and the Vehicle data sets: Two hidden layers, the method to substitute values of missing attributes in local tables—Three artificial objects generated based on one original object, MLP networks aggregation using sum of weights and various number of neurons in the hidden layer (3AO-2HL-SUM).** Designation I is used for the number of neurons in the input layer.

| Data set | No. tables | Second HL | No. of neurons in the first hidden layer | | | | | | | | | | | | | | | |
|---|---|---|---|---|---|---|---|---|---|---|---|---|---|---|---|---|---|---|
| | | | 0.25 × I | 0.5 × I | 0.75 × I | 1 × I | 1.5 × I | 1.75 × I | 2 × I | 2.5 × I | 2.75 × I | 3 × I | 3.5 × I | 3.75 × I | 4 × I | 4.5 × I | 4.75 × I | 5 × I |
| Vehicle imbalanced | 3 | 1 × I | 0.274 | 0.546 | 0.591 | 0.584 | 0.647 | 0.636 | 0.672 | 0.631 | 0.61 | 0.644 | 0.61 | 0.636 | 0.395 | 0.63 | 0.615 | 0.665 |
| | | 2 × I | 0.461 | 0.587 | 0.6 | 0.631 | 0.663 | 0.605 | 0.635 | 0.629 | 0.664 | 0.65 | **0.713** | 0.636 | 0.656 | 0.66 | 0.644 | 0.671 |
| | | 3 × I | 0.448 | 0.644 | 0.669 | 0.677 | 0.681 | 0.684 | 0.689 | 0.64 | 0.672 | 0.69 | 0.676 | 0.682 | 0.665 | 0.684 | 0.673 | 0.686 |
| | | 4 × I | 0.491 | 0.638 | 0.655 | 0.648 | 0.681 | 0.669 | 0.699 | 0.669 | 0.701 | 0.697 | 0.684 | 0.681 | 0.673 | 0.654 | 0.68 | 0.698 |
| | | 5 × I | 0.479 | 0.605 | 0.644 | 0.642 | 0.672 | 0.682 | 0.676 | 0.681 | 0.673 | 0.694 | 0.677 | 0.689 | 0.668 | 0.693 | 0.677 | 0.694 |
| | 5 | 1 × I | 0.417 | 0.554 | 0.63 | 0.654 | 0.65 | 0.639 | 0.622 | 0.671 | 0.643 | 0.652 | 0.661 | 0.663 | 0.642 | 0.636 | 0.682 | 0.669 |
| | | 2 × I | 0.283 | 0.622 | 0.644 | 0.65 | 0.684 | 0.622 | 0.664 | 0.654 | 0.656 | 0.6 | 0.677 | 0.681 | 0.698 | 0.703 | 0.689 | 0.672 |
| | | 3 × I | 0.283 | 0.618 | 0.64 | 0.692 | 0.68 | 0.672 | 0.701 | 0.646 | 0.689 | 0.68 | 0.703 | 0.668 | 0.703 | 0.688 | 0.718 | 0.68 |
| | | 4 × I | 0.453 | 0.58 | 0.652 | 0.69 | 0.69 | 0.638 | 0.694 | 0.693 | 0.661 | 0.651 | 0.717 | 0.69 | 0.686 | 0.694 | 0.697 | 0.667 |
| | | 5 × I | 0.333 | 0.64 | 0.665 | 0.65 | 0.673 | 0.702 | 0.684 | 0.686 | **0.72** | 0.693 | 0.714 | 0.698 | 0.709 | 0.682 | 0.696 | 0.717 |
| | 7 | 1 × I | 0.466 | 0.56 | 0.656 | 0.648 | 0.675 | 0.654 | 0.672 | 0.546 | 0.664 | 0.668 | 0.665 | 0.669 | 0.639 | 0.655 | 0.598 | 0.665 |
| | | 2 × I | 0.383 | 0.644 | 0.664 | 0.68 | 0.689 | 0.647 | 0.701 | 0.724 | 0.706 | 0.669 | 0.696 | 0.685 | 0.702 | 0.68 | 0.664 | 0.684 |
| | | 3 × I | 0.631 | 0.665 | 0.693 | 0.669 | 0.689 | 0.667 | 0.655 | 0.68 | 0.677 | 0.673 | 0.689 | 0.688 | 0.698 | 0.68 | 0.709 | 0.707 |
| | | 4 × I | 0.52 | 0.594 | 0.651 | 0.69 | 0.681 | 0.69 | 0.703 | 0.714 | 0.702 | 0.711 | 0.702 | 0.722 | 0.709 | 0.702 | 0.682 | **0.724** |
| | | 5 × I | 0.377 | 0.622 | 0.669 | 0.665 | 0.706 | 0.702 | 0.694 | 0.685 | 0.69 | 0.69 | 0.698 | 0.694 | 0.693 | 0.71 | 0.702 | 0.699 |
| | 9 | 1 × I | 0.606 | 0.416 | 0.629 | 0.671 | 0.66 | 0.675 | 0.588 | 0.591 | 0.639 | 0.654 | 0.626 | 0.644 | 0.668 | 0.667 | 0.664 | 0.655 |
| | | 2 × I | 0.524 | 0.659 | 0.63 | 0.644 | 0.663 | 0.693 | 0.65 | 0.672 | 0.678 | 0.684 | 0.709 | 0.677 | 0.64 | 0.681 | 0.675 | 0.676 |
| | | 3 × I | 0.276 | 0.541 | 0.668 | 0.667 | 0.706 | 0.694 | 0.681 | 0.681 | 0.671 | 0.677 | 0.678 | 0.676 | 0.703 | 0.667 | 0.681 | 0.677 |
| | | 4 × I | 0.4 | 0.512 | 0.657 | 0.682 | 0.693 | 0.697 | 0.682 | 0.675 | 0.68 | 0.69 | 0.685 | 0.682 | 0.663 | 0.698 | 0.68 | 0.69 |
| | | 5 × I | 0.5 | 0.629 | 0.686 | 0.68 | 0.693 | **0.723** | 0.699 | 0.673 | 0.673 | 0.673 | 0.697 | 0.685 | 0.692 | 0.686 | 0.692 | 0.707 |
| | 11 | 1 × I | 0.283 | 0.652 | 0.686 | 0.665 | 0.664 | 0.675 | 0.671 | 0.663 | 0.694 | 0.665 | 0.684 | 0.689 | 0.44 | 0.671 | 0.676 | 0.69 |
| | | 2 × I | 0.566 | 0.672 | 0.615 | 0.689 | 0.655 | 0.706 | 0.702 | 0.693 | 0.693 | 0.69 | 0.682 | 0.685 | 0.714 | 0.696 | 0.684 | 0.703 |
| | | 3 × I | 0.52 | 0.684 | 0.692 | 0.675 | 0.689 | 0.68 | 0.668 | 0.719 | 0.698 | 0.696 | 0.718 | 0.689 | 0.694 | 0.696 | 0.69 | 0.713 |
| | | 4 × I | 0.633 | 0.625 | 0.698 | 0.667 | 0.689 | 0.684 | 0.693 | 0.71 | 0.71 | 0.69 | 0.707 | 0.707 | 0.702 | 0.677 | 0.703 | 0.694 |
| | | 5 × I | 0.35 | 0.68 | **0.739** | 0.667 | 0.709 | 0.71 | 0.701 | 0.705 | 0.696 | 0.707 | 0.707 | 0.723 | 0.697 | 0.663 | 0.72 | 0.718 |
| Vehicle blanced | 3 | 1 × I | 0.567 | 0.625 | 0.566 | 0.698 | 0.636 | 0.685 | 0.612 | 0.685 | 0.651 | 0.677 | 0.664 | 0.667 | 0.684 | 0.702 | 0.644 | 0.283 |
| | | 2 × I | 0.672 | 0.572 | 0.651 | 0.71 | 0.701 | 0.65 | 0.66 | 0.668 | 0.723 | 0.669 | 0.706 | 0.724 | 0.706 | 0.685 | 0.698 | 0.677 |
| | | 3 × I | 0.677 | 0.512 | 0.711 | 0.72 | 0.707 | 0.681 | 0.692 | 0.723 | 0.722 | 0.723 | 0.68 | 0.715 | 0.706 | 0.709 | 0.697 | 0.709 |
| | | 4 × I | 0.379 | 0.623 | 0.676 | 0.705 | 0.694 | 0.713 | 0.715 | 0.714 | 0.731 | 0.701 | **0.743** | 0.715 | 0.694 | 0.719 | 0.689 | 0.724 |
| | | 5 × I | 0.669 | 0.469 | 0.72 | 0.69 | 0.707 | 0.722 | 0.684 | 0.706 | 0.707 | 0.676 | 0.699 | 0.684 | 0.701 | 0.693 | 0.727 | 0.722 |
| | 5 | 1 × I | 0.356 | 0.675 | 0.606 | 0.689 | 0.648 | 0.686 | 0.707 | 0.702 | 0.678 | 0.677 | 0.713 | 0.664 | 0.686 | 0.613 | 0.667 | 0.686 |
| | | 2 × I | 0.382 | 0.681 | 0.714 | 0.714 | 0.665 | 0.694 | 0.702 | 0.684 | 0.719 | 0.719 | 0.72 | 0.714 | 0.727 | 0.693 | 0.719 | 0.728 |
| | | 3 × I | 0.567 | 0.689 | 0.69 | 0.66 | 0.705 | 0.727 | 0.69 | 0.707 | 0.696 | 0.686 | 0.72 | 0.707 | 0.715 | 0.722 | 0.714 | 0.702 |
| | | 4 × I | 0.49 | 0.673 | 0.61 | 0.699 | 0.718 | 0.698 | 0.696 | 0.713 | 0.727 | 0.705 | 0.713 | **0.734** | 0.722 | 0.694 | 0.722 | 0.713 |
| | | 5 × I | 0.631 | 0.705 | 0.685 | 0.692 | 0.72 | 0.706 | 0.689 | 0.73 | 0.709 | 0.727 | 0.717 | 0.706 | 0.724 | 0.73 | 0.722 | 0.714 |
| | 7 | 1 × I | 0.509 | 0.65 | 0.707 | 0.702 | 0.677 | 0.707 | 0.705 | 0.727 | 0.747 | 0.437 | 0.719 | 0.706 | 0.719 | 0.283 | 0.728 | 0.719 |
| | | 2 × I | 0.487 | 0.685 | 0.689 | 0.681 | 0.724 | 0.701 | 0.717 | 0.686 | 0.719 | 0.72 | 0.744 | 0.72 | 0.732 | 0.722 | 0.741 | 0.743 |
| | | 3 × I | 0.281 | 0.686 | 0.705 | 0.702 | 0.693 | 0.728 | 0.726 | 0.719 | 0.731 | 0.749 | **0.765** | 0.732 | 0.741 | 0.74 | 0.71 | 0.73 |
| | | 4 × I | 0.433 | 0.572 | 0.672 | 0.697 | 0.723 | 0.732 | 0.727 | 0.728 | 0.707 | 0.739 | 0.736 | 0.756 | 0.734 | 0.743 | 0.723 | 0.709 |
| | | 5 × I | 0.28 | 0.72 | 0.735 | 0.731 | 0.714 | 0.717 | 0.744 | 0.736 | 0.744 | 0.743 | 0.744 | 0.744 | 0.722 | 0.732 | 0.73 | 0.739 |
| | 9 | 1 × I | 0.283 | 0.678 | 0.651 | 0.681 | 0.661 | 0.663 | 0.682 | 0.701 | 0.672 | 0.718 | 0.731 | 0.705 | 0.696 | 0.626 | 0.69 | 0.696 |
| | | 2 × I | 0.526 | 0.638 | 0.65 | 0.642 | 0.701 | 0.693 | 0.681 | 0.688 | 0.706 | 0.689 | 0.713 | 0.689 | 0.705 | 0.724 | 0.701 | 0.705 |
| | | 3 × I | 0.283 | 0.625 | 0.669 | 0.64 | 0.693 | 0.703 | 0.711 | 0.694 | 0.722 | 0.702 | 0.718 | 0.707 | 0.724 | 0.728 | 0.686 | 0.707 |
| | | 4 × I | 0.605 | 0.672 | 0.678 | 0.707 | 0.689 | 0.717 | 0.722 | 0.696 | 0.701 | 0.705 | 0.715 | 0.713 | 0.736 | 0.715 | 0.717 | 0.701 |
| | | 5 × I | 0.57 | 0.731 | 0.696 | 0.709 | 0.697 | 0.722 | 0.703 | 0.727 | 0.73 | 0.717 | 0.71 | 0.701 | 0.717 | 0.722 | **0.741** | 0.705 |
| | 11 | 1 × I | 0.579 | 0.648 | 0.661 | 0.655 | 0.682 | 0.696 | 0.706 | 0.705 | 0.702 | 0.711 | 0.701 | 0.702 | 0.709 | 0.698 | 0.707 | 0.69 |
| | | 2 × I | 0.497 | 0.642 | 0.709 | 0.711 | 0.719 | 0.713 | 0.714 | 0.684 | 0.703 | 0.724 | 0.73 | 0.706 | 0.724 | 0.728 | 0.711 | 0.728 |
| | | 3 × I | 0.648 | 0.631 | 0.714 | 0.719 | 0.734 | 0.699 | 0.72 | 0.732 | 0.711 | 0.723 | 0.705 | 0.735 | 0.714 | 0.714 | 0.696 | 0.734 |
| | | 4 × I | 0.503 | 0.673 | 0.715 | 0.706 | 0.743 | 0.744 | **0.747** | 0.706 | 0.703 | 0.726 | 0.706 | 0.715 | 0.722 | 0.701 | 0.718 | 0.723 |
| | | 5 × I | 0.644 | 0.597 | 0.718 | 0.698 | 0.713 | 0.726 | 0.736 | 0.714 | 0.713 | 0.727 | 0.726 | 0.732 | 0.727 | 0.732 | 0.724 | 0.715 |

**Table 19. Results of classification accuracy *acc* for the proposed approach and the Dry Bean data sets: Two hidden layers, the method to substitute values of missing attributes in local tables—Three artificial objects generated based on one original object, MLP networks aggregation using sum of weights and various number of neurons in the hidden layer (3AO-2HL-SUM).** Designation I is used for the number of neurons in the input layer.

| Data set | No. tables | Second HL | No. of neurons in the first hidden layer | | | | | | | | | | | | | | | |
|---|---|---|---|---|---|---|---|---|---|---|---|---|---|---|---|---|---|---|
| | | | 0.25 × I | 0.5 × I | 0.75 × I | 1 × I | 1.5 × I | 1.75 × I | 2 × I | 2.5 × I | 2.75 × I | 3 × I | 3.5 × I | 3.75 × I | 4 × I | 4.5 × I | 4.75 × I | 5 × I |
| Dry Bean imbalanced | 3 | 1 × I | 0.887 | 0.893 | 0.89 | 0.902 | 0.903 | 0.91 | 0.903 | 0.9 | 0.903 | 0.909 | 0.906 | 0.914 | 0.915 | 0.915 | 0.914 | 0.913 |
| | | 2 × I | 0.898 | 0.898 | 0.905 | 0.906 | 0.911 | 0.91 | 0.909 | 0.908 | 0.908 | 0.912 | 0.916 | 0.91 | 0.915 | 0.911 | 0.914 | 0.917 |
| | | 3 × I | 0.892 | 0.899 | 0.913 | 0.9 | 0.91 | 0.91 | 0.914 | 0.914 | 0.913 | 0.914 | 0.915 | 0.917 | 0.916 | 0.916 | **0.919** | 0.916 |
| | | 4 × I | 0.889 | 0.907 | 0.908 | 0.912 | 0.91 | 0.909 | 0.914 | 0.915 | 0.914 | 0.916 | 0.918 | 0.916 | 0.917 | 0.915 | **0.919** | 0.918 |
| | | 5 × I | 0.901 | 0.907 | 0.898 | 0.912 | 0.918 | 0.913 | 0.917 | 0.917 | 0.918 | 0.917 | 0.917 | 0.917 | **0.919** | 0.918 | 0.917 | 0.918 |
| | 5 | 1 × I | 0.888 | 0.885 | 0.892 | 0.897 | 0.901 | 0.91 | 0.908 | 0.905 | 0.902 | 0.907 | 0.904 | 0.909 | 0.906 | 0.902 | 0.91 | 0.901 |
| | | 2 × I | 0.895 | 0.897 | 0.899 | 0.901 | 0.904 | 0.904 | 0.906 | 0.908 | 0.914 | 0.913 | 0.907 | 0.909 | 0.912 | 0.913 | 0.914 | 0.907 |
| | | 3 × I | 0.896 | 0.9 | 0.909 | 0.911 | 0.914 | 0.905 | 0.914 | 0.914 | 0.909 | 0.913 | 0.916 | 0.916 | 0.914 | 0.915 | 0.915 | 0.916 |
| | | 4 × I | 0.89 | 0.9 | 0.902 | 0.903 | 0.909 | 0.906 | 0.909 | 0.914 | 0.914 | **0.917** | 0.913 | 0.915 | 0.914 | 0.916 | 0.915 | 0.915 |
| | | 5 × I | 0.888 | 0.898 | 0.909 | 0.913 | 0.901 | 0.908 | 0.91 | 0.909 | **0.917** | 0.915 | 0.916 | 0.916 | 0.913 | 0.915 | 0.914 | 0.913 |
| | 7 | 1 × I | 0.885 | 0.89 | 0.892 | 0.893 | 0.899 | 0.895 | 0.9 | 0.904 | 0.904 | 0.901 | 0.904 | 0.9 | 0.899 | 0.9 | 0.904 | 0.907 |
| | | 2 × I | 0.884 | 0.896 | 0.893 | 0.895 | 0.902 | 0.903 | 0.897 | 0.904 | 0.901 | 0.907 | 0.9 | 0.905 | 0.907 | 0.91 | 0.909 | 0.906 |
| | | 3 × I | 0.888 | 0.895 | 0.894 | 0.898 | 0.9 | 0.899 | 0.898 | 0.907 | 0.91 | 0.909 | 0.91 | 0.906 | 0.914 | 0.911 | 0.908 | 0.91 |
| | | 4 × I | 0.893 | 0.894 | 0.901 | 0.902 | 0.906 | 0.911 | 0.91 | 0.902 | 0.908 | 0.907 | 0.906 | 0.909 | 0.912 | 0.912 | 0.913 | 0.914 |
| | | 5 × I | 0.884 | 0.895 | 0.906 | 0.909 | 0.908 | 0.911 | 0.905 | 0.909 | 0.91 | 0.91 | 0.912 | 0.91 | 0.91 | 0.914 | **0.915** | 0.912 |
| | 9 | 1 × I | 0.886 | 0.886 | 0.894 | 0.89 | 0.891 | 0.896 | 0.893 | 0.901 | 0.897 | 0.895 | 0.899 | 0.894 | 0.903 | 0.9 | 0.903 | 0.903 |
| | | 2 × I | 0.883 | 0.888 | 0.896 | 0.891 | 0.9 | 0.901 | 0.897 | 0.903 | 0.9 | 0.902 | 0.903 | 0.906 | 0.909 | 0.902 | 0.903 | 0.907 |
| | | 3 × I | 0.895 | 0.892 | 0.892 | 0.899 | 0.899 | 0.905 | 0.899 | 0.908 | 0.907 | 0.903 | 0.909 | 0.907 | 0.908 | 0.904 | 0.9 | 0.91 |
| | | 4 × I | 0.893 | 0.896 | 0.893 | 0.898 | 0.899 | 0.906 | 0.906 | 0.902 | 0.908 | 0.909 | 0.908 | 0.909 | 0.906 | 0.907 | 0.914 | 0.912 |
| | | 5 × I | 0.89 | 0.895 | 0.896 | 0.901 | 0.904 | 0.9 | 0.911 | 0.909 | 0.91 | 0.914 | 0.907 | **0.915** | 0.91 | **0.915** | 0.912 | 0.911 |
| | 11 | 1 × I | 0.881 | 0.889 | 0.89 | 0.896 | 0.894 | 0.893 | 0.896 | 0.897 | 0.896 | 0.895 | 0.895 | 0.897 | 0.902 | 0.896 | 0.897 | 0.899 |
| | | 2 × I | 0.888 | 0.893 | 0.896 | 0.897 | 0.894 | 0.893 | 0.897 | 0.9 | 0.902 | 0.901 | 0.9 | 0.901 | 0.901 | 0.904 | 0.902 | 0.905 |
| | | 3 × I | 0.889 | 0.894 | 0.894 | 0.893 | 0.897 | 0.9 | 0.902 | 0.902 | 0.9 | 0.901 | 0.906 | 0.904 | 0.904 | 0.898 | 0.903 | 0.903 |
| | | 4 × I | 0.889 | 0.893 | 0.896 | 0.899 | 0.897 | 0.9 | 0.898 | 0.906 | 0.905 | 0.905 | 0.904 | 0.904 | 0.905 | 0.906 | 0.902 | 0.908 |
| | | 5 × I | 0.895 | 0.894 | 0.895 | 0.901 | 0.9 | 0.901 | 0.9 | 0.903 | 0.904 | 0.907 | 0.902 | 0.906 | 0.902 | **0.908** | 0.906 | **0.908** |
| Dry Bean blanced | 3 | 1 × I | 0.89 | 0.896 | 0.902 | 0.91 | 0.903 | 0.908 | 0.908 | 0.909 | 0.913 | 0.912 | 0.91 | 0.914 | 0.908 | 0.914 | 0.91 | 0.914 |
| | | 2 × I | 0.902 | 0.898 | 0.903 | 0.908 | 0.906 | 0.912 | 0.914 | 0.912 | 0.915 | 0.915 | 0.915 | 0.912 | 0.918 | 0.918 | 0.916 | 0.919 |
| | | 3 × I | 0.91 | 0.904 | 0.903 | 0.902 | 0.912 | 0.916 | 0.913 | 0.912 | 0.913 | 0.905 | 0.917 | 0.913 | 0.905 | 0.917 | 0.917 | 0.917 |
| | | 4 × I | 0.905 | 0.903 | 0.906 | 0.913 | 0.913 | 0.913 | 0.914 | 0.916 | 0.918 | 0.916 | 0.915 | 0.919 | 0.915 | 0.916 | 0.917 | 0.916 |
| | | 5 × I | 0.87 | 0.894 | 0.908 | 0.915 | 0.91 | 0.913 | 0.915 | 0.919 | 0.918 | 0.916 | 0.919 | 0.917 | 0.918 | 0.917 | **0.92** | 0.915 |
| | 5 | 1 × I | 0.888 | 0.893 | 0.895 | 0.893 | 0.896 | 0.908 | 0.905 | 0.909 | 0.912 | 0.914 | 0.902 | 0.912 | 0.915 | 0.916 | 0.914 | 0.915 |
| | | 2 × I | 0.907 | 0.907 | 0.904 | 0.909 | 0.905 | 0.905 | 0.911 | 0.91 | 0.914 | 0.913 | 0.917 | 0.916 | 0.915 | 0.916 | 0.917 | 0.917 |
| | | 3 × I | 0.891 | 0.909 | 0.91 | 0.91 | 0.91 | 0.914 | 0.913 | 0.918 | 0.916 | 0.916 | 0.915 | 0.918 | 0.916 | 0.915 | 0.916 | 0.917 |
| | | 4 × I | 0.887 | 0.909 | 0.907 | 0.907 | 0.911 | 0.917 | 0.917 | 0.915 | 0.912 | 0.915 | 0.918 | 0.914 | 0.918 | 0.915 | **0.919** | 0.916 |
| | | 5 × I | 0.893 | 0.91 | 0.905 | 0.914 | 0.915 | 0.914 | 0.916 | 0.917 | 0.918 | **0.919** | 0.916 | 0.917 | 0.918 | 0.917 | 0.918 | 0.918 |
| | 7 | 1 × I | 0.883 | 0.891 | 0.893 | 0.897 | 0.896 | 0.897 | 0.903 | 0.898 | 0.903 | 0.905 | 0.902 | 0.902 | 0.901 | 0.906 | 0.908 | 0.908 |
| | | 2 × I | 0.895 | 0.895 | 0.896 | 0.898 | 0.906 | 0.901 | 0.908 | 0.906 | 0.915 | 0.912 | 0.909 | 0.911 | 0.902 | 0.911 | 0.907 | 0.911 |
| | | 3 × I | 0.898 | 0.901 | 0.903 | 0.911 | 0.905 | 0.903 | 0.909 | 0.906 | 0.91 | 0.908 | 0.913 | 0.913 | 0.912 | 0.911 | 0.916 | 0.916 |
| | | 4 × I | 0.889 | 0.901 | 0.904 | 0.899 | 0.902 | 0.911 | 0.91 | 0.908 | 0.913 | 0.918 | 0.913 | 0.912 | 0.916 | 0.916 | 0.917 | **0.918** |
| | | 5 × I | 0.899 | 0.897 | 0.898 | 0.902 | 0.904 | 0.908 | 0.906 | 0.915 | 0.912 | 0.916 | 0.916 | 0.912 | 0.915 | 0.917 | 0.916 | **0.918** |
| | 9 | 1 × I | 0.884 | 0.89 | 0.891 | 0.89 | 0.901 | 0.896 | 0.897 | 0.904 | 0.9 | 0.897 | 0.902 | 0.902 | 0.909 | 0.898 | 0.907 | 0.903 |
| | | 2 × I | 0.892 | 0.888 | 0.891 | 0.893 | 0.901 | 0.901 | 0.902 | 0.898 | 0.907 | 0.903 | 0.908 | 0.901 | 0.91 | 0.912 | 0.909 | 0.911 |
| | | 3 × I | 0.888 | 0.891 | 0.899 | 0.896 | 0.908 | 0.906 | 0.905 | 0.904 | 0.91 | 0.91 | 0.908 | 0.911 | 0.907 | 0.915 | 0.913 | 0.913 |
| | | 4 × I | 0.888 | 0.895 | 0.893 | 0.908 | 0.905 | 0.901 | 0.901 | 0.908 | 0.905 | 0.911 | 0.911 | 0.912 | 0.908 | 0.911 | 0.915 | 0.913 |
| | | 5 × I | 0.891 | 0.896 | 0.901 | 0.906 | 0.91 | 0.907 | 0.903 | 0.909 | 0.909 | 0.909 | 0.914 | **0.916** | 0.913 | 0.913 | 0.913 | 0.912 |
| | 11 | 1 × I | 0.882 | 0.888 | 0.891 | 0.897 | 0.895 | 0.893 | 0.895 | 0.898 | 0.897 | 0.903 | 0.902 | 0.898 | 0.902 | 0.905 | 0.898 | 0.902 |
| | | 2 × I | 0.884 | 0.892 | 0.897 | 0.895 | 0.901 | 0.9 | 0.898 | 0.903 | 0.903 | 0.9 | 0.905 | 0.903 | 0.904 | 0.903 | 0.903 | 0.911 |
| | | 3 × I | 0.891 | 0.894 | 0.894 | 0.896 | 0.896 | 0.901 | 0.902 | 0.91 | 0.905 | 0.904 | 0.904 | 0.909 | 0.91 | 0.913 | 0.912 | 0.909 |
| | | 4 × I | 0.887 | 0.891 | 0.897 | 0.899 | 0.9 | 0.903 | 0.908 | 0.908 | 0.912 | 0.906 | 0.91 | 0.912 | 0.91 | 0.914 | 0.914 | 0.91 |
| | | 5 × I | 0.89 | 0.899 | 0.898 | 0.901 | 0.908 | 0.905 | 0.906 | 0.909 | 0.913 | 0.909 | 0.907 | 0.906 | 0.916 | 0.915 | **0.92** | 0.917 |

**Table 20. Results of classification accuracy *acc* for the proposed approach and the Sensorless data sets: Two hidden layers, the method to substitute values of missing attributes in local tables—Three artificial objects generated based on one original object, MLP networks aggregation using sum of weights and various number of neurons in the hidden layer (3AO-2HL-SUM).** Designation I is used for the number of neurons in the input layer.

| Data set | No. tables | Second HL | No. of neurons in the first hidden layer | | | | | | | | | | | | | | | |
|---|---|---|---|---|---|---|---|---|---|---|---|---|---|---|---|---|---|---|
| | | | 0.25 × I | 0.5 × I | 0.75 × I | 1 × I | 1.5 × I | 1.75 × I | 2 × I | 2.5 × I | 2.75 × I | 3 × I | 3.5 × I | 3.75 × I | 4 × I | 4.5 × I | 4.75 × I | 5 × I |
| Sensorless imbalanced | 3 | 1 × I | 0.917 | 0.92 | 0.928 | 0.929 | 0.948 | 0.94 | 0.938 | 0.945 | 0.942 | 0.943 | 0.935 | 0.942 | 0.942 | 0.937 | 0.94 | 0.941 |
| | | 2 × I | 0.917 | 0.933 | 0.934 | 0.944 | 0.948 | 0.942 | 0.939 | 0.943 | 0.942 | 0.945 | 0.942 | 0.945 | 0.942 | 0.942 | 0.94 | 0.943 |
| | | 3 × I | 0.937 | 0.931 | 0.945 | 0.945 | **0.952** | 0.944 | 0.946 | 0.943 | 0.947 | 0.945 | 0.942 | 0.938 | 0.941 | 0.934 | 0.942 | 0.947 |
| | | 4 × I | 0.927 | 0.946 | 0.942 | 0.949 | 0.944 | 0.945 | 0.941 | 0.947 | 0.942 | 0.945 | 0.941 | 0.944 | 0.943 | 0.939 | 0.93 | 0.942 |
| | | 5 × I | 0.929 | 0.949 | 0.943 | 0.952 | 0.947 | 0.946 | 0.949 | 0.945 | 0.946 | 0.95 | 0.943 | 0.94 | 0.945 | 0.937 | 0.943 | 0.943 |
| | 5 | 1 × I | 0.902 | 0.911 | 0.916 | 0.921 | 0.918 | 0.925 | 0.923 | 0.919 | 0.92 | 0.927 | 0.925 | 0.919 | 0.919 | 0.918 | 0.913 | 0.925 |
| | | 2 × I | 0.904 | 0.912 | 0.924 | 0.928 | 0.926 | 0.928 | 0.924 | 0.923 | 0.923 | 0.926 | 0.926 | 0.918 | 0.918 | 0.92 | 0.922 | 0.92 |
| | | 3 × I | 0.911 | 0.915 | 0.919 | 0.922 | 0.924 | 0.922 | 0.927 | 0.927 | 0.915 | 0.923 | 0.916 | 0.922 | 0.92 | 0.916 | 0.928 | 0.909 |
| | | 4 × I | 0.907 | 0.914 | **0.932** | 0.922 | 0.919 | 0.928 | 0.924 | 0.922 | 0.919 | 0.916 | 0.913 | 0.914 | 0.919 | 0.914 | 0.916 | 0.923 |
| | | 5 × I | 0.917 | 0.915 | 0.919 | 0.924 | 0.919 | 0.916 | 0.926 | 0.913 | 0.917 | 0.915 | 0.914 | 0.922 | 0.915 | 0.92 | 0.912 | 0.918 |
| | 7 | 1 × I | 0.905 | 0.915 | 0.912 | 0.926 | 0.921 | 0.927 | 0.924 | 0.93 | 0.93 | 0.933 | 0.93 | 0.924 | 0.927 | 0.93 | 0.928 | 0.929 |
| | | 2 × I | 0.916 | 0.92 | 0.92 | 0.923 | 0.917 | 0.923 | 0.93 | 0.93 | 0.929 | 0.928 | 0.918 | 0.929 | 0.927 | 0.93 | 0.93 | 0.93 |
| | | 3 × I | 0.915 | 0.918 | 0.927 | 0.928 | 0.929 | 0.93 | 0.924 | 0.921 | 0.922 | 0.931 | 0.929 | 0.932 | 0.933 | 0.92 | 0.927 | 0.929 |
| | | 4 × I | 0.918 | 0.918 | 0.927 | 0.929 | 0.929 | 0.927 | 0.925 | 0.93 | 0.93 | 0.926 | 0.928 | 0.929 | **0.934** | 0.926 | 0.92 | 0.93 |
| | | 5 × I | 0.92 | 0.921 | 0.924 | 0.925 | 0.929 | 0.93 | 0.923 | 0.922 | 0.927 | 0.929 | 0.931 | 0.922 | 0.926 | 0.924 | 0.925 | 0.922 |
| | 9 | 1 × I | 0.896 | 0.912 | 0.921 | 0.918 | 0.927 | 0.926 | 0.926 | 0.924 | 0.925 | 0.932 | 0.925 | 0.935 | 0.93 | 0.926 | 0.931 | 0.925 |
| | | 2 × I | 0.899 | 0.923 | 0.925 | 0.923 | 0.928 | **0.937** | 0.935 | 0.928 | 0.925 | 0.935 | 0.926 | 0.923 | 0.931 | 0.926 | 0.923 | 0.93 |
| | | 3 × I | 0.909 | 0.914 | 0.927 | 0.932 | 0.929 | 0.934 | 0.926 | 0.924 | 0.929 | 0.922 | 0.931 | 0.934 | 0.931 | 0.926 | 0.925 | 0.927 |
| | | 4 × I | 0.911 | 0.922 | 0.928 | 0.927 | 0.924 | 0.927 | 0.931 | 0.928 | 0.933 | 0.93 | 0.931 | 0.928 | 0.931 | 0.923 | 0.926 | 0.928 |
| | | 5 × I | 0.911 | 0.926 | 0.925 | 0.921 | 0.928 | 0.928 | 0.93 | 0.926 | 0.921 | 0.919 | 0.919 | 0.922 | 0.926 | 0.928 | 0.929 | 0.926 |
| | 11 | 1 × I | 0.879 | 0.908 | 0.914 | 0.925 | 0.925 | 0.931 | 0.93 | 0.932 | 0.921 | 0.93 | 0.932 | 0.929 | 0.934 | 0.933 | 0.927 | 0.922 |
| | | 2 × I | 0.894 | 0.916 | 0.921 | 0.918 | 0.93 | 0.926 | 0.926 | 0.932 | 0.924 | 0.93 | 0.929 | 0.931 | 0.93 | 0.932 | 0.932 | 0.932 |
| | | 3 × I | 0.908 | 0.917 | 0.92 | **0.935** | 0.928 | 0.934 | 0.934 | 0.93 | 0.929 | 0.932 | 0.933 | 0.931 | 0.929 | 0.93 | 0.929 | 0.931 |
| | | 4 × I | 0.898 | 0.91 | 0.929 | 0.932 | 0.932 | 0.934 | 0.929 | 0.929 | 0.932 | 0.929 | 0.932 | 0.925 | 0.927 | 0.923 | 0.93 | 0.926 |
| | | 5 × I | 0.913 | 0.915 | 0.929 | 0.926 | 0.93 | 0.926 | 0.927 | 0.93 | 0.923 | 0.919 | 0.929 | 0.926 | 0.927 | 0.928 | 0.93 | 0.925 |

**Table 21. Results of classification accuracy *acc* for the proposed approach and the Crowd Sourced data sets: Two hidden layers, the method to substitute values of missing attributes in local tables—Three artificial objects generated based on one original object, MLP networks aggregation using sum of weights and various number of neurons in the hidden layer (3AO-2HL-SUM).** Designation I is used for the number of neurons in the input layer.

| Data set | No. tables | Second HL | No. of neurons in the first hidden layer | | | | | | | | | | | | | | | |
| | | | 0.25 × I | 0.5 × I | 0.75 × I | 1 × I | 1.5 × I | 1.75 × I | 2 × I | 2.5 × I | 2.75 × I | 3 × I | 3.5 × I | 3.75 × I | 4 × I | 4.5 × I | 4.75 × I | 5 × I |
|---|---|---|---|---|---|---|---|---|---|---|---|---|---|---|---|---|---|---|
| Crowd Sourced imbalanced | 3 | 1 × I | 0.676 | 0.738 | 0.742 | 0.728 | 0.755 | 0.779 | 0.759 | 0.77 | 0.752 | 0.774 | 0.764 | 0.778 | 0.767 | 0.787 | 0.789 | 0.768 |
| | | 2 × I | 0.665 | 0.747 | 0.744 | 0.764 | 0.776 | 0.776 | 0.756 | 0.752 | 0.772 | 0.777 | 0.803 | 0.79 | 0.788 | 0.784 | 0.787 | 0.774 |
| | | 3 × I | 0.691 | 0.753 | 0.775 | 0.735 | 0.77 | 0.77 | 0.768 | 0.792 | 0.775 | 0.791 | 0.78 | 0.784 | 0.788 | 0.786 | 0.794 | 0.805 |
| | | 4 × I | 0.643 | 0.729 | 0.74 | 0.762 | 0.759 | 0.798 | 0.781 | 0.783 | 0.794 | 0.789 | 0.794 | 0.804 | 0.8 | 0.782 | 0.801 | 0.773 |
| | | 5 × I | 0.715 | 0.736 | 0.738 | 0.755 | 0.778 | 0.776 | 0.76 | 0.763 | 0.771 | 0.778 | 0.794 | 0.782 | 0.797 | 0.775 | **0.806** | 0.806 |
| | 5 | 1 × I | 0.615 | 0.758 | 0.738 | 0.757 | 0.773 | 0.77 | 0.779 | 0.756 | 0.814 | 0.782 | 0.805 | 0.799 | 0.792 | 0.788 | 0.785 | 0.835 |
| | | 2 × I | 0.68 | 0.734 | 0.718 | 0.773 | 0.772 | 0.777 | 0.757 | 0.769 | 0.773 | 0.794 | 0.826 | 0.823 | 0.774 | 0.814 | 0.809 | 0.833 |
| | | 3 × I | 0.702 | 0.755 | 0.748 | 0.782 | 0.778 | 0.794 | 0.776 | 0.793 | 0.807 | 0.812 | 0.812 | 0.81 | 0.796 | 0.815 | 0.821 | **0.838** |
| | | 4 × I | 0.691 | 0.74 | 0.766 | 0.771 | 0.782 | 0.784 | 0.794 | 0.808 | 0.777 | 0.815 | 0.816 | 0.807 | 0.8 | 0.816 | 0.826 | 0.805 |
| | | 5 × I | 0.692 | 0.739 | 0.761 | 0.795 | 0.776 | 0.799 | 0.798 | 0.792 | 0.807 | 0.807 | 0.819 | 0.814 | 0.818 | 0.822 | 0.832 | 0.829 |
| | 7 | 1 × I | 0.677 | 0.751 | 0.72 | 0.759 | 0.761 | 0.759 | 0.776 | 0.785 | 0.781 | 0.785 | 0.786 | 0.773 | 0.785 | 0.788 | 0.812 | 0.794 |
| | | 2 × I | 0.694 | 0.761 | 0.781 | 0.746 | 0.769 | 0.793 | 0.794 | 0.8 | 0.797 | 0.801 | 0.798 | 0.82 | 0.794 | 0.8 | 0.795 | 0.821 |
| | | 3 × I | 0.712 | 0.775 | 0.775 | 0.756 | 0.784 | 0.777 | 0.797 | 0.79 | 0.79 | 0.8 | 0.809 | 0.812 | 0.814 | 0.822 | 0.813 | 0.831 |
| | | 4 × I | 0.614 | 0.76 | 0.741 | 0.776 | 0.793 | 0.792 | 0.788 | 0.799 | 0.77 | 0.817 | 0.827 | 0.82 | 0.818 | 0.823 | 0.826 | 0.83 |
| | | 5 × I | 0.717 | 0.76 | 0.757 | 0.753 | 0.803 | 0.784 | 0.813 | 0.801 | 0.811 | 0.794 | 0.825 | 0.803 | 0.82 | **0.834** | 0.827 | 0.824 |
| | 9 | 1 × I | 0.756 | 0.77 | 0.775 | 0.763 | 0.785 | 0.788 | 0.738 | 0.785 | 0.809 | 0.786 | 0.806 | 0.804 | 0.816 | 0.812 | 0.827 | 0.819 |
| | | 2 × I | 0.74 | 0.765 | 0.783 | 0.776 | 0.811 | 0.785 | 0.817 | 0.825 | 0.804 | 0.806 | 0.824 | 0.813 | 0.814 | 0.837 | 0.794 | 0.829 |
| | | 3 × I | 0.675 | 0.779 | 0.754 | 0.758 | 0.819 | 0.815 | 0.807 | 0.8 | 0.804 | 0.825 | 0.815 | 0.812 | 0.832 | 0.829 | 0.825 | 0.827 |
| | | 4 × I | 0.722 | 0.793 | 0.816 | 0.774 | 0.782 | 0.811 | 0.791 | 0.824 | 0.82 | 0.829 | 0.81 | **0.844** | 0.817 | 0.832 | 0.828 | 0.83 |
| | | 5 × I | 0.719 | 0.773 | 0.79 | 0.782 | 0.835 | 0.793 | 0.811 | 0.799 | 0.812 | 0.829 | 0.829 | 0.821 | **0.838** | 0.835 | 0.843 | 0.842 |
| | 11 | 1 × I | 0.706 | 0.767 | 0.749 | 0.776 | 0.773 | 0.741 | 0.765 | 0.812 | 0.802 | 0.813 | 0.791 | 0.815 | 0.821 | 0.808 | 0.817 | 0.799 |
| | | 2 × I | 0.727 | 0.747 | 0.772 | 0.771 | 0.765 | 0.798 | 0.798 | 0.802 | 0.811 | 0.812 | 0.821 | 0.81 | 0.802 | 0.837 | **0.814** | 0.811 |
| | | 3 × I | 0.67 | 0.76 | 0.799 | 0.803 | 0.766 | 0.802 | 0.775 | 0.806 | 0.814 | 0.82 | 0.821 | 0.793 | 0.802 | 0.815 | 0.827 | 0.824 |
| | | 4 × I | 0.741 | 0.796 | 0.786 | 0.743 | 0.81 | 0.78 | 0.819 | 0.816 | 0.824 | 0.789 | 0.825 | 0.821 | 0.827 | 0.832 | 0.83 | 0.824 |
| | | 5 × I | 0.71 | 0.759 | 0.81 | 0.799 | 0.812 | 0.82 | 0.783 | 0.78 | 0.805 | 0.822 | 0.817 | 0.821 | 0.838 | 0.838 | 0.829 | 0.826 |
| Crowd Sourced balanced | 3 | 1 × I | 0.57 | 0.68 | 0.715 | 0.711 | 0.736 | 0.703 | 0.703 | 0.737 | 0.736 | 0.762 | 0.739 | 0.744 | 0.74 | 0.753 | 0.761 | 0.758 |
| | | 2 × I | 0.63 | 0.653 | 0.715 | 0.697 | 0.759 | 0.727 | 0.755 | 0.728 | 0.724 | 0.725 | 0.739 | 0.765 | 0.806 | 0.758 | 0.814 | 0.774 |
| | | 3 × I | 0.596 | 0.669 | 0.712 | 0.69 | 0.649 | 0.784 | 0.731 | 0.763 | 0.727 | 0.784 | 0.734 | 0.791 | 0.782 | 0.785 | 0.721 | 0.77 |
| | | 4 × I | 0.591 | 0.647 | 0.702 | 0.666 | 0.711 | 0.676 | 0.755 | 0.728 | 0.753 | 0.763 | 0.737 | 0.803 | 0.748 | 0.763 | 0.759 | 0.786 |
| | | 5 × I | 0.575 | 0.704 | 0.734 | 0.708 | 0.738 | 0.72 | 0.746 | 0.769 | 0.731 | 0.745 | 0.692 | 0.775 | 0.781 | 0.763 | 0.774 | 0.789 |
| | 5 | 1 × I | 0.442 | 0.68 | 0.701 | 0.734 | 0.787 | 0.792 | 0.761 | 0.827 | 0.813 | 0.811 | 0.795 | 0.848 | 0.83 | 0.827 | 0.847 | 0.861 |
| | | 2 × I | 0.386 | 0.713 | 0.732 | 0.746 | 0.768 | 0.8 | 0.803 | 0.83 | 0.816 | 0.829 | 0.823 | 0.831 | 0.848 | 0.861 | 0.847 | 0.857 |
| | | 3 × I | 0.528 | 0.612 | 0.695 | 0.774 | 0.776 | 0.809 | 0.806 | 0.832 | 0.825 | 0.819 | 0.839 | 0.83 | 0.849 | 0.864 | **0.875** | 0.811 |
| | | 4 × I | 0.466 | 0.737 | 0.744 | 0.763 | 0.791 | 0.812 | 0.803 | 0.838 | 0.816 | 0.814 | 0.84 | 0.861 | 0.84 | 0.843 | 0.866 | 0.872 |
| | | 5 × I | 0.521 | 0.727 | 0.717 | 0.794 | 0.835 | 0.791 | 0.817 | 0.792 | 0.854 | 0.822 | 0.826 | 0.821 | 0.834 | 0.859 | 0.833 | 0.872 |
| | 7 | 1 × I | 0.403 | 0.716 | 0.75 | 0.745 | 0.78 | 0.775 | 0.725 | 0.832 | 0.833 | 0.84 | 0.863 | 0.831 | 0.854 | 0.85 | 0.845 | 0.866 |
| | | 2 × I | 0.585 | 0.649 | 0.762 | 0.765 | 0.81 | 0.805 | 0.796 | 0.837 | 0.849 | 0.848 | 0.863 | 0.857 | 0.852 | 0.874 | 0.857 | 0.872 |
| | | 3 × I | 0.394 | 0.683 | 0.721 | 0.762 | 0.827 | 0.82 | 0.813 | 0.862 | 0.854 | 0.844 | 0.876 | 0.844 | 0.852 | 0.864 | 0.877 | 0.891 |
| | | 4 × I | 0.513 | 0.739 | 0.735 | 0.794 | 0.814 | 0.812 | 0.83 | 0.852 | 0.849 | 0.853 | 0.844 | 0.869 | 0.852 | 0.878 | 0.879 | 0.883 |
| | | 5 × I | 0.6 | 0.726 | 0.788 | 0.781 | 0.796 | 0.842 | 0.849 | 0.864 | 0.864 | 0.846 | 0.864 | 0.868 | 0.873 | 0.875 | **0.894** | 0.879 |
| | 9 | 1 × I | 0.541 | 0.717 | 0.747 | 0.775 | 0.777 | 0.799 | 0.805 | 0.794 | 0.801 | 0.815 | 0.849 | 0.839 | 0.879 | 0.862 | 0.853 | 0.849 |
| | | 2 × I | 0.547 | 0.699 | 0.738 | 0.733 | 0.808 | 0.801 | 0.837 | 0.838 | 0.843 | 0.83 | 0.833 | 0.841 | 0.858 | 0.879 | 0.875 | 0.877 |
| | | 3 × I | 0.551 | 0.679 | 0.795 | 0.765 | 0.789 | 0.815 | 0.826 | 0.858 | 0.831 | 0.847 | 0.87 | 0.867 | 0.87 | 0.885 | 0.873 | 0.883 |
| | | 4 × I | 0.333 | 0.704 | 0.747 | 0.766 | 0.837 | 0.796 | 0.824 | 0.836 | 0.853 | 0.862 | 0.878 | 0.855 | 0.874 | 0.883 | 0.888 | 0.886 |
| | | 5 × I | 0.486 | 0.726 | 0.751 | 0.776 | 0.792 | 0.849 | 0.858 | 0.848 | 0.864 | 0.873 | 0.883 | 0.889 | 0.864 | 0.879 | 0.842 | **0.902** |
| | 11 | 1 × I | 0.475 | 0.664 | 0.703 | 0.739 | 0.786 | 0.818 | 0.787 | 0.815 | 0.818 | 0.829 | 0.841 | 0.837 | 0.857 | 0.809 | 0.858 | 0.875 |
| | | 2 × I | 0.522 | 0.703 | 0.72 | 0.779 | 0.738 | 0.806 | 0.817 | 0.836 | 0.845 | 0.851 | 0.863 | 0.855 | 0.866 | 0.876 | 0.893 | 0.894 |
| | | 3 × I | 0.662 | 0.689 | 0.749 | 0.774 | 0.79 | 0.794 | 0.784 | 0.845 | 0.874 | 0.865 | 0.865 | 0.874 | 0.84 | 0.877 | 0.872 | 0.894 |
| | | 4 × I | 0.594 | 0.73 | 0.769 | 0.765 | 0.821 | 0.84 | 0.838 | 0.859 | 0.858 | 0.868 | 0.872 | 0.853 | 0.888 | 0.881 | 0.893 | 0.888 |
| | | 5 × I | 0.607 | 0.737 | 0.792 | 0.762 | 0.816 | 0.825 | 0.833 | 0.873 | 0.854 | 0.871 | 0.875 | 0.887 | 0.881 | 0.887 | 0.896 | **0.904** |

**Table 22. Comparison of classification accuracy *acc* obtained for different numbers of artificial objects.**

| Data set | No. tables | 1AO 1HL | 3AO 1HL | 1AO 2HL | 3AO 2HL | |
|---|---|---|---|---|---|---|
| Vehicle imbalanced | 3 | 0.693 | **0.718** | **0.739** | 0.732 | AVG |
| | 5 | **0.703** | 0.693 | 0.709 | **0.711** | |
| | 7 | **0.69** | 0.686 | **0.724** | 0.702 | |
| | 9 | 0.677 | **0.694** | **0.713** | 0.686 | |
| | 11 | **0.696** | 0.69 | **0.706** | 0.698 | |
| Vehicle balanced | 3 | **0.727** | 0.722 | 0.736 | **0.76** | |
| | 5 | 0.703 | **0.72** | 0.728 | **0.732** | |
| | 7 | 0.722 | **0.732** | **0.757** | 0.74 | |
| | 9 | **0.73** | 0.702 | 0.719 | **0.728** | |
| | 11 | 0.688 | **0.692** | **0.723** | 0.71 | |
| Dry Bean imbalanced | 3 | **0.909** | **0.909** | 0.906 | **0.91** | |
| | 5 | 0.908 | **0.91** | **0.902** | 0.901 | |
| | 7 | **0.908** | 0.907 | **0.903** | 0.902 | |
| | 9 | **0.906** | **0.906** | **0.904** | 0.903 | |
| | 11 | 0.906 | **0.907** | **0.903** | **0.903** | |
| Dry Bean balanced | 3 | **0.913** | 0.911 | 0.912 | **0.918** | |
| | 5 | 0.909 | **0.91** | **0.901** | **0.901** | |
| | 7 | **0.907** | **0.907** | **0.903** | 0.902 | |
| | 9 | **0.909** | 0.906 | 0.903 | **0.904** | |
| | 11 | 0.904 | **0.905** | 0.901 | **0.904** | |
| Sensorless | 3 | **0.921** | 0.92 | 0.95 | **0.951** | |
| | 5 | **0.916** | 0.914 | **0.945** | 0.944 | |
| | 7 | **0.919** | **0.919** | **0.947** | 0.943 | |
| | 9 | **0.918** | 0.916 | **0.943** | 0.94 | |
| | 11 | 0.918 | **0.922** | **0.942** | 0.939 | |
| Crowd Sourced imbalanced | 3 | **0.827** | 0.806 | **0.838** | 0.833 | |
| | 5 | **0.84** | 0.827 | **0.875** | 0.869 | |
| | 7 | **0.851** | 0.837 | 0.855 | **0.867** | |
| | 9 | **0.851** | 0.846 | 0.859 | 0.859 | |
| | 11 | 0.846 | **0.871** | **0.876** | 0.861 | |
| Crowd Sourced balanced | 3 | **0.892** | 0.833 | **0.915** | 0.858 | |
| | 5 | **0.914** | 0.892 | **0.927** | 0.919 | |
| | 7 | **0.922** | 0.916 | 0.918 | 0.918 | |
| | 9 | 0.916 | **0.921** | 0.915 | **0.916** | |
| | 11 | 0.904 | **0.914** | 0.921 | **0.923** | |

(*Continued*)

**Table 22.** (Continued)

| Data set | No. tables | 1AO 1HL | 3AO 1HL | 1AO 2HL | 3AO 2HL | |
|---|---|---|---|---|---|---|
| Vehicle imbalanced | 3 | **0.675** | 0.665 | **0.713** | **0.713** | SUM |
| | 5 | **0.673** | 0.671 | 0.715 | **0.72** | |
| | 7 | 0.682 | **0.696** | **0.735** | 0.724 | |
| | 9 | 0.697 | **0.717** | **0.752** | 0.723 | |
| | 11 | **0.694** | 0.688 | **0.743** | 0.739 | |
| Vehicle balanced | 3 | **0.703** | 0.677 | 0.726 | **0.743** | |
| | 5 | 0.71 | **0.717** | **0.748** | 0.734 | |
| | 7 | 0.723 | **0.735** | 0.759 | **0.765** | |
| | 9 | **0.743** | 0.711 | **0.745** | 0.741 | |
| | 11 | **0.714** | 0.705 | **0.768** | 0.747 | |
| Dry Bean imbalanced | 3 | **0.912** | 0.911 | **0.92** | 0.919 | |
| | 5 | 0.912 | **0.913** | **0.918** | 0.917 | |
| | 7 | **0.914** | 0.913 | **0.917** | 0.915 | |
| | 9 | **0.914** | 0.913 | **0.915** | **0.915** | |
| | 11 | **0.913** | 0.911 | **0.914** | 0.908 | |
| Dry Bean balanced | 3 | **0.912** | 0.911 | 0.919 | **0.92** | |
| | 5 | **0.916** | 0.913 | 0.917 | **0.919** | |
| | 7 | 0.912 | **0.913** | **0.919** | 0.918 | |
| | 9 | **0.913** | 0.912 | **0.917** | 0.916 | |
| | 11 | **0.912** | **0.912** | 0.917 | **0.92** | |
| Sensorless | 3 | **0.92** | 0.918 | 0.949 | **0.952** | |
| | 5 | **0.911** | 0.904 | **0.934** | 0.932 | |
| | 7 | **0.915** | 0.908 | **0.942** | 0.934 | |
| | 9 | **0.917** | 0.908 | **0.942** | 0.937 | |
| | 11 | **0.914** | 0.91 | **0.941** | 0.935 | |
| Crowd Sourced imbalanced | 3 | 0.796 | **0.806** | **0.838** | 0.806 | |
| | 5 | **0.817** | 0.802 | **0.845** | 0.838 | |
| | 7 | **0.829** | 0.818 | **0.842** | 0.834 | |
| | 9 | **0.832** | 0.808 | 0.839 | **0.844** | |
| | 11 | 0.827 | **0.842** | **0.861** | 0.838 | |
| Crowd Sourced balanced | 3 | **0.864** | 0.782 | **0.879** | 0.814 | |
| | 5 | **0.874** | 0.839 | **0.894** | 0.875 | |
| | 7 | 0.864 | **0.866** | **0.899** | 0.894 | |
| | 9 | **0.86** | 0.855 | 0.893 | **0.902** | |
| | 11 | 0.861 | **0.863** | 0.903 | **0.904** | |

are marked in bold. Comparisons of experimental results with respect to different factors are made in separate sections.

## 5.1 Comparison of classification quality for different number of artificial objects created based on one original object in local table

Table 22 shows a comparison of the classification accuracy obtained for one and three generated artificial objects at various other settings (these are the best results which have been presented in bold in previous tables). For each setting and data set, the better result is marked in bold. In ninety-three cases, better results are obtained with one artificial object, and in fifty-four cases with three artificial objects generated. Thus, in most cases, generating just one

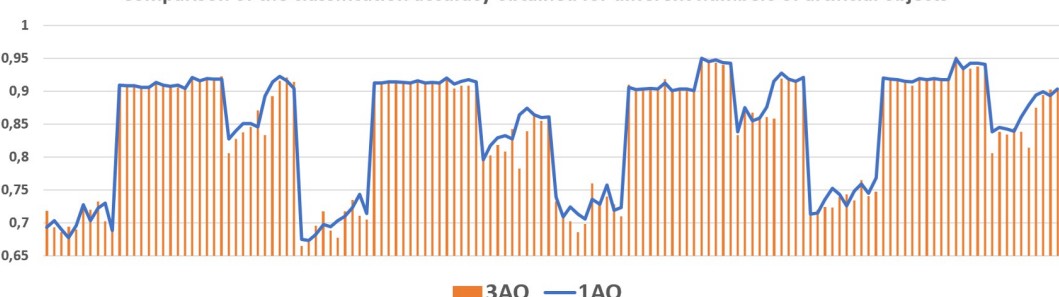

**Fig 5. Comparison of classification accuracy *acc* obtained for different aggregation method.**

artificial object based on original object is enough to get better result. For statistical test [51], the two dependent groups are created (1AO and 3AO), each with one hundred and forty objects. Initially, the null hypothesis, $H_\theta$ is defined, whereby $H_\theta$ means there is no significance in terms of the number of artificial objects used in the model. The Wilcoxon test for dependent samples confirmed the statistical significance of the differences with $p = 0.0003$ (we have justification for rejecting the null hypothesis) and the medians are equal 0.9 and 0.893 for groups 1AO and 3AO respectively. This proves that actually using one artificial object on average generates better results than using three artificial objects.

In addition, Fig 5 is created on which the results for all the settings and data sets analyzed are marked with respect to different artificial objects generated (individual cases are not labeled on the x-axis for clarity). In the graph, one can observe that indeed using one artificial object in many cases generates better results.

## 5.2 Comparison of classification quality for different approaches: Average of weights and sum of weights to aggregating local neural networks

Table 23 shows a comparison of the classification accuracy obtained for methods for aggregating local networks: average and sum. For each setting and data set, the better result is marked in bold. In seventy-four cases, better results are obtained for average, and in sixty-nine cases for sum method. Thus, in most cases, the global network obtained by the sum of the weights provides better results. A statistical test is performed to confirm the significance of differences. The null hypothesis, $H_\theta$ is defined, whereby $H_\theta$ means there is no significance in terms of the method for aggregating local neural networks in the model. The two dependent groups are created (AVG and SUM), each with one hundred and forty objects. The Wilcoxon test for dependent samples confirmed the statistical significance of the differences in accuracy with $p = 0.04$ —so we have justification for rejecting the null hypothesis. The medians are equal 0.902 and 0.87 for groups AVG and SUM respectively. This proves that actually using the average approach generates better results. However, it should be noted that the results are very dependent on the data set. For the Vehicle and the Dry Bean data sets, it is definitely apparent that the sum method provides better results. However, for the Sensorless and the Crowd Sourced data sets, it is the average method that provides better results.

In addition, Fig 6 is created on which the results for all the settings and data sets analyzed are marked (individual cases are not labeled on the x-axis for clarity). In the graph, one can observer that indeed the average in many cases generates better results.

**Table 23. Comparison of classification accuracy *acc* obtained for different aggregation method and different numbers of hidden layers.**

| Data set | No. tables | AVG 1HL | SUM 1HL | AVG 2HL | SUM 2HL | |
|---|---|---|---|---|---|---|
| Vehicle imbalanced | 3 | **0.693** | 0.675 | 0.739 | 0.713 | 1AO |
| | 5 | 0.703 | **0.673** | 0.709 | 0.715 | |
| | 7 | **0.69** | 0.682 | 0.724 | 0.735 | |
| | 9 | 0.677 | **0.697** | 0.713 | 0.752 | |
| | 11 | **0.696** | 0.694 | 0.706 | 0.743 | |
| Vehicle balanced | 3 | **0.727** | 0.703 | 0.736 | 0.726 | |
| | 5 | 0.703 | **0.71** | 0.728 | 0.748 | |
| | 7 | 0.722 | **0.723** | 0.757 | 0.759 | |
| | 9 | 0.73 | **0.743** | 0.719 | 0.745 | |
| | 11 | 0.688 | **0.714** | 0.723 | 0.768 | |
| Dry Bean imbalanced | 3 | 0.909 | **0.912** | 0.906 | 0.92 | |
| | 5 | 0.908 | **0.912** | 0.902 | 0.918 | |
| | 7 | 0.908 | **0.914** | 0.903 | 0.917 | |
| | 9 | 0.906 | **0.914** | 0.904 | 0.915 | |
| | 11 | 0.906 | **0.913** | 0.903 | 0.914 | |
| Dry Bean balanced | 3 | 0.913 | 0.912 | 0.912 | 0.919 | |
| | 5 | 0.909 | **0.916** | 0.901 | 0.917 | |
| | 7 | 0.907 | **0.912** | 0.903 | 0.919 | |
| | 9 | 0.909 | **0.913** | 0.903 | 0.917 | |
| | 11 | 0.904 | **0.912** | 0.901 | 0.917 | |
| Sensorless | 3 | **0.921** | 0.92 | 0.95 | 0.949 | |
| | 5 | **0.916** | 0.911 | 0.945 | 0.934 | |
| | 7 | **0.919** | 0.915 | 0.947 | 0.942 | |
| | 9 | **0.918** | 0.917 | 0.943 | 0.942 | |
| | 11 | **0.918** | 0.914 | 0.942 | 0.941 | |
| Crowd Sourced imbalanced | 3 | **0.827** | 0.796 | **0.838** | **0.838** | |
| | 5 | **0.84** | 0.817 | **0.875** | 0.845 | |
| | 7 | **0.851** | 0.829 | **0.855** | 0.842 | |
| | 9 | **0.851** | 0.832 | **0.859** | 0.839 | |
| | 11 | **0.846** | 0.827 | **0.876** | 0.861 | |
| Crowd Sourced balanced | 3 | **0.892** | 0.864 | **0.915** | 0.879 | |
| | 5 | **0.914** | 0.874 | **0.927** | 0.894 | |
| | 7 | **0.922** | 0.864 | 0.918 | 0.899 | |
| | 9 | **0.916** | 0.86 | 0.915 | 0.893 | |
| | 11 | **0.904** | 0.861 | **0.921** | 0.903 | |

(*Continued*)

**Table 23.** (Continued)

| Data set | No. tables | AVG 1HL | SUM 1HL | AVG 2HL | SUM 2HL | |
|---|---|---|---|---|---|---|
| Vehicle imbalanced | 3 | **0.718** | 0.665 | <u>0.732</u> | <u>0.713</u> | 3AO |
| | 5 | **0.693** | 0.671 | <u>0.711</u> | <u>0.72</u> | |
| | 7 | 0.686 | **0.696** | <u>0.702</u> | <u>0.724</u> | |
| | 9 | <u>0.694</u> | **0.717** | 0.686 | <u>0.723</u> | |
| | 11 | **0.69** | 0.688 | <u>0.698</u> | <u>0.739</u> | |
| Vehicle balanced | 3 | **0.722** | 0.677 | <u>0.76</u> | <u>0.743</u> | |
| | 5 | **0.72** | 0.717 | <u>0.732</u> | <u>0.734</u> | |
| | 7 | 0.732 | **0.735** | <u>0.74</u> | <u>0.765</u> | |
| | 9 | 0.702 | **0.711** | <u>0.728</u> | <u>0.741</u> | |
| | 11 | 0.692 | **0.705** | <u>0.71</u> | <u>0.747</u> | |
| Dry Bean imbalanced | 3 | 0.909 | **0.911** | <u>0.91</u> | <u>0.919</u> | |
| | 5 | <u>0.91</u> | **0.913** | 0.901 | <u>0.917</u> | |
| | 7 | <u>0.907</u> | **0.913** | 0.902 | <u>0.915</u> | |
| | 9 | <u>0.906</u> | **0.913** | 0.903 | <u>0.915</u> | |
| | 11 | <u>0.907</u> | <u>0.911</u> | 0.903 | **0.908** | |
| Dry Bean balanced | 3 | **0.911** | **0.911** | <u>0.918</u> | <u>0.92</u> | |
| | 5 | <u>0.91</u> | **0.913** | 0.901 | <u>0.919</u> | |
| | 7 | <u>0.907</u> | **0.913** | 0.902 | <u>0.918</u> | |
| | 9 | <u>0.906</u> | **0.912** | 0.904 | <u>0.916</u> | |
| | 11 | <u>0.905</u> | **0.912** | 0.904 | <u>0.92</u> | |
| Sensorless | 3 | **0.92** | 0.918 | <u>0.951</u> | <u>0.952</u> | |
| | 5 | **0.914** | 0.904 | <u>0.944</u> | <u>0.932</u> | |
| | 7 | **0.919** | 0.908 | <u>0.943</u> | <u>0.934</u> | |
| | 9 | **0.916** | 0.908 | <u>0.94</u> | <u>0.937</u> | |
| | 11 | **0.922** | 0.91 | <u>0.939</u> | <u>0.935</u> | |
| Crowd Sourced imbalanced | 3 | **0.806** | <u>0.806</u> | **0.833** | 0.806 | |
| | 5 | **0.827** | 0.802 | **0.869** | <u>0.838</u> | |
| | 7 | **0.837** | 0.818 | **0.867** | <u>0.834</u> | |
| | 9 | **0.846** | 0.808 | **0.859** | <u>0.844</u> | |
| | 11 | **0.871** | <u>0.842</u> | **0.861** | 0.838 | |
| Crowd Sourced balanced | 3 | **0.833** | 0.782 | **0.858** | <u>0.814</u> | |
| | 5 | **0.892** | 0.839 | **0.919** | <u>0.875</u> | |
| | 7 | **0.916** | 0.866 | **0.918** | <u>0.894</u> | |
| | 9 | <u>0.921</u> | 0.855 | **0.916** | <u>0.902</u> | |
| | 11 | **0.914** | 0.863 | **0.923** | <u>0.904</u> | |

### 5.3 Comparison of classification quality for different numbers of hidden layers in the local and global networks

Table 23 also indicates a comparison of the results obtained for one and two hidden layers for each data set and settings analyzed. The better results are underlined. As can be seen, in twenty-seven cases the better results are obtained when using one hidden layer, while in one hundred and seventeen cases the better results are obtained when using two hidden layers. Thus, the use of two hidden layers generates better results in most cases. For statistical tests the

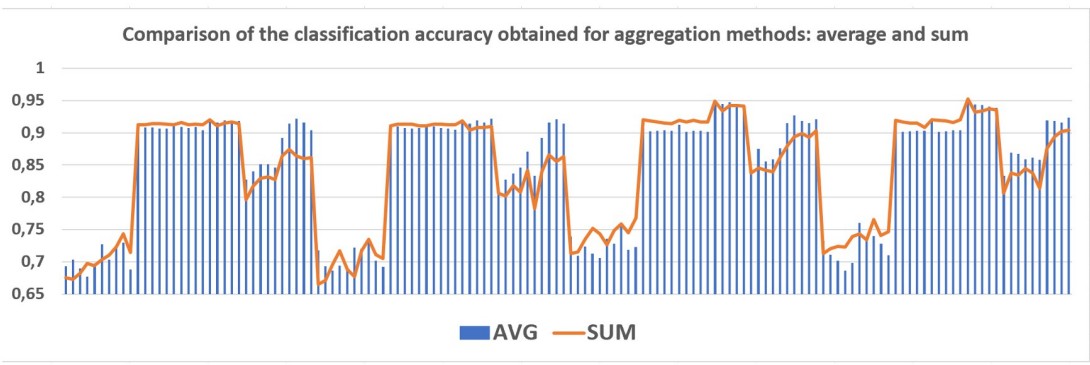

**Fig 6. Comparison of classification accuracy *acc* obtained for different aggregation method.**

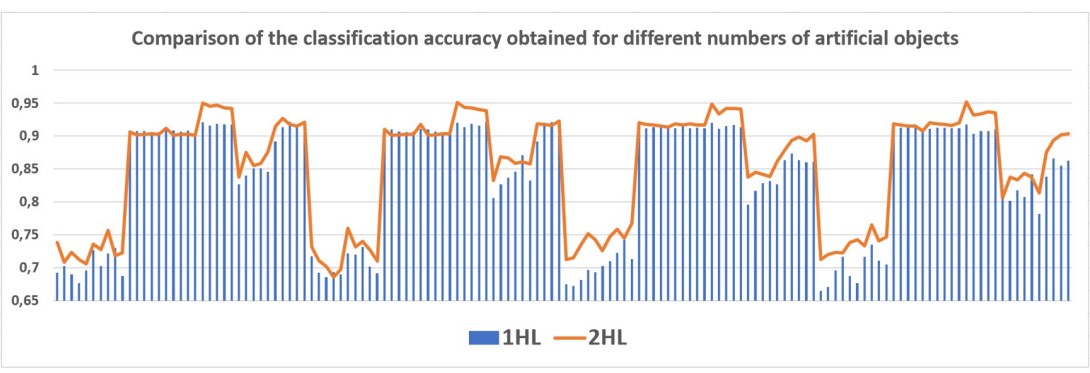

**Fig 7. Comparison of classification accuracy *acc* obtained for different numbers of hidden layers.**

null hypothesis, $H_\theta$ is defined, whereby $H_\theta$ means there is no significance in terms of the method for aggregating local neural networks in the model. The Wilcoxon test for dependent samples confirmed the statistical significance of the differences in accuracy with $p = 0.0001$. Also, Fig 7 is created on which the results for all the settings and data sets analyzed are compared for one and two hidden layers. In the graph, it is evident the two hidden layers networks in many cases generates better results.

## 5.4 Comparison of classification quality of the proposed method versus other approaches

Table 24 shows all the results obtained for the proposed approach, different setting and all analyzed each data set. Based on the previous analyzes, it is concluded that in most cases, the best results are obtained with using one artificial object, the sum as aggregation method, and two hidden layers. But now, based on the summarized results in Table 24, the best approach and result is selected for each data set (marked in bold).

Table 25 shows the best obtained accuracy for the proposed approach and two known approaches from the literature: homogeneous ensemble of MLP network classifiers and ensemble of classifiers (KNN, DT, NB) with soft voting, which are described in detail in the previous section. The best result is shown in bold. The proposed method is the best which

**Table 24. Comparison of classification accuracy *acc* obtained for the proposed method and approaches: 1AO-1HL-AVG, 3AO-1HL-AVG, 1AO-1HL-SUM, 3AO-1HL-SUM, 1AO-2HL-AVG, 3AO-2HL-AVG, 1AO-2HL-SUM, 3AO-2HL-SUM.**

| Data set | No. tables | Number of artificial objects generated based on one original object | | | | | | | |
|---|---|---|---|---|---|---|---|---|---|
| | | 1AO- | 3AO- | 1AO- | 3AO- | 1AO- | 3AO- | 1AO- | 3AO- |
| | | Number of hidden layers | | | | | | | |
| | | 1HL- | 1HL- | 1HL- | 1HL- | 2HL- | 2HL- | 2HL- | 2HL- |
| | | Aggregation method | | | | | | | |
| | | AVG | AVG | SUM | SUM | AVG | AVG | SUM | SUM |
| Vehicle imbalanced | 3 | 0.693 | 0.718 | 0.675 | 0.665 | **0.739** | 0.732 | 0.713 | 0.713 |
| | 5 | 0.703 | 0.693 | 0.673 | 0.671 | 0.709 | 0.711 | 0.715 | **0.72** |
| | 7 | 0.69 | 0.686 | 0.682 | 0.696 | 0.724 | 0.702 | **0.735** | 0.724 |
| | 9 | 0.677 | 0.694 | 0.697 | 0.717 | 0.713 | 0.686 | **0.752** | 0.723 |
| | 11 | 0.696 | 0.69 | 0.694 | 0.688 | 0.706 | 0.698 | **0.743** | 0.739 |
| Vehicle balanced | 3 | 0.727 | 0.722 | 0.703 | 0.677 | 0.736 | **0.76** | 0.726 | 0.743 |
| | 5 | 0.703 | 0.72 | 0.71 | 0.717 | 0.728 | 0.732 | **0.748** | 0.734 |
| | 7 | 0.722 | 0.732 | 0.723 | 0.735 | 0.757 | 0.74 | 0.759 | **0.765** |
| | 9 | 0.73 | 0.702 | 0.743 | 0.711 | 0.719 | 0.728 | **0.745** | 0.741 |
| | 11 | 0.688 | 0.692 | 0.714 | 0.705 | 0.723 | 0.71 | **0.768** | 0.747 |
| Dry Bean imbalanced | 3 | 0.909 | 0.909 | 0.912 | 0.911 | 0.906 | 0.91 | **0.92** | 0.919 |
| | 5 | 0.908 | 0.91 | 0.912 | 0.913 | 0.902 | 0.901 | **0.918** | 0.917 |
| | 7 | 0.908 | 0.907 | 0.914 | 0.913 | 0.903 | 0.902 | **0.917** | 0.915 |
| | 9 | 0.906 | 0.906 | 0.914 | 0.913 | 0.904 | 0.903 | **0.915** | **0.915** |
| | 11 | 0.906 | 0.907 | 0.913 | 0.911 | 0.903 | 0.903 | **0.914** | 0.908 |
| Dry Bean balanced | 3 | 0.913 | 0.911 | 0.912 | 0.911 | 0.912 | 0.918 | 0.919 | **0.92** |
| | 5 | 0.909 | 0.91 | 0.916 | 0.913 | 0.901 | 0.901 | 0.917 | **0.919** |
| | 7 | 0.907 | 0.907 | 0.912 | 0.913 | 0.903 | 0.902 | **0.919** | 0.918 |
| | 9 | 0.909 | 0.906 | 0.913 | 0.912 | 0.903 | 0.904 | **0.917** | 0.916 |
| | 11 | 0.904 | 0.905 | 0.912 | 0.912 | 0.901 | 0.904 | 0.917 | **0.92** |
| Sensorless | 3 | 0.921 | 0.92 | 0.92 | 0.918 | 0.95 | 0.951 | 0.949 | **0.952** |
| | 5 | 0.916 | 0.914 | 0.911 | 0.904 | **0.945** | 0.944 | 0.934 | 0.932 |
| | 7 | 0.919 | 0.919 | 0.915 | 0.908 | **0.947** | 0.943 | 0.942 | 0.934 |
| | 9 | 0.918 | 0.916 | 0.917 | 0.908 | **0.943** | 0.94 | 0.942 | 0.937 |
| | 11 | 0.918 | 0.922 | 0.914 | 0.91 | **0.942** | 0.939 | 0.941 | 0.935 |
| Crowd Sourced imbalanced | 3 | 0.827 | 0.806 | 0.796 | 0.806 | **0.838** | 0.833 | **0.838** | 0.806 |
| | 5 | 0.84 | 0.827 | 0.817 | 0.802 | **0.875** | 0.869 | 0.845 | 0.838 |
| | 7 | 0.851 | 0.837 | 0.829 | 0.818 | 0.855 | **0.867** | 0.842 | 0.834 |
| | 9 | 0.851 | 0.846 | 0.832 | 0.808 | 0.859 | **0.859** | 0.839 | 0.844 |
| | 11 | 0.846 | 0.871 | 0.827 | 0.842 | **0.876** | 0.861 | 0.861 | 0.838 |
| Crowd Sourced balanced | 3 | 0.892 | 0.833 | 0.864 | 0.782 | **0.915** | 0.858 | 0.879 | 0.814 |
| | 5 | 0.914 | 0.892 | 0.874 | 0.839 | **0.927** | 0.919 | 0.894 | 0.875 |
| | 7 | **0.922** | 0.916 | 0.864 | 0.866 | 0.918 | 0.918 | 0.899 | 0.894 |
| | 9 | 0.916 | **0.921** | 0.86 | 0.855 | 0.915 | 0.916 | 0.893 | 0.902 |
| | 11 | 0.904 | 0.914 | 0.861 | 0.863 | 0.921 | **0.923** | 0.903 | 0.904 |

virtually always generates better results. Statistical tests are performed in order to confirm the importance in the differences in the obtained results *acc*. At first, the values of the classification accuracy in three dependent groups (proposed method, homogeneous ensemble of MPL and ensemble of classifiers KNN, DT, NB) are analyzed. For accuracy the Friedman statistics is

**Table 25. Comparison of classification accuracy *acc* obtained for the proposed method and other methods known from the literature.**

| Dataset | No. tables | Proposed method | Homogeneous ensemble of MPL networks classifiers | Ensemble of classifiers (KNN, DT, NB) |
|---|---|---|---|---|
| Vehicle imbalanced | 3 | 0.739 | **0.74** | 0.709 |
| | 5 | 0.72 | **0.724** | 0.709 |
| | 7 | 0.735 | **0.74** | 0.693 |
| | 9 | **0.752** | 0.685 | 0.717 |
| | 11 | **0.743** | 0.685 | 0.685 |
| Vehicle balanced | 3 | **0.76** | 0.756 | 0.732 |
| | 5 | **0.748** | 0.717 | 0.728 |
| | 7 | **0.765** | 0.728 | 0.752 |
| | 9 | **0.745** | 0.689 | 0.728 |
| | 11 | **0.768** | 0.685 | 0.677 |
| Dry Bean imbalanced | 3 | **0.92** | 0.91 | 0.906 |
| | 5 | **0.918** | 0.903 | 0.902 |
| | 7 | **0.917** | 0.901 | 0.899 |
| | 9 | **0.915** | 0.896 | 0.894 |
| | 11 | **0.914** | 0.901 | 0.9 |
| Dry Bean balanced | 3 | **0.92** | 0.911 | 0.909 |
| | 5 | **0.919** | 0.907 | 0.899 |
| | 7 | **0.919** | 0.905 | 0.9 |
| | 9 | **0.917** | 0.902 | 0.898 |
| | 11 | **0.92** | 0.904 | 0.903 |
| Sensorless | 3 | **0.952** | 0.93 | 0.888 |
| | 5 | **0.945** | 0.9 | 0.912 |
| | 7 | **0.947** | 0.878 | 0.897 |
| | 9 | **0.943** | 0.877 | 0.912 |
| | 11 | **0.942** | 0.86 | 0.926 |
| Crowd Sourced imbalanced | 3 | 0.838 | 0.86 | **0.884** |
| | 5 | **0.875** | 0.825 | 0.833 |
| | 7 | **0.867** | 0.797 | 0.809 |
| | 9 | **0.859** | 0.753 | 0.773 |
| | 11 | **0.876** | 0.735 | 0.763 |
| Crowd Sourced balanced | 3 | **0.915** | 0.879 | 0.866 |
| | 5 | **0.927** | 0.748 | 0.831 |
| | 7 | **0.922** | 0.74 | 0.782 |
| | 9 | **0.921** | 0.673 | 0.718 |
| | 11 | **0.923** | 0.634 | 0.682 |

38.98 with df = 2, $p$ = 0.000001 and we can again reject the null hypothesis. The average ranks are the following: Proposed approach 2.86; Homogeneous ensemble MLP 1.61; Ensemble of classifiers (KNN, DT, NB) 1.53. The critical value of difference of the Nemenyi test between the average ranks of two methods is 0.96. We can claim that classification accuracy of proposed approach is significantly better to all other classifiers (Fig 8). The Wilcoxon-each-pair test confirmed the significant differences between the average accuracy values for all pairs with $p$−value lower than 0.00002 between proposed method and the other analyzed. Also the post-hoc Dunn Bonferroni test confirmed that with $p$ = 0.000001.

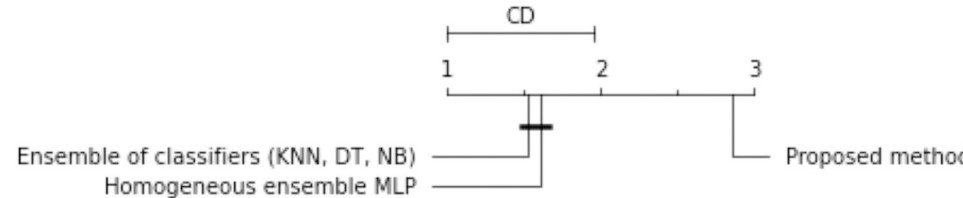

**Fig 8. Critical value of difference (CD) and ranks for the Nemenyi test and accuracy values and methods: Proposed approach; Homogeneous ensemble MLP; Ensemble of classifiers (KNN, DT, NB).** Groups of methods that are not significantly different (with the level of significance at 0.05) are connected.

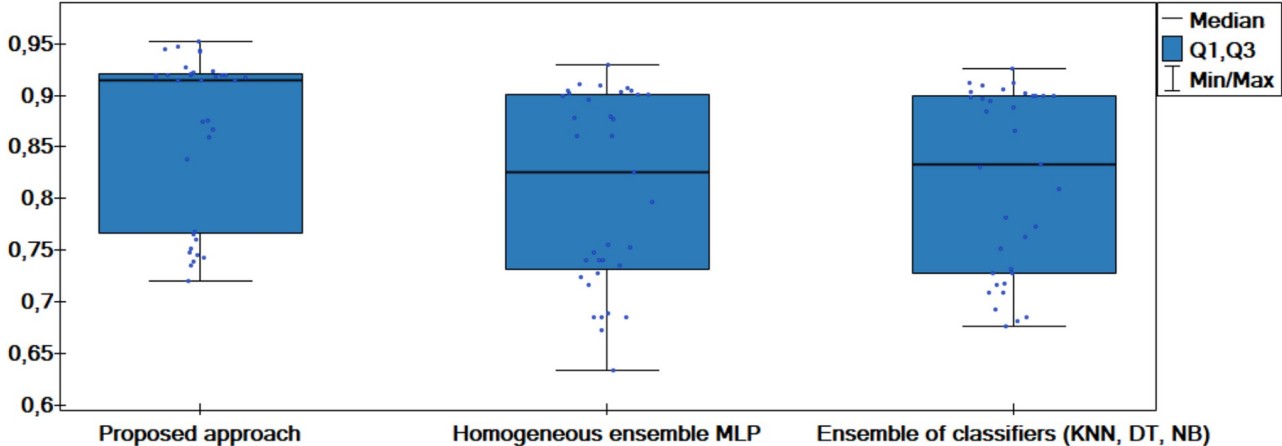

**Fig 9. Box-plot chart with (Median, the first quartile—Q1, the third quartile—Q3) the value of classification accuracy *acc* for the proposed method and the other approaches.**

Additionally, comparative box-plot charts for the values of the classification accuracy and different approaches are created (Fig 9). As can be seen, the proposed approach generates by far the best quality of classification as this is confirmed by the highest positioned box plot and median.

In addition to classification accuracy, other measures are used for comparisons, which are good for unbalanced data and give more reliable comparisons. Table 26 shows the balanced accuracy for the proposed approach and homogeneous ensemble of MLP network classifiers and ensemble of classifiers (KNN, DT, NB) with soft voting. Balanced accuracy is calculated as the average of the sensitivity (true positive rate) for each class in a multiclass classification problem. The best result is shown in bold. Also for balanced accuracy, the proposed method yields the best results. Statistical tests are performed in order to confirm the importance in the differences in the obtained results *bacc*. At first, the values of the balanced accuracy in three dependent groups (proposed method, homogeneous ensemble of MPL and ensemble of classifiers KNN, DT, NB) are analyzed. For accuracy the Friedman statistics is 38 with df = 2, $p = 0.000001$ and we can again reject the null hypothesis. The average ranks are the following: Proposed approach 2.8; Homogeneous ensemble MLP 1.84; Ensemble of classifiers (KNN, DT, NB) 1.36. The critical value of difference of the Nemenyi test between the average ranks of two methods is 0.96. We can claim that classification accuracy of proposed approach is significantly better to all other classifiers (Fig 10). The Wilcoxon-each-pair test confirmed the

**Table 26. Comparison of classification balanced accuracy *bacc* obtained for the proposed method and other methods known from the literature.**

| Data set | No. tables | Proposed method | Homogeneous ensemble of MPL networks classifiers | Ensemble of classifiers (KNN, DT, NB) |
|---|---|---|---|---|
| Vehicle imbalanced | 3 | 0.709 | **0.715** | 0.686 |
| | 5 | 0.694 | **0.722** | 0.69 |
| | 7 | 0.708 | **0.709** | 0.67 |
| | 9 | **0.719** | 0.658 | 0.694 |
| | 11 | **0.717** | 0.66 | 0.672 |
| Vehicle balanced | 3 | **0.731** | 0.729 | 0.705 |
| | 5 | **0.725** | 0.661 | 0.698 |
| | 7 | **0.732** | 0.699 | 0.721 |
| | 9 | **0.721** | 0.649 | 0.7 |
| | 11 | **0.746** | 0.656 | 0.646 |
| Dry Bean imbalanced | 3 | **0.931** | 0.92 | 0.916 |
| | 5 | **0.928** | 0.908 | 0.908 |
| | 7 | **0.928** | 0.906 | 0.905 |
| | 9 | **0.925** | 0.9 | 0.897 |
| | 11 | **0.924** | 0.907 | 0.905 |
| Dry Bean balanced | 3 | **0.931** | 0.926 | 0.919 |
| | 5 | **0.929** | 0.919 | 0.907 |
| | 7 | **0.93** | 0.917 | 0.909 |
| | 9 | **0.926** | 0.913 | 0.907 |
| | 11 | **0.929** | 0.916 | 0.912 |
| Sensorless | 3 | **0.952** | 0.93 | 0.888 |
| | 5 | **0.945** | 0.9 | 0.912 |
| | 7 | **0.947** | 0.878 | 0.897 |
| | 9 | **0.943** | 0.877 | 0.912 |
| | 11 | **0.942** | 0.86 | 0.926 |
| Crowd Sourced imbalanced | 3 | 0.617 | **0.856** | 0.591 |
| | 5 | 0.847 | 0.804 | 0.49 |
| | 7 | 0.654 | **0.801** | 0.452 |
| | 9 | 0.645 | **0.905** | 0.389 |
| | 11 | 0.668 | **0.728** | 0.366 |
| Crowd Sourced balanced | 3 | **0.915** | 0.881 | 0.866 |
| | 5 | **0.927** | 0.793 | 0.831 |
| | 7 | **0.922** | 0.759 | 0.782 |
| | 9 | **0.921** | 0.71 | 0.718 |
| | 11 | **0.923** | 0.752 | 0.682 |

significant differences between the average of balanced accuracy values for two approaches from the literature and the proposed approach with $p$−value lower than 0.004. The difference in average of balanced accuracy is not significant between the approaches the homogeneous ensemble of MLP network and the ensemble of classifiers (KNN, DT, NB). These conclusions are also confirmed graphically in Fig 11. Also the post-hoc Dunn Bonferroni test confirmed that with $p = 0.0002$.

The $F1$−score measure values are compared next (Table 27). This measure provides a balance between precision and recall. It helps evaluate the trade-off between making accurate

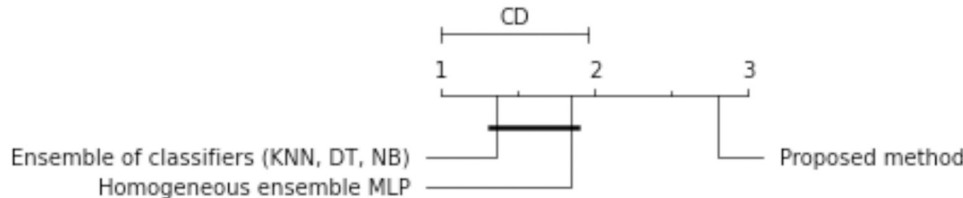

**Fig 10. Critical value of difference (CD) and ranks for the Nemenyi test and balanced accuracy values and methods: Proposed approach; Homogeneous ensemble MLP; Ensemble of classifiers (KNN, DT, NB).** Groups of methods that are not significantly different (with the level of significance at 0.05) are connected.

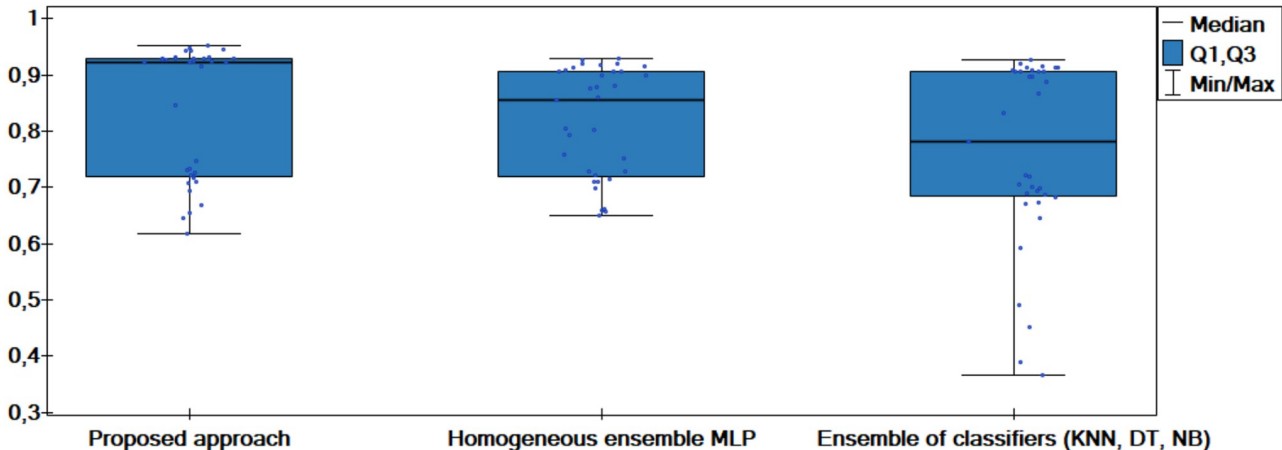

**Fig 11. Box-plot chart with (Median, the first quartile—Q1, the third quartile—Q3) the value of balanced accuracy *bacc* for the proposed method and the other approaches.**

positive predictions (precision) and capturing all positive instances (recall). The *F*1−score is a good choice when you want to find a model that performs well in terms of both precision and recall. As we can see this time, the advantage of the proposed model over those from the literature is even greater. The Friedman test confirmed a statistically significant difference in the results obtained for the considered approaches, $\chi^2(35, 2) = 44.4$, $p = 0.000001$. The average ranks are the following: Proposed approach 2.91; Homogeneous ensemble MLP 1.63; Ensemble of classifiers (KNN, DT, NB) 1.46. The critical value of difference of the Nemenyi test between the average ranks of two methods is 0.96. We can claim that classification accuracy of proposed approach is significantly better to all other classifiers (Fig 12). The Wilcoxon-each-pair test confirmed the presence of significant differences in the average *F*1-score values between the two approaches from the literature and the proposed approach, with a *p*-value of less than 0.00001. These findings are visually reinforced by the data presented in Fig 13. Also the post-hoc Dunn Bonferroni test confirmed that with $p = 0.000001$.

In the last step, the precisions are compared. It quantifies the ability of a model to correctly identify positive instances while minimizing false positives. Precision is particularly important in scenarios where the cost of false positives is high, or when you want to ensure that the positive predictions made by the model are highly reliable. In Table 28 results are compared with the best score highlighted. Here, again, the proposed approach performs much better than the

**Table 27. Comparison of classification $F1$−score obtained for the proposed method and other methods known from the literature.**

| Data set | No. tables | Proposed method | Homogeneous ensemble of MPL networks classifiers | Ensemble of classifiers (KNN, DT, NB) |
|---|---|---|---|---|
| Vehicle imbalanced | 3 | 0.725 | **0.728** | 0.694 |
| | 5 | **0.714** | 0.695 | 0.696 |
| | 7 | **0.727** | 0.722 | 0.677 |
| | 9 | **0.739** | 0.652 | 0.699 |
| | 11 | **0.732** | 0.676 | 0.677 |
| Vehicle balanced | 3 | **0.751** | 0.746 | 0.716 |
| | 5 | **0.73** | 0.642 | 0.711 |
| | 7 | **0.753** | 0.708 | 0.733 |
| | 9 | **0.743** | 0.654 | 0.712 |
| | 11 | **0.762** | 0.662 | 0.65 |
| Dry Bean imbalanced | 3 | **0.92** | 0.91 | 0.906 |
| | 5 | **0.918** | 0.902 | 0.901 |
| | 7 | **0.917** | 0.9 | 0.899 |
| | 9 | **0.915** | 0.895 | 0.893 |
| | 11 | **0.914** | 0.901 | 0.9 |
| Dry Bean balanced | 3 | **0.919** | 0.911 | 0.909 |
| | 5 | **0.919** | 0.907 | 0.898 |
| | 7 | **0.919** | 0.905 | 0.899 |
| | 9 | **0.916** | 0.902 | 0.898 |
| | 11 | **0.919** | 0.904 | 0.902 |
| Sensorless | 3 | **0.952** | 0.93 | 0.881 |
| | 5 | **0.944** | 0.899 | 0.909 |
| | 7 | **0.947** | 0.877 | 0.894 |
| | 9 | **0.943** | 0.876 | 0.911 |
| | 11 | **0.942** | 0.859 | 0.926 |
| Crowd Sourced imbalanced | 3 | 0.84 | 0.867 | **0.874** |
| | 5 | **0.872** | 0.843 | 0.81 |
| | 7 | **0.867** | 0.827 | 0.78 |
| | 9 | **0.859** | 0.832 | 0.732 |
| | 11 | **0.876** | 0.824 | 0.713 |
| Crowd Sourced balanced | 3 | **0.914** | 0.879 | 0.866 |
| | 5 | **0.927** | 0.743 | 0.831 |
| | 7 | **0.922** | 0.742 | 0.78 |
| | 9 | **0.921** | 0.667 | 0.715 |
| | 11 | **0.923** | 0.62 | 0.678 |

others. The Friedman test confirmed a statistically significant difference in the results obtained for the considered approaches, $\chi^2(35, 2) = 32.8$, $p = 0.000001$. Just as before, the Wilcoxon-each-pair test confirmed the presence of significant differences in the average precision values between the two approaches from the literature and the proposed approach, with a $p$-value of less than 0.0001. The post-hoc Dunn Bonferroni test confirmed that differences are significant between the proposed approach and the besline methods with $p = 0.000001$. The average ranks are the following: Proposed approach 2.76; Homogeneous ensemble MLP 1.8; Ensemble of classifiers (KNN, DT, NB) 1.44. The critical value of difference of the Nemenyi test between

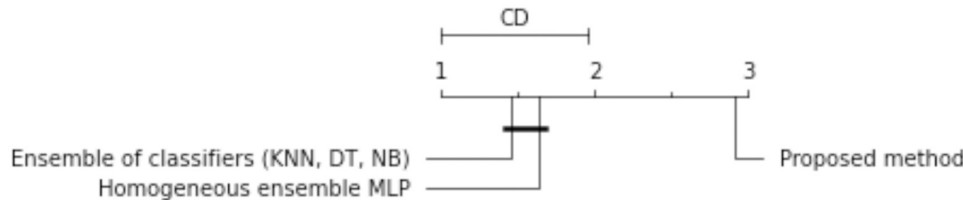

**Fig 12. Critical value of difference (CD) and ranks for the Nemenyi test and F1-scorevalues and methods: Proposed approach; Homogeneous ensemble MLP; Ensemble of classifiers (KNN, DT, NB).** Groups of methods that are not significantly different (with the level of significance at 0.05) are connected.

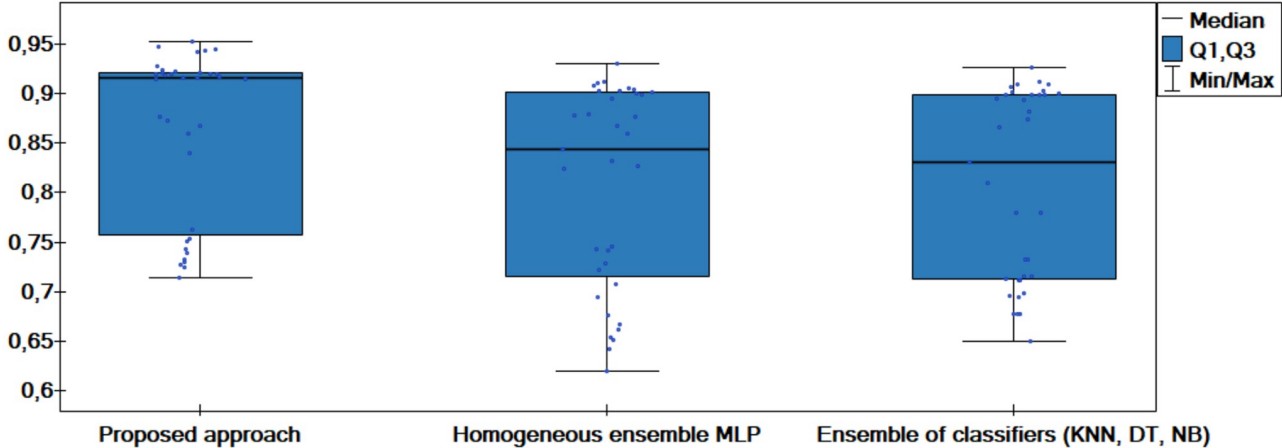

**Fig 13. Box-plot chart with (Median, the first quartile—Q1, the third quartile—Q3) the value of $F1-$score for the proposed method and the other approaches.**

the average ranks of two methods is 0.96. We can claim that classification accuracy of proposed approach is significantly better to all other classifiers (Fig 14). The data presented in Fig 15 visually supports and reinforces these findings.

Based on the preceding analysis, it is clearly evident that the proposed approach consistently delivers very good results. To provide a more in-depth comparative analysis and demonstrate why the proposed model is superior, let us delve into the specific examples illustrating the advantages of the proposed approach. Let us notice once again that the proposed model involves training local MLP networks based on extended local tables (where missing values are imputed), aggregating these local models (using average or sum of weights), and re-training the global model with a shared sample of data. So the result is a single model that is more interpretable and easier to use. On the other hand, ensemble of classifiers (homogeneous MLP or heterogeneous KNN, DT, NB) involves creating base classifiers for each local table. The final decision is made by voting, so we do not get one model—one interpretation. The comparative analysis revealed that the proposed model consistently outperforms the baseline methods across all evaluation criteria. But the advantage of the proposed model obtained due to increased complexity is also justified practically. In the healthcare sector, predicting high-risk patients for sepsis across multiple hospitals is crucial for timely intervention and treatment. Each hospital has its own data set with various patient attributes, some of which may be missing or incomplete. With a single model obtained from the proposed approach, it is enough to

**Table 28. Comparison of classification precision obtained for the proposed method and other methods known from the literature.**

| Data set | No. tables | Proposed method | Homogeneous ensemble of MPL networks classifiers | Ensemble of classifiers (KNN, DT, NB) |
|---|---|---|---|---|
| Vehicle imbalanced | 3 | **0.725** | **0.725** | 0.701 |
| | 5 | 0.711 | **0.761** | 0.718 |
| | 7 | **0.735** | 0.73 | 0.69 |
| | 9 | **0.746** | 0.655 | 0.708 |
| | 11 | **0.74** | 0.67 | 0.705 |
| Vehicle balanced | 3 | **0.756** | 0.742 | 0.733 |
| | 5 | **0.748** | 0.582 | 0.726 |
| | 7 | **0.755** | 0.711 | 0.748 |
| | 9 | **0.744** | 0.66 | 0.718 |
| | 11 | **0.768** | 0.683 | 0.667 |
| Dry Bean imbalanced | 3 | **0.921** | 0.911 | 0.908 |
| | 5 | **0.918** | 0.903 | 0.904 |
| | 7 | **0.917** | 0.902 | 0.901 |
| | 9 | **0.916** | 0.898 | 0.897 |
| | 11 | **0.915** | 0.902 | 0.902 |
| Dry Bean balanced | 3 | **0.92** | 0.911 | 0.91 |
| | 5 | **0.92** | 0.907 | 0.9 |
| | 7 | **0.919** | 0.906 | 0.901 |
| | 9 | **0.917** | 0.903 | 0.899 |
| | 11 | **0.92** | 0.905 | 0.903 |
| Sensorless | 3 | **0.953** | 0.931 | 0.906 |
| | 5 | **0.945** | 0.9 | 0.921 |
| | 7 | 0.948 | 0.878 | 0.904 |
| | 9 | **0.943** | 0.879 | 0.918 |
| | 11 | **0.943** | 0.863 | 0.928 |
| Crowd Sourced imbalanced | 3 | 0.847 | **0.886** | 0.877 |
| | 5 | 0.874 | **0.879** | 0.818 |
| | 7 | 0.872 | **0.889** | 0.789 |
| | 9 | 0.868 | **0.96** | 0.74 |
| | 11 | 0.881 | **0.969** | 0.719 |
| Crowd Sourced balanced | 3 | **0.915** | 0.882 | 0.874 |
| | 5 | **0.927** | 0.772 | 0.837 |
| | 7 | **0.923** | 0.76 | 0.789 |
| | 9 | **0.922** | 0.684 | 0.726 |
| | 11 | **0.924** | 0.748 | 0.691 |

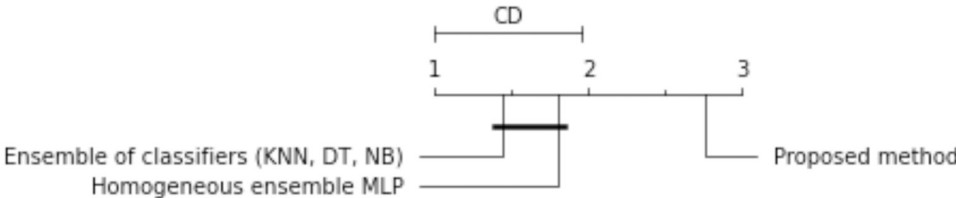

**Fig 14. Critical value of difference (CD) and ranks for the Nemenyi test and precision values and methods: Proposed approach; Homogeneous ensemble MLP; Ensemble of classifiers (KNN, DT, NB).** Groups of methods that are not significantly different (with the level of significance at 0.05) are connected.

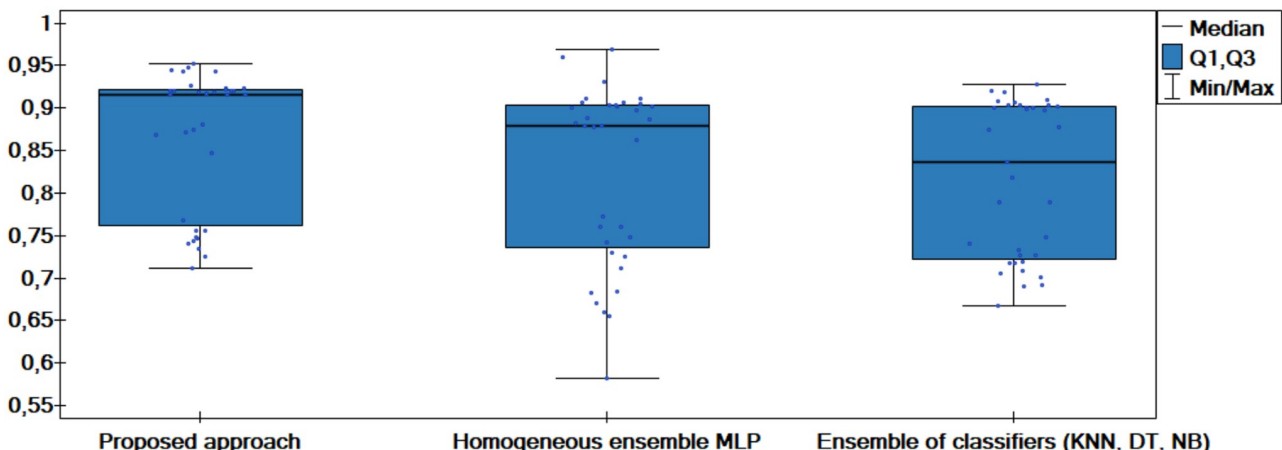

**Fig 15. Box-plot chart with (Median, the first quartile—Q1, the third quartile—Q3) the value of precision for the proposed method and the other approaches.**

check all values on the attributes of the global model of the diagnosed patient—without having to refer to local hospitals and their databases. In smart agriculture, yield prediction is essential for effective farm management and planning. Multiple farms collect data on various attributes affecting crop yield, but these data sets can be incomplete or missing certain information. Also in this case the proposed model outperforms the baseline methods in integrating diverse and incomplete data from multiple farms to improve predictive accuracy, aiding farmers in making more informed decisions. These case studies illustrate the practical benefits and reliability of the proposed model in real-world applications, such as healthcare and agriculture, where accurate predictions are crucial for improving outcomes.

Additionally, AUCROC charts are prepared to demonstrate that the proposed approach outperforms those known from the literature. Due to limited space, we do not present the graphs for all thirty-five dispersed data sets. Figs 16 and 17 shows AUCROC graphs for Crowd Sourced imbalanced and balanced data sets, as well as all versions of dispersion—with 3, 5, 7, 9, and 11 local tables (since the proposed approach performed the worst for these data sets). Each row first displays the curve plot for the homogeneous ensemble of MPL networks classifiers, followed by the ensemble of classifiers (KNN, DT, NB), and finally, the proposed approach. Since the analyzed data sets are multi-class data, the graph shows the ROC curves for each decision class versus to the others, as well as the averaged ROC curve. It is clear that the proposed approach outperforms the other approaches. We can confidently say that for dispersed data, building a global neural network yields better results than using either heterogeneous or homogeneous ensembles of classifiers.

## 6 Conclusions

This paper proposes a new method for generating MLP global networks based on dispersed data with different sets of attributes. This method involves generating local MLP neural networks with identical structure based on local tables. Artificial objects based on the original objects are generated in order to realize that. In the next step, the networks are aggregated using appropriate weights proportional to the classification accuracy of the local models and one of two proposed methods—sum and average. The paper shows that the proposed model generated better results that other methods known from the literature. In addition, it is verified

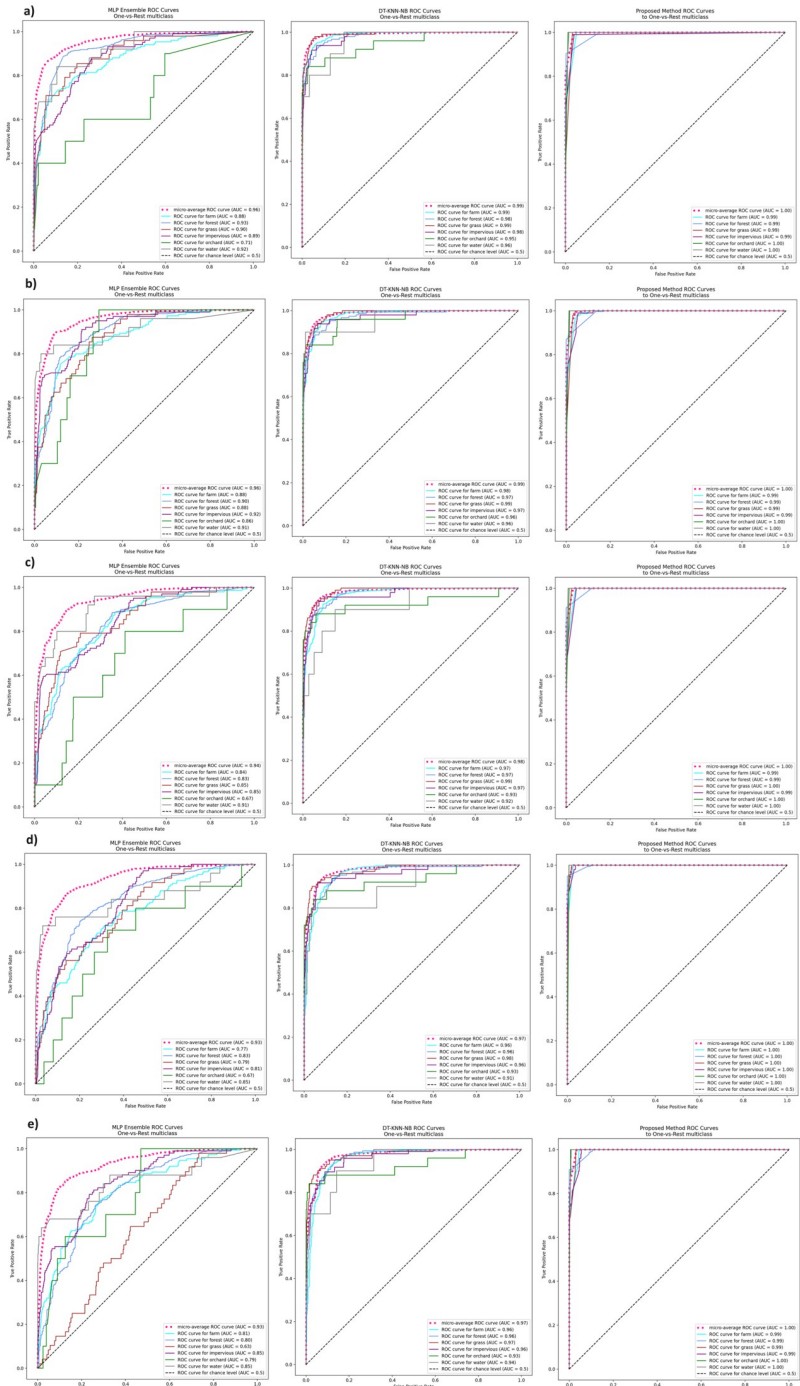

**Fig 16.** AUCROC graph for Crowd Sourced imbalanced data sets and all versions of dispersion: a) 3 local tables, b) 5 local tables, c) 7 local tables, d) 9 local tables, e) 11 local tables and three different approaches: first row graph—homogeneous ensemble of MPL networks classifiers, second row graph—ensemble of classifiers (KNN, DT, NB), third row graph—proposed approach.

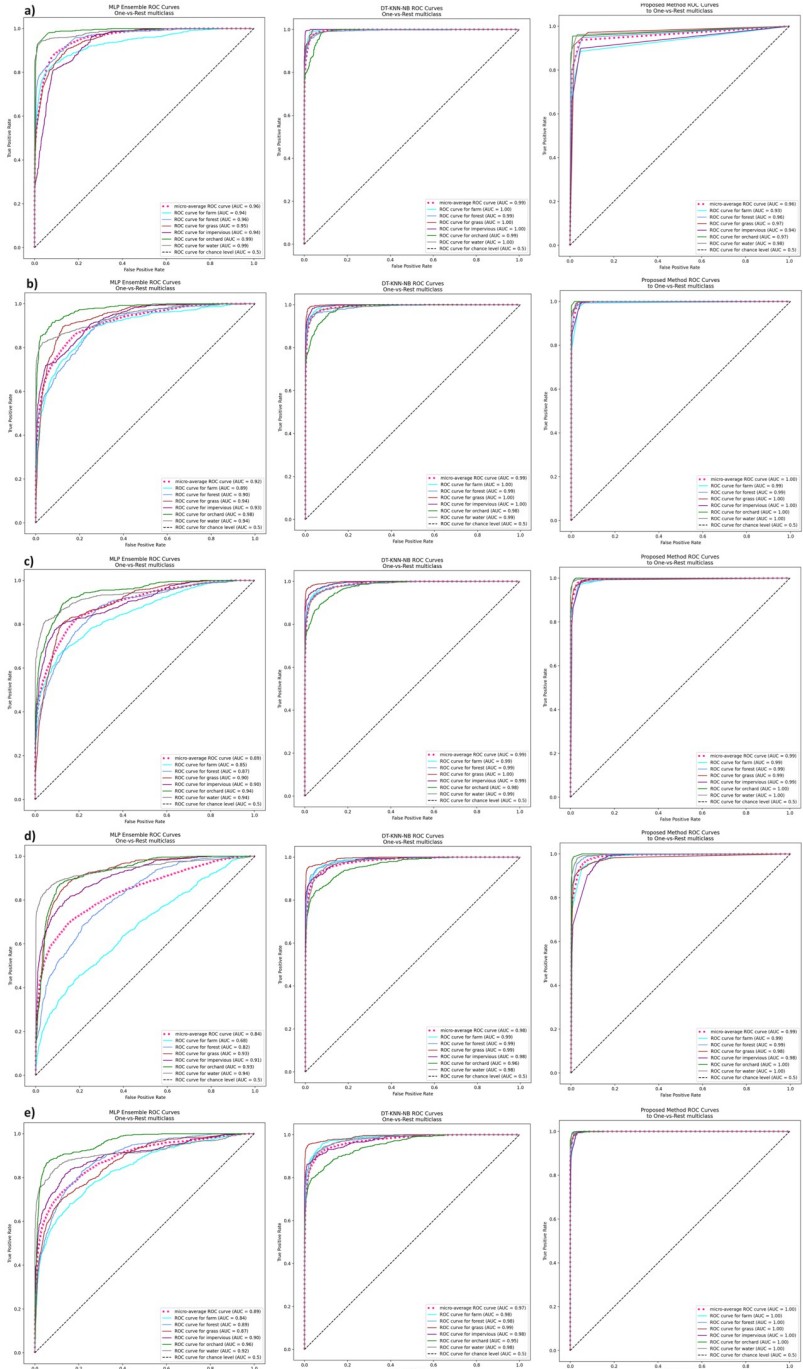

**Fig 17.** AUCROC graph for Crowd Sourced balanced data sets and all versions of dispersion: a) 3 local tables, b) 5 local tables, c) 7 local tables, d) 9 local tables, e) 11 local tables and three different approaches: first row graph—homogeneous ensemble of MPL networks classifiers, second row graph—ensemble of classifiers (KNN, DT, NB), third row graph—proposed approach.

that on average, the best quality is achieved using only one artificial object with two hidden layers.

While the proposed model demonstrates significant improvements over traditional ensemble classifiers and homogeneous ensembles of MLPs, it is important to acknowledge certain limitations that could impact its performance and applicability in various scenarios. The model's ability to handle missing attributes heavily relies on the quality of the imputation methods used. If the imputation process introduces biases or inaccuracies, it can adversely affect the overall performance of the model. As the number of local tables increases, the model's ability to efficiently aggregate and re-train the global model might become a bottleneck. The model's performance is sensitive to the choice of parameters, such as the number of hidden layers and neurons, the method of aggregating local networks, and the strategies for handling missing data. Identifying the optimal settings requires extensive experimentation, which may not always be feasible. Developing automated parameter tuning methods or adaptive algorithms could mitigate this limitation. An important limitation of the method is also the need for a validation set, which must contain the combined characteristics of object—descriptions from the perspective of all local tables.

In further works, it is planned to use conflict analysis and coalitions of local networks to generate a global model as well as to develop a method for generating the artificial objects necessary for the global network's re-training stage. Also, it is planned to use other neural network architectures in the proposed approach.

## Author Contributions

**Conceptualization:** Małgorzata Przybyła-Kasperek, Kwabena Frimpong Marfo.

**Formal analysis:** Małgorzata Przybyła-Kasperek, Kwabena Frimpong Marfo.

**Investigation:** Małgorzata Przybyła-Kasperek, Kwabena Frimpong Marfo.

**Methodology:** Małgorzata Przybyła-Kasperek, Kwabena Frimpong Marfo.

**Project administration:** Małgorzata Przybyła-Kasperek.

**Software:** Kwabena Frimpong Marfo.

**Supervision:** Małgorzata Przybyła-Kasperek.

**Validation:** Małgorzata Przybyła-Kasperek, Kwabena Frimpong Marfo.

**Visualization:** Małgorzata Przybyła-Kasperek, Kwabena Frimpong Marfo.

**Writing – original draft:** Małgorzata Przybyła-Kasperek.

**Writing – review & editing:** Małgorzata Przybyła-Kasperek, Kwabena Frimpong Marfo.

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
