## [Decision Letter · Decision Letter 0]

11 Jun 2024

PONE-D-23-41488A Multi-Layer Perceptron Neural Network for Varied Conditional Attributes in Tabular Dispersed DataPLOS ONE

Dear Dr. Przybyla-Kasperek,

Thank you for submitting your manuscript to PLOS ONE. After careful consideration, we feel that it has merit but does not fully meet PLOS ONE’s publication criteria as it currently stands. Therefore, we invite you to submit a revised version of the manuscript that addresses the points raised during the review process. Please submit your revised manuscript by Jul 26 2024 11:59PM. If you will need more time than this to complete your revisions, please reply to this message or contact the journal office at plosone@plos.org. Please include the following items when submitting your revised manuscript:A rebuttal letter that responds to each point raised by the academic editor and reviewer(s). You should upload this letter as a separate file labeled 'Response to Reviewers'.A marked-up copy of your manuscript that highlights changes made to the original version. You should upload this as a separate file labeled 'Revised Manuscript with Track Changes'.An unmarked version of your revised paper without tracked changes. You should upload this as a separate file labeled 'Manuscript'.

We look forward to receiving your revised manuscript.

Kind regards,

Dariusz Siudak, Ph.D., DSc.

Academic Editor

PLOS ONE

Reviewers' comments:

Reviewer's Responses to Questions

**Comments to the Author**

1. Is the manuscript technically sound, and do the data support the conclusions?

Reviewer #1: Yes

Reviewer #2: Yes

Reviewer #3: Yes

Reviewer #4: Yes

2. Has the statistical analysis been performed appropriately and rigorously? 

Reviewer #1: Yes

Reviewer #2: Yes

Reviewer #3: Yes

Reviewer #4: Yes

3. Have the authors made all data underlying the findings in their manuscript fully available?

Reviewer #1: Yes

Reviewer #2: Yes

Reviewer #3: Yes

Reviewer #4: Yes

4. Is the manuscript presented in an intelligible fashion and written in standard English?

Reviewer #1: Yes

Reviewer #2: Yes

Reviewer #3: Yes

Reviewer #4: Yes

5. Review Comments to the Author

Reviewer #1: This paper proposed A Multi-Layer Perceptron Neural Network for Varied Conditional Attributes in Tabular Dispersed Data. I acknowledged the novelty of this paper. However, this paper require major revisions.

1. The introduction is very vague. It is full of citation without proper justification. The author should discuss in detail each citation that contributed to the work.

2. The discussion of recent MLP should be include in Section 2.

3. The author should include more recent works from 2022 - 2023 in Section 1 and Section 2.

4. The author should include a schematic diagram of the MLP used in the proposed model. The author may refer/ cite to the following paper regarding the schematic diagram:

(a) Weighted Random k Satisfiability for k = 1,2 (r2SAT) in Discrete Hopfield Neural Network

5. The author should analyze more number of dataset.

6. Fig 9 is awkwardly placed in Page 37.

7. I strongly suggest the author avoid short paragraphs throughout the paper.

8. The experimental setup of the conducted simulation is too simple and vague. I suggest the author include a section on Experimental Setup which include the following:

(a) Simulation Design

(b) Dataset

(c) Parameter Assignments

(d) Formulations of the Performance Metrics

(e) Reproducibility of the proposed model

(f) Baseline methods

The author may refer/cite to this paper: Multi-discrete genetic algorithm in hopfield neural network with weighted random k satisfiability

9. Can the author justify how the parameters in MLP were chosen? If possible, can the author conduct a parameter tuning process?

10. I suggest the author conduct more comparative analysis with other Artificial Neural Network.

11. The author should consider Friedman test/other non-parametric test to validate the superiority of the proposed model as compared to other algorithms/baselines. Kindly refer/cite the following papers on how to do the Friedman test

(a) YRAN2SAT: A novel flexible random satisfiability logical rule in discrete hopfield neural network

(b) Random Maximum 2 Satisfiability Logic in Discrete Hopfield Neural Network Incorporating Improved Election Algorithm

(c) A modified reverse-based analysis logic mining model with Weighted Random 2 Satisfiability logic in Discrete Hopfield Neural Network and multi-objective training of Modified Niched Genetic Algorithm

12. Can the author justify the complexity value of the proposed model?

Reviewer #2: 1. Elaborate on how data is collected and organized in local tables. Clarifying the selection criteria for objects and attributes in these tables will help readers understand the foundation of your local models.

2. Provide more detailed explanations and methodological steps for the development of local models, including how artificial objects are integrated.

3. Expand on the weighted techniques used for aggregating local models. Detail the criteria for weighting and how they influence the final model performance.

4. Describe the retraining process with global objects more thoroughly. Specify how this step integrates with the overall model development and its impact on model accuracy.

5. While you mention outperforming existing approaches, additional details about these methods and why your approach is superior would be valuable. Include a more in-depth comparative analysis or case studies.

6. Provide more robust statistical evidence to support the claims of superiority in F1-score and precision. This could include confidence intervals, p-values, or other statistical tests.

7. Discuss the practical applications of your approach. How can it be implemented in real-world scenarios, and what are the potential challenges or limitations?

8. The literature review section could effectively start by discussing the profound influence of artificial intelligence (AI) across multiple sectors. Such as:

Artificial intelligence (AI) has revolutionized a range of fields by incorporating human-like capabilities such as learning, reasoning, and perception into software systems. This technological progression has empowered computers to perform tasks traditionally handled by humans. Boosted by advancements in computing power, the availability of extensive datasets, and the development of cutting-edge AI algorithms, AI applications are now widespread. Notable applications include finger vein recognition [1], diabetic retinopathy detection [2-6], RNA Engineering [7-8], cancer detection [9-11], biomathematical challenges [12,13], and smart agriculture [14].

1.Finger-vein recognition using a novel enhancement method with convolutional neural network

2.Improved Support Vector Machine based on CNN-SVD for vision-threatening diabetic retinopathy detection and classification

3.NIMEQ-SACNet: A novel self-attention precision medicine model for vision-threatening diabetic retinopathy using image data

4.EdgeSVDNet: 5G-Enabled Detection and Classification of Vision-Threatening Diabetic Retinopathy in Retinal Fundus Images

5.AI-Based Automatic Detection and Classification of Diabetic Retinopathy Using U-Net and Deep Learning

6.Diabetic Retinopathy Detection and Classification Using Mixed Models for a Disease Grading Database

7.iDNA-OpenPrompt: OpenPrompt learning model for identifying DNA methylation

8.Advancing single-cell RNA-seq data analysis through the fusion of multi-layer perceptron and graph neural network

9.BC-QNet: A Quantum-Infused ELM Model for Breast Cancer Diagnosis

10.IGWO-IVNet3: DL-based automatic diagnosis of lung nodules using an improved gray wolf optimization and InceptionNet-V3

11.Lung nodules detection using weighted filters and classification using CNN

12.Neuro-optimized numerical treatment of HIV infection model

13.Neuro-optimized numerical solution of non-linear problem based on Flierl–Petviashivili equation

14.Increasing Crop Quality and Yield with a Machine Learning-Based Crop Monitoring System

Reviewer #3: The authors propose an MLP neural networks and disparate data sources to construct a global model. These sources are independently collected in separate local tables, each potentially containing different objects and attributes, but with some shared elements. The approach involves building local models based on these tables, supplemented with artificial objects. The model is novel and may gain so many intersts. However, I have the followings concerns:

- More details (2 - 3 extra statements) should be added to the abstract highlighting the importance, and abit of detailed resuls, if possible.

-How this model avoided overfitting, detailed answer of how the authors ensure avoid overfitting must include checking the performance for training and testing performance, at which point the model start overfitting.

-AUCROC must be plotted for the model and the comparison methods.

-More about the hyper-parameter and how they were obtained/optimized.

- The conclusion or discussion may detailed the limitations of the model.

Reviewer #4: The authors have done the work with the novelty and the results are presented in a well standard format. But the English languages need to be improved in overall of the paper. Specially, the introduction section needs to be improved.

6. PLOS authors have the option to publish the peer review history of their article (what does this mean?). If published, this will include your full peer review and any attached files.

Reviewer #1: No

Reviewer #2: No

Reviewer #3: **Yes: **Abedalrhman Alkhateeb

Reviewer #4: No

---

## [Author Response · Author response to Decision Letter 0]

18 Jul 2024

All responses were written in the file Responses to Reviewers

---

## [Decision Letter · Decision Letter 1]

11 Sep 2024

A Multi-Layer Perceptron Neural Network for Varied Conditional Attributes in Tabular Dispersed Data

PONE-D-23-41488R1

Dear Dr. Przybyla-Kasperek,

We’re pleased to inform you that your manuscript has been judged scientifically suitable for publication and will be formally accepted for publication once it meets all outstanding technical requirements.

Kind regards,

Kalapraveen Bagadi

Academic Editor

PLOS ONE

Additional Editor Comments (optional):

Comments from PLOS Editorial Office: We note that one or more reviewers has recommended that you cite specific previously published works in an earlier round of revision. As always, we recommend that you please review and evaluate the requested works to determine whether they are relevant and should be cited. It is not a requirement to cite these works and you may remove them before the manuscript proceeds to publication. We appreciate your attention to this request.

Reviewers' comments:

Reviewer's Responses to Questions

**Comments to the Author**

1. If the authors have adequately addressed your comments raised in a previous round of review and you feel that this manuscript is now acceptable for publication, you may indicate that here to bypass the “Comments to the Author” section, enter your conflict of interest statement in the “Confidential to Editor” section, and submit your "Accept" recommendation.

Reviewer #1: All comments have been addressed

Reviewer #4: All comments have been addressed

2. Is the manuscript technically sound, and do the data support the conclusions?

Reviewer #1: Yes

Reviewer #4: Yes

3. Has the statistical analysis been performed appropriately and rigorously? 

Reviewer #1: Yes

Reviewer #4: Yes

4. Have the authors made all data underlying the findings in their manuscript fully available?

Reviewer #1: Yes

Reviewer #4: Yes

5. Is the manuscript presented in an intelligible fashion and written in standard English?

Reviewer #1: Yes

Reviewer #4: Yes

6. Review Comments to the Author

Reviewer #1: The author has addressed all my previous comment. The data obtained from this paper will be useful for the future readers.

Reviewer #4: The authors revised the manuscript as per the reviewers comments. The manuscript may be accepted for publication.

7. PLOS authors have the option to publish the peer review history of their article (what does this mean?). If published, this will include your full peer review and any attached files.

Reviewer #1: No

Reviewer #4: No

---

## [Editor Report · Acceptance letter]

16 Sep 2024

PONE-D-23-41488R1 

PLOS ONE

Dear Dr. Przybyła-Kasperek, 

I'm pleased to inform you that your manuscript has been deemed suitable for publication in PLOS ONE. Congratulations! Your manuscript is now being handed over to our production team.

Kind regards, 

on behalf of

Dr. Kalapraveen Bagadi 

Academic Editor

PLOS ONE